# Structural basis for FGF hormone signalling

Lingfeng Chen[1,2,10], Lili Fu[1,3,4,10], Jingchuan Sun[1,5,10], Zhiqiang Huang[1,3,10], Mingzhen Fang[1,3,10], Allen Zinkle[6], Xin Liu[1,3], Junliang Lu[1,3], Zixiang Pan[1,3], Yang Wang[1,7], Guang Liang[2], Xiaokun Li[1,4,8✉], Gaozhi Chen[1,3,9✉] & Moosa Mohammadi[1,3✉]

α/βKlotho coreceptors simultaneously engage fibroblast growth factor (FGF) hormones (FGF19, FGF21 and FGF23)[1,2] and their cognate cell-surface FGF receptors (FGFR1–4) thereby stabilizing the endocrine FGF–FGFR complex[3–6]. However, these hormones still require heparan sulfate (HS) proteoglycan as an additional coreceptor to induce FGFR dimerization/activation and hence elicit their essential metabolic activities[6]. To reveal the molecular mechanism underpinning the coreceptor role of HS, we solved cryo-electron microscopy structures of three distinct 1:2:1:1 FGF23–FGFR–αKlotho–HS quaternary complexes featuring the 'c' splice isoforms of FGFR1 (FGFR1c), FGFR3 (FGFR3c) or FGFR4 as the receptor component. These structures, supported by cell-based receptor complementation and heterodimerization experiments, reveal that a single HS chain enables FGF23 and its primary FGFR within a 1:1:1 FGF23–FGFR–αKlotho ternary complex to jointly recruit a lone secondary FGFR molecule leading to asymmetric receptor dimerization and activation. However, αKlotho does not directly participate in recruiting the secondary receptor/dimerization. We also show that the asymmetric mode of receptor dimerization is applicable to paracrine FGFs that signal solely in an HS-dependent fashion. Our structural and biochemical data overturn the current symmetric FGFR dimerization paradigm and provide blueprints for rational discovery of modulators of FGF signalling[2] as therapeutics for human metabolic diseases and cancer.

The mammalian fibroblast growth factor (FGF) family comprises 18 β-trefoil homology-domain-containing polypeptides arranged into five paracrine subfamilies and one endocrine subfamily[7]. Paracrine subfamilies govern multiple events during embryonic development[8] whereas endocrine subfamily members (FGF19, FGF21 and FGF23) are hormones that regulate bile acid, lipid, glucose, vitamin D and mineral ion homeostasis[1,4,9,10]. FGF hormones are promising targets for the treatment of a spectrum of metabolic diseases, including type II diabetes, obesity, non-alcoholic steatohepatitis, primary biliary cirrhosis, bile acid diarrhoea, renal phosphate wasting disorders and chronic kidney disease[2,11–19]. FGFs mediate their actions by binding, dimerizing and thereby activating single-pass transmembrane FGF receptor tyrosine kinases (FGFR1–4)[20,21]. The extracellular region of a prototypical FGFR contains three immunoglobulin (Ig)-like domains (D1, D2 and D3). D2, D3 and the short D2–D3 linker are necessary and sufficient for ligand binding and receptor dimerization. In FGFR1–FGFR3, alternative splicing of two mutually exclusive exons (termed 'b' and 'c') alters the composition of major ligand-binding sites within the D3 domains of these three FGFRs, effectively expanding the number of principal FGFR isoforms to seven (that is, FGFR1b–3b, FGFR1c–3c and FGFR4)[22–24].

Paracrine FGFs depend on heparan sulfate (HS) glycosaminoglycans as a mandatory coreceptor to stably bind and dimerize their cognate FGFRs. HS are linear glycan chains of HS proteoglycans (HSPGs) that are abundantly expressed in the extracellular matrix of all tissues[25]. According to the crystal structure of the 2:2:2 FGF2–FGFR1c–HS dimer, HS concurrently engages the HS binding sites of paracrine FGF and FGFR, thereby enforcing FGF–FGFR proximity. In doing so, HS (1) enhances 1:1 FG–FGFR binding affinity; and (2) fortifies interactions between two 1:1 complexes to give rise to two-fold symmetric 2:2 dimers[26]. Extracellular FGFR dimerization promotes the formation of a thermodynamically weak asymmetric complex of the intracellular kinase domains[27] that mediates (A)-loop tyrosine transphosphorylation and hence kinase activation and intracellular signalling.

The HS binding sites of FGF hormones diverge both compositionally and conformationally from those of paracrine FGFs, dramatically weakening their affinity for HS[28]. Consequently, FGF hormones avoid entrapment by the HSPGs in the extracellular matrix and can enter the circulation. Moreover, FGF hormones have weak affinity for FGFRs[28,29] owing to substitutions of their key receptor binding residues. While these structural and biochemical idiosyncrasies confer hormonal mode of action, they render HS insufficient for endocrine FGF to bind FGFR and induce receptor dimerization. Indeed, to offset these deficiencies, FGF hormones have evolved an absolute dependency on αKlotho or βKlotho as an additional coreceptor for signalling. α- and βKlotho are

[1]Oujiang Laboratory (Zhejiang Lab for Regenerative Medicine, Vision, and Brain Health), School of Pharmaceutical Sciences, Wenzhou Medical University, Wenzhou, China. [2]School of Pharmaceutical Sciences, Hangzhou Medical College, Hangzhou, China. [3]Institute of Cell Growth Factor, Oujiang Laboratory (Zhejiang Lab for Regenerative Medicine, Vision, and Brain Health), Wenzhou, China. [4]State Key Laboratory for Macromolecule Drugs and Large-scale Preparation, Wenzhou Medical University, Wenzhou, China. [5]Laboratory of Cell Fate Control, School of Life Sciences, Westlake University, Hangzhou, China. [6]Department of Physiology and Cellular Biophysics, Columbia University Irving Medical Center, New York, NY, USA. [7]Center of Biomedical Physics, Wenzhou Institute, University of Chinese Academy of Sciences, Wenzhou, China. [8]National Engineering Research Center of Cell Growth Factor Drugs and Protein Biologics, Wenzhou Medical University, Wenzhou, China. [9]Institute of chronic kidney disease, Wenzhou Medical University, Wenzhou, China. [10]These authors contributed equally: Lingfeng Chen, Lili Fu, Jingchuan Sun, Zhiqiang Huang, Mingzhen Fang. ✉e-mail: lixk1964@163.com; gaozhichen@wmu.edu.cn; mohammadimoosa@gmail.com

single-pass transmembrane proteins with a large extracellular domain comprising two tandem glycosidase-like domains (KL1 and KL2), and a short intracellular domain[3,5]. αKlotho (or βKlotho) coreceptor simultaneously binds the FGF hormone and its cognate FGFR, thereby enforcing binding of endocrine FGF to FGFR. Klotho coreceptors have unique FGF hormone and FGFR binding specificities which ultimately dictate FGFR binding specificity and target tissue/organ selectivity of endocrine FGFs. αKlotho binds exclusively to FGF23 (ref. 30) whereas βKlotho binds both FGF19 and FGF21 (refs. 31–34). With respect to FGFR interaction, αKlotho and βKlotho show shared specificity for FGFR1c and FGFR4 but neither recognizes the 'b' splice isoforms of FGFR1–3. However, they display opposite specificity towards FGFR2c and FGFR3c with αKlotho binding FGFR3c but not FGFR2c, while βKlotho binds FGFR2c but not FGFR3c (ref. 35).

The coreceptor mechanism of αKlotho in FGF23 signalling was illuminated by the crystal structure of a ternary complex comprising FGF23, the FGFR1c ligand-binding domain[6] and the soluble ectodomain of αKlotho (a naturally occurring isoform produced via shedding of the transmembrane form)[36,37]. In the structure, the long α1β1 loop (denoted receptor binding arm (RBA)) extends from the KL2 domain of αKlotho and clasps a hydrophobic groove in the D3 domain of FGFR (a conserved hallmark of FGFR1c–3c and FGFR4), while a large cleft at the junction between KL1 and KL2 embraces FGF23's long C-tail. In doing so, αKlotho enforces FGF23–FGFR proximity and complex stability. As to the role of HS, we previously postulated that HS enables two 1:1:1 FGF23–FGFR–αKlotho complexes to assemble into a symmetric 2:2:2:2 FGF23–FGFR–αKlotho–HS quaternary signalling unit reminiscent of 2:2:2 paracrine FGF–FGFR–HS dimers[6]. To establish the mechanism underlying dual coreceptor (that is, αKlotho and HS) dependent FGFR dimerization/activation by FGF hormones, we solved cryogenic electron microscopy (cryo-EM) structures of all three physiologically possible FGF23–FGFR–αKlotho–HS quaternary complexes. Unexpectedly, these structures, supported by comprehensive cell-based data, reveal that HS fortifies interactions of FGF23 and its primary FGFR within a 1:1:1 FGF23–FGFR–αKlotho with a secondary lone FGFR, thereby inducing asymmetric receptor dimerization and activation. Notably, the asymmetric mode of receptor dimerization is generalizable to paracrine FGFs, which thus overturns our current symmetric model for FGFR dimerization and activation[26].

## 1:2:1:1 FGF23–FGFR–αKlotho–HS complexes

We prepared 1:1:1:1 quaternary mixtures of full-length mature human FGF23, the extracellular ligand-binding portions (that is, encompassing D2 and D3 domains) of three human cognate FGFRs of FGF23 (that is, FGFR1c, FGFR3c and FGFR4), whole ectodomain of human αKlotho coreceptor and a fully sulfated heparin dodecasaccharide (hereafter, referred to as HS). The three resulting quaternary complexes (that is, FGF23–FGFR1c–αKlotho–HS, FGF23–FGFR3c–αKlotho–HS and FGF23–FGFR4–αKlotho–HS) were isolated by size-exclusion chromatography (SEC) and were used directly for vitrification, cryo-EM image collection and structure determination (Extended Data Fig. 1, Extended Data Table 1 and Supplementary Fig. 2). Contrary to our prediction, all three cryo-EM structures reveal identical asymmetric 1:2:1:1 FGF23–FGFR–αKlotho–HS quaternary assemblies in which HS enables a 1:1:1 FGF23–FGFR–αKlotho ternary complex to recruit a lone FGFR chain (termed FGFR[S] to differentiate it from the 'primary' receptor FGFR[P] within the ternary complex) (Fig. 1). At the membrane distal end of each quaternary complex, HS engages in tripartite interactions with juxtaposed HS binding sites of FGF23, FGFR[P] and FGFR[S] (Fig. 2a). In doing so, HS augments interactions of FGF23 and FGFR[P] from the FGF23–FGFR[P]–αKlotho ternary complex with the FGFR[S] chain, and thus induces receptor (that is, FGFR[P]–FGFR[S]) dimerization (Fig. 3a). Notably, the proximity and orientation (perpendicular to plasma membrane) of C-terminal ends of FGFR chains (approximately 25 Å

apart) acquiesces with the formation of an A-loop transphosphorylating asymmetric dimer of intracellular kinase domain[27] (Extended Data Fig. 2a). Although αKlotho does not directly engage FGFR[S], it presides over the FGFR[S] recruitment by stabilizing the FGF23–FGFR[P] complex. Without αKlotho's assistance, HS cannot independently generate a stable FGF23–FGFR[P] complex that is necessary to recruit a secondary FGFR[S]. Indeed, cell-based experiments confirm that FGF23 signalling is strictly αKlotho dependent (Extended Data Fig. 2b). The conformation of the FGF23–FGFR[P]–αKlotho ternary complex within the 1:2:1:1 FGF23–FGFR–αKlotho–HS quaternary assemblies is similar to that of the X-ray structure of HS-free FGF23–FGFR–αKlotho ternary complex (root mean square deviation (RMSD) of only 1.16 Å, Extended Data Fig. 2c)[6]. However, the FGFR[S] component of the quaternary complex adopts a distorted conformation that is incompatible with ligand binding, which thus explains the distinct asymmetry of the quaternary complex (Supplementary Fig. 4).

## Tripartite interactions of HS

FGF23–HS interactions are mediated by three residues (Arg48, Arg140 and Tyr154) from the atypical HS binding site within the core of FGF23 (Fig. 2b). Compared with FGF23–HS contacts, HS interacts more extensively with HBS in the D2 domains of the two FGFR chains, involving residues Lys177, Lys207, Arg209 and Ser214 of FGFR1c[P] and Lys175, Lys177, Val208, Arg209 and Thr212 of FGFR1c[S] (Fig. 2b). To validate the physiological importance of three-way interactions of HS with FGF23, FGFR1c[P] and FGFR1c[S] for the induction of receptor (that is, FGFR[P]–FGFR[S]) dimerization, we introduced an R48A/R140A double mutation into FGF23 (FGF23[ΔHBS]) and K175Q/K177Q and K207Q/R209Q double mutations (FGFR1c[ΔHBS1] and FGFR1c[ΔHBS2]) separately or in combination (K175Q/K177Q/K207Q/R209Q termed FGFR1c[ΔHBS1+2]) into the full-length FGFR1c. Wild-type FGFR1c (FGFR1c[WT]) and its mutated variants were stably expressed on the surface of rat skeletal myoblast cells (L6), an αKlotho- and FGFR-deficient cell line. Endoglycosidase H (Endo H) and Peptide:N-glycosidase F (PNGase F) treatment of cell lysates followed by immunoblot analysis with FGFR-specific antibodies showed that mutated FGFR1c variants contain complex sugars, which implies that they reside on the cell surface (Extended Data Fig. 3a). Equal homogeneity/quantity of FGF23[WT] and FGF23[ΔHBS] samples was verified by SDS–PAGE (Supplementary Fig. 2d). Cotreatment with FGF23[WT] and soluble αKlotho of L6-FGFR1c[WT] led to robust FGFR1c activation/signalling as measured by phosphorylation of FGFR1c on A-loop tyrosines, its two direct substrates PLCγ1 (on the regulatory Y783) and FRS2α (on the Grb2 recruitment site Y196) and subsequent activation of a RAS–mitogen-activated protein kinase (MAPK) pathway as monitored by extracellular signal-regulated kinase (ERK) phosphorylation on T202/Y204. In contrast, both FGFR1c[ΔHBS1] and FGFR1c[ΔHBS2] double mutants suffered major losses in their ability to induce FGF23 signalling, and the FGFR1c[ΔHBS1+2] quadruple mutant became totally silent (Fig. 2c). Likewise, FGF23[ΔHBS] was significantly retarded in its ability to activate L6-FGFR1c[WT] in the presence of soluble αKlotho. FGFR activation/signalling data were mirrored by proximity ligation assay (PLA) data. Specifically, cotreatment with FGF23[WT] and αKlotho led to appearance of copious and intense punctate fluorescent signals on the surface of the L6-FGFR1c[WT] cell line. In contrast, there were far fewer fluorescent signals on the surface of FGFR1c[ΔHBS1], FGFR1c[ΔHBS2] and FGFR1c[ΔHBS1+2] cell lines upon cotreatment. Similarly, markedly fewer fluorescent dots were present on the surface of the L6-FGFR1c[WT] cell line when cotreated with FGF23[ΔHBS] and soluble αKlotho (Fig. 2d). These cell-based data confirm the indispensability of FGF23–HS and FGFR1c–HS contacts in inducing formation of a FGF23–FGFR1c–αKlotho–HS quaternary cell-surface signalling complex. Consistent with the 1:2:1:1 FGF23–FGFR–αKlotho–HS stoichiometry in the cryo-EM structures, size-exclusion chromatography–multi-angle light scattering experiments showed that the FGF23–FGFR1c–αKlotho–HS quaternary complex migrates

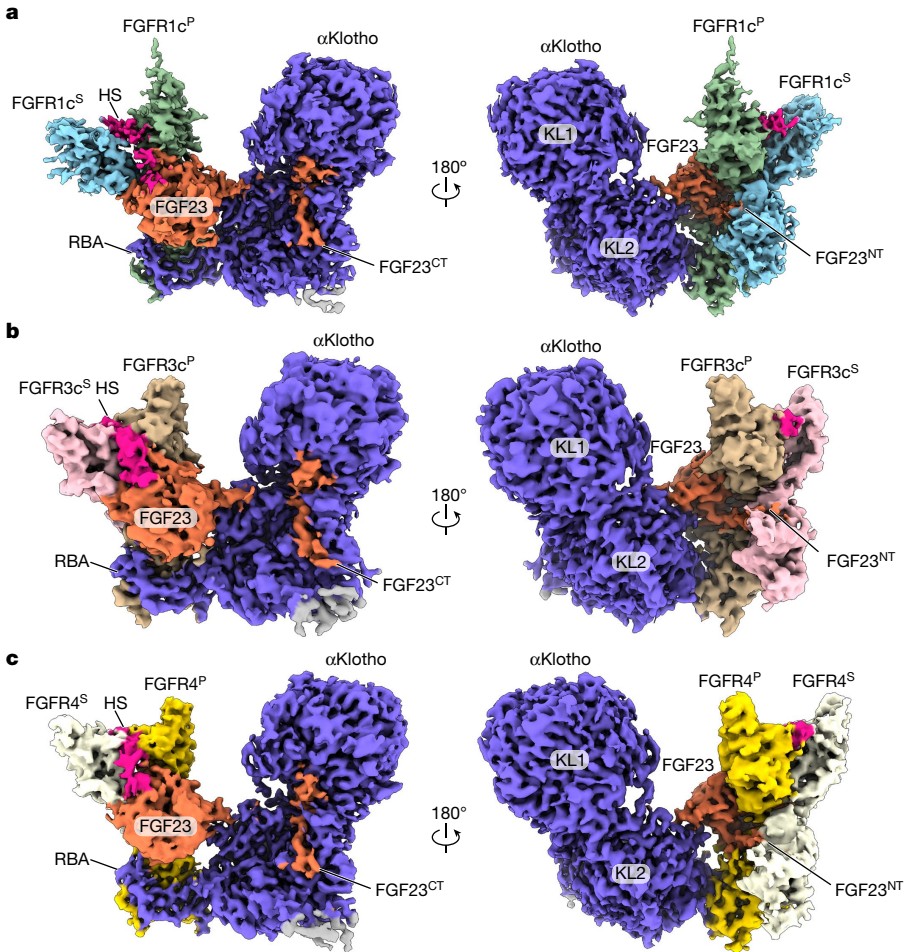

**Fig. 1 | HS promotes formation of 1:2:1:1 asymmetric FGF23–FGFR–αKlotho–HS quaternary complexes.** Overall view of the cryo-EM reconstructions of FGF23–FGFR1c–αKlotho–HS (**a**), FGF23–FGFR3c–αKlotho–HS (**b**) and FGF23–FGFR4–αKlotho–HS (**c**) quaternary complexes displayed at threshold levels of 0.6, 0.45 and 0.5, respectively. The quaternary complex is shown in two different orientations related by a 180° rotation along the vertical axis. FGF23 is coloured in orange, αKlotho is shown in deep blue and HS is in magenta. Primary receptor (FGFR$^P$) and secondary receptor (FGFR$^S$) are shown in green and light blue in the FGF23–FGFR1c–αKlotho–HS complex, in tan and pink in the FGF23–FGFR3c–αKlotho–HS complex and in gold and light yellow in the FGF23–FGFR4–αKlotho–HS complex. Two tandem glycosidase-like domains (KL1 and KL2) and RBA of αKlotho are labelled. The relevance of the extra weak density (in grey) seen in all three 1:2:1:1 quaternary complexes is discussed in Supplementary Fig. 3.

as a single species with a calculated molecular mass of approximately 220 kDa, which matches closely the theoretical value for a 1:2:1:1 FGF23–FGFR1c–αKlotho–HS (215 kDa) (Supplementary Fig. 2e).

## Direct FGFR$^P$–FGFR$^S$ interactions

All three subdomains (that is, D2, D2–D3 linker and D3) of both FGFR$^P$ and FGFR$^S$ chains take part in recruitment of FGFR$^S$ to the ternary complex and hence in promoting receptor dimerization (Fig. 3a). The direct FGFR$^P$–FGFR$^S$ contacts bury a modest solvent-exposed surface area (1542.6 Å$^2$) and are preserved among the three cryo-EM structures. The FGFR$^P$–FGFR$^S$ interface can be split into two sites (that is, sites 1 and 2). At site 1, residues from discontinuous loop regions (that is, βA′–βB and βE–βF loops) at the bottom (C-terminal) corner of D2 of FGFR$^P$ asymmetrically pack against D2, D2–D3 linker and D3 of the FGFR$^S$ (Fig. 3a). At site 2, a contiguous stretch of residues encompassing D2–D3 linker, βA strand, βA–βA′ loop and βB′ strand of D3, engage the corresponding region of the secondary receptor in a pseudo-symmetric fashion (Fig. 3a and Supplementary Table 1). Notably, the FGFR$^P$–FGFR$^S$ interface traps two crucial ligand-binding residues of FGFR$^S$, namely, Arg250 and Ser346, which further deprives FGFR$^S$ from ligand binding (Supplementary Fig. 4b).

We targeted site 2 of the FGFR$^P$–FGFR$^S$ interface in each of the three quaternary complexes by introducing four different point mutations individually into full-length FGFR1c (E249A, R254A, I256A, Y280A), FGFR3c (E247A, R252A, I254A, Y278A) or FGFR4 (E243A, R248A, I250A, Y274A). In addition, we introduced a A170D/A171D/S219K triple mutation into FGFR1c to target site 1 of the FGFR1c$^P$–FGFR1c$^S$ dimer interface (Fig. 3a) as the representative of the three quaternary complexes. Full-length FGFR1c, FGFR3c and FGFR4 variants along with their wild-type counterparts were stably expressed in L6 cells. As above, Endo H sensitivity was used to ensure that the mutations do not affect receptor glycosylation/maturation and cell-surface presentation (Extended Data Fig. 3b–d). Apart from the FGFR1c$^{I256A}$ and its corresponding FGFR4$^{I250A}$ mutant, all mutants showed comparable ratios of Endo H-resistant to total FGFR protein-like wild-type FGFRs. These data imply that only FGFR1c$^{I256A}$ and corresponding FGFR4$^{I250A}$ had reduced cell-surface expression. As in the case of the L6-FGFR1c$^{WT}$ cell line, cotreatment with FGF23 and soluble αKlotho of L6-FGFR3c$^{WT}$ and L6-FGFR4$^{WT}$ cell lines robustly stimulated FGFR signalling as detected by phosphorylation/activation of FGFRs and downstream PLCγ1/FRS2α/ERK signal transducers. In contrast, cells expressing mutated FGFR1c, FGFR3c and FGFR4 had diminished FGFR A-loop tyrosine phosphorylation and reduced PLCγ1, FRS2α and MAPK activation (Fig. 3b–d).

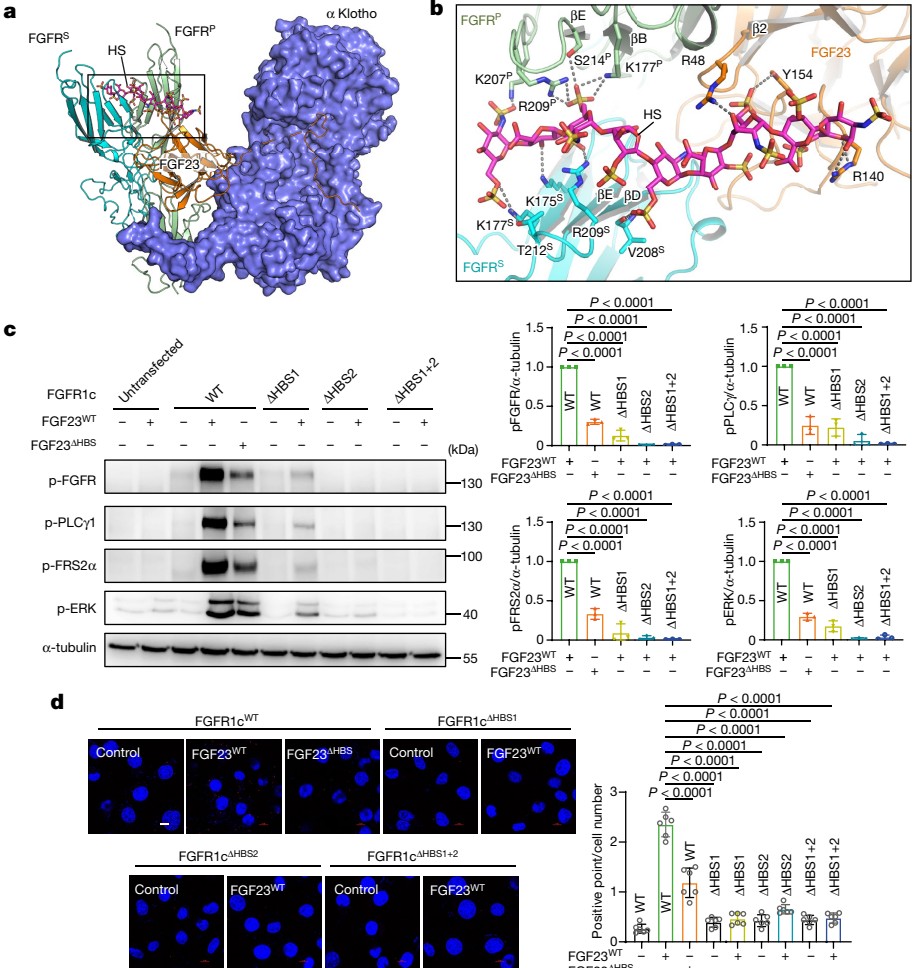

**Fig. 2 | HS promotes asymmetric 1:2:1:1 FGF2–FGFR–αKlotho–HS quaternary complex by simultaneously engaging HBS of FGF23, FGFR1c^P and FGFR1c^S.** **a**, FGF23–FGFR1c–αKlotho–HS asymmetric quaternary complex displayed as a hybrid of cartoon (FGF23, FGFR and HS) and surface (αKlotho) in the same orientation as in Fig. 1a (left). **b**, Expanded view of the boxed region in panel **a** showing tripartite interaction of HS with FGF23, FGFR^P and FGFR^S. HS interacting residues are shown as sticks and labelled. Hydrogen bonds in this figure and subsequent figures are represented as black dashed lines. Throughout the figures, nitrogen, oxygen and sulfur atoms are coloured blue, red and yellow, respectively. All structural illustrations were made using Pymol (v.2.5.2). **c**, L6-FGFR1c^WT, L6-FGFR1c^ΔHBS1, L6-FGFR1c^ΔHBS1 or L6-FGFR1c^ΔHBS1+2 were cotreated with 20 nM 1:1 mixture of FGF23^WT + αKlotho or FGF23^R48A/R140A (FGF23^ΔHBS) + αKlotho (in the case of L6-FGFR1c^WT only) or left untreated. Total cell lysates were immunoblotted with an anti-pY656/Y657-FGFR, anti-pY783-PLCγ1,

anti-pY196-FRS2α, anti-pT202/Y204-ERK and anti-β-tubulin antibody (loading control). **d**, Loss in the abilities of FGF23^ΔHBS and FGFR1c^ΔHBS mutants to induce formation of the FGF23–FGFR1c–αKlotho–HS quaternary complex as visualized by PLA. Red spots signify FGF23–FGFR1c–αKlotho–HS quaternary complexes on the cell surface and blue-stained large circles/ovals are cell nuclei. Note that, compared with FGF23^WT-treated L6-FGFR1c^WT cells, there are far fewer red spots in FGF23^ΔHBS-treated L6-FGFRc^WT and FGF23^WT-treated L6-FGFR1c^ΔHBS cells. Scale bar, 10 μm. Immunoblotting (**c**, normalized against wild-type FGFR1 and FGF23, *n* = 3 biologically independent experiments) and PLA (**d**, *n* = 6 randomly chosen microscope fields from two biologically independent experiments) were quantitated as described in the Methods section and are presented as the mean ± standard deviation. *P* values were determined by two-way ANOVA followed by Tukey's multiple comparisons post hoc test.

It is unlikely that the impaired signalling by FGFR1c^I256A and FGFR4^I250A is solely due to reduced cell-surface expression because the glycosylation and cell-surface expression of the corresponding FGFR3c^I254A is unaffected.

## Secondary FGF23–FGFR^S contacts

FGF23 also participates in recruiting FGFR^S to the ternary complex and hence in buttressing FGFR^P–FGFR^S dimerization. These secondary FGF23–FGFR^S contacts are mediated by both the rigid trefoil core and the flexible N terminus of FGF23. Specifically, the β8–β9 and β10–β12 loops within FGF23's trefoil core engage the βC–βD and βE–βF loops at the bottom edge (C-terminal) of the FGFR^S D2 domain (Fig. 4a). In parallel, hydrophobic residues at the very distal end of the FGF23 N terminus extend from the FGF23–FGFR^P–αKlotho complex and interact

with a hydrophobic groove in the FGFR^S D3 domain that corresponds to the D3 groove in FGFR^P engaged by the RBA of αKlotho (Fig. 4a and Extended Data Fig. 4a). In doing so, the FGF23 N terminus is likely to discourage binding of another αKlotho to FGFR^S and hence contributes to the quaternary complex asymmetry. When compared with the crystal structure of the HS-free 1:1:1 FGF23–FGFR–αKlotho complex, the N terminus of FGF23 displays a major conformational change in the 1:2:1:1 FGF23–FGFR–αKlotho–HS quaternary complex (Extended Data Fig. 4b–d). Notably, in all three cryo-EM structures, electron densities for the FGF23 N terminus are poorly defined, which implies that the FGF23 N terminus engages FGFR^S D3 in a rather degenerate and flexible fashion (Fig. 1a–c and Extended Data Fig. 4e). Interactions of the FGF23 N terminus with FGFR^S D3 were corroborated by a 300 ns all-atom molecular dynamics (MD) simulation of the FGF23–FGFR1c–αKlotho–HS complex (Extended Data Fig. 4f–i). Akin to the FGFR^P–FGFR^S

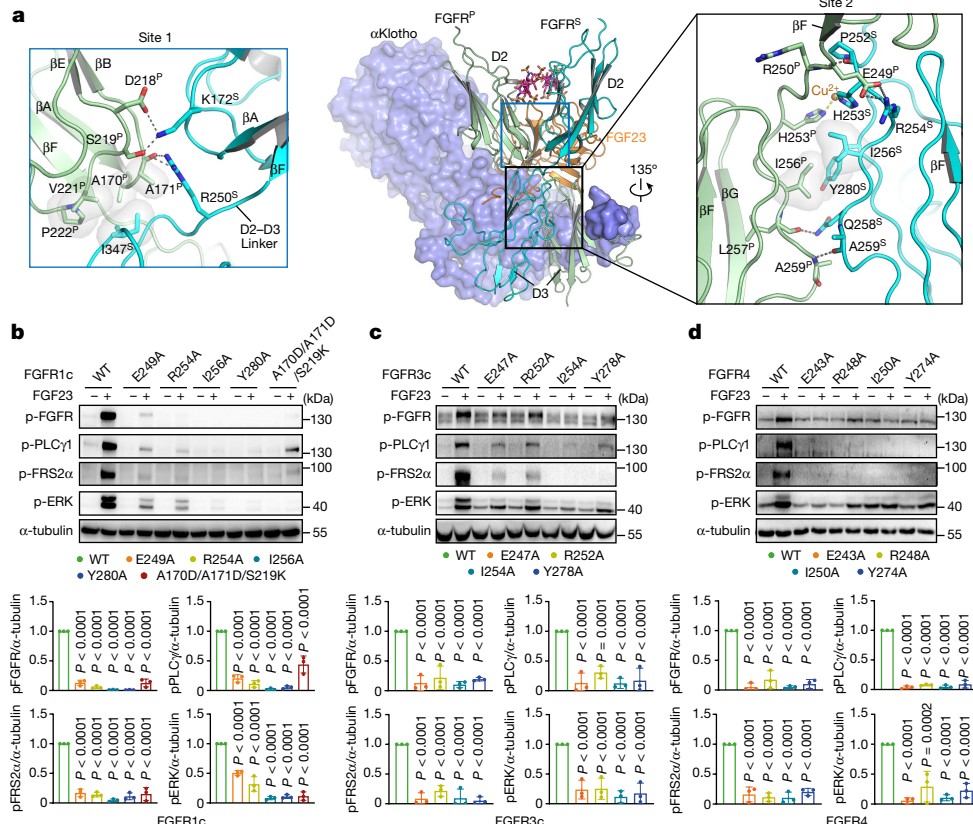

**Fig. 3 | Direct FGFR^P–FGFR^S contacts are required for receptor dimerization and activation. a**, Representation of the FGF23–FGFR1c–αKlotho–HS complex structure as a mix of cartoon (FGF23 and FGFR) and surface (αKlotho). View is related to that in Fig. 1a (right) by 90° rotation along the vertical axis. Contact sites 1 and 2 of the FGFR1c^P–FGFR1c^S dimer interface are boxed in blue and black, respectively. Note that αKlotho does not directly participate in recruiting FGFR1^S. Left, magnified view of site 1 involving D2 domain of FGFR1c^P and D2, D2–D3 linker and D3 of FGFR1c^S. Right, close-up view of site 2 between D3 domains of FGFR1c^P and FGFR1c^S. Side chains of the interacting residues are shown as sticks. Selected secondary structure elements are labelled. Hydrophobic contacts are highlighted as a semitransparent surface. A Cu^{2+} ion (orange sphere) is coordinated by analogous histidine residues from FGFR1c^P

and FGFR1c^S. Cu^{2+} assignment was based on a previous publication implicating specific Cu^{2+} interactions with extracellular domains of FGFRs[39]. Cu^{2+} ions probably derive from cell culture media (DMEM and DME/F12) used to grow HEK293S GnTI^– cells that secrete minimally glycosylated FGFR ectodomains. **b–d**, Immunoblot analyses of whole cell extracts probed as in Fig. 2c from unstimulated or FGF23 (20 nM) and αKlotho (20 nM) costimulated FGFR1c (**b**), FGFR3c (**c**) or FGFR4 (**d**) expressed L6 cell lines. Immunoblotting data were quantitated as described in the Methods section and are presented as the mean ± standard deviation. Data normalized against wild-type FGFR and FGF23, n = 3 biologically independent experiments. P values were determined by two-way ANOVA followed by Tukey's multiple comparisons post hoc test.

interface, the FGF23–FGFR^S interface (Supplementary Table 2) is also conserved among the three quaternary complexes and masks a rather modest surface exposed area (1668.6 Å²). Notably, the FGF23 N terminus and D3 of the FGFR^S engage each other in the proximity of site 2 of the FGFR^P–FGFR^S interface (Fig. 4a). This implies that FGF23–FGFR^S and FGFR^P–FGFR^S contacts act in concert to recruit FGFR^S to the ternary complex and promote receptor dimerization (that is, FGFR^P–FGFR^S).

Next, we studied the impact of interfering with the secondary FGF23–FGFR^S contacts on formation of the 1:2:1:1 FGF23–FGFR–αKlotho–HS quaternary signalling complex. For this purpose, we generated two FGF23 variants: one lacking its first 12 N-terminal residues (that is, Y25 to W36; termed FGF23^{ΔNT}), and the other harbouring a M149A/N150A/P151A triple mutation within the core region of the FGF23 (termed FGF23^{ΔSRBS}). According to the cryo-EM structures, the M149A/N150A/P151A triple mutation and N-terminal truncation are predicted to abrogate secondary interactions of FGF23 with D2 and D3 domains of FGFR^S, respectively, and hence impair FGF23 signalling (Fig. 4a). To test this, an L6 cell line co-expressing FGFR1c^{WT} and transmembrane αKlotho was generated and exposed to increasing concentrations of FGF23^{WT}, FGF23^{ΔNT} or FGF23^{ΔSRBS}. Receptor dimerization/activation efficacies of increasing concentrations of ligands were assessed by immunoblotting for FGFR A-loop tyrosine phosphorylation and by

PLA (Fig. 4b,c). Comparison of dose–response curves showed that, at all concentrations, neither FGF23^{ΔNT} nor FGF23^{ΔSRBS} could reach the maximal activity (E_{max}) exerted by FGF23^{WT} in both assays. Moreover, when combined with FGF23^{WT}, both FGF23^{ΔSRBS} and FGF23^{ΔNT} acted as competitive antagonists producing a net decrease in FGFR1c activation (Fig. 4d). However, relative to FGF23^{ΔNT}, FGF23^{ΔSRBS} showed a greater loss in its dimerization/signalling capacity (Fig. 4b–d), which implies that the FGF23–FGFR^S contacts mediated via the core of the FGF23 contribute more to the overall stability and functionality of the FGF23–FGFR1c–αKlotho–HS quaternary signalling complex than those mediated by FGF23 N terminus.

## Cell-based receptor complementation

We devised a cell-based receptor complementation assay to interrogate the asymmetry of FGF23–FGFR–αKlotho–HS quaternary signalling complexes. Specifically, based on the distinct roles played by primary and secondary receptors in the asymmetric dimerization, we engineered two FGFR1c variants capable of acting exclusively either as the primary or secondary receptor. We reasoned that although these variants would be dysfunctional individually, they should complement each other and form a functional 1:2:1:1 quaternary complex in response

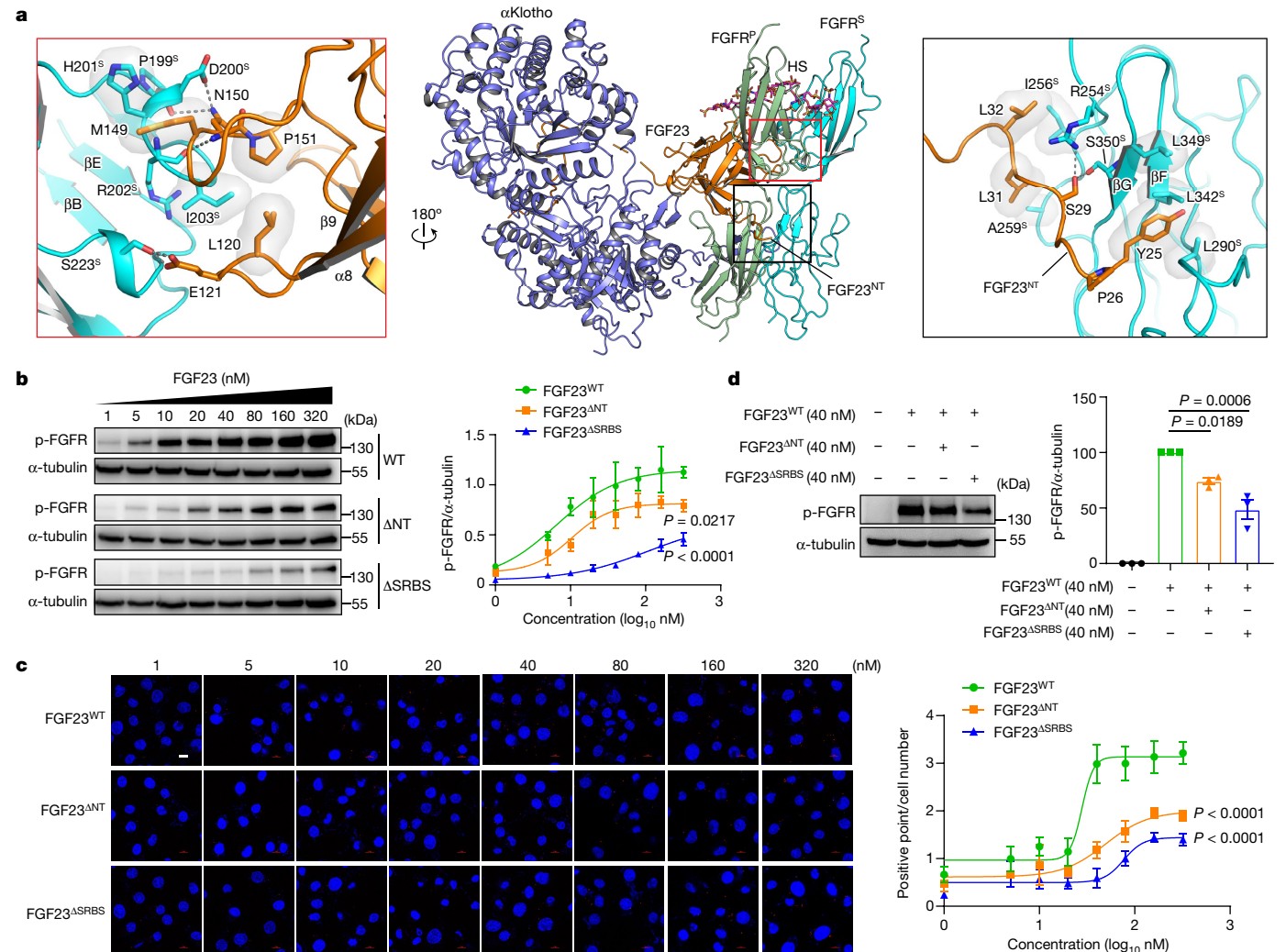

**Fig. 4 | Secondary contacts between FGF23 and FGFR$^S$ are essential for receptor dimerization and activation. a**, Cartoon representation of the FGF23–FGFR–αKlotho–HS structure in the same view as in Fig. 1a (right). Orange and black boxes signify the two contact regions between FGF23 and FGFR$^S$, namely, FGF23$^{core}$:FGFR1c$^S$ D2 domain (left, orange box) and FGF23$^{NT}$: FGFR1c$^S$ D3 domain (right). **b**, L6-FGFR1c$^{WT}$ cells were treated with increasing concentrations of FGF23$^{WT}$, FGF23$^{ΔNT}$ or FGF23$^{ΔSRBS}$ and whole cell lysates were probed as in Fig. 2. Equal homogeneity/quantity of FGF23$^{WT}$, FGF23$^{ΔNT}$ and FGF23$^{ΔSRBS}$ samples was verified by SDS–PAGE (Supplementary Fig. 2d). **c**, Loss in the abilities of FGF23$^{ΔNT}$ and FGF23$^{ΔSRBS}$ mutants to induce FGF23–FGFR–αKlotho–HS quaternary complex formation (that is, receptor dimerization)

relative to FGF23$^{WT}$ as assessed by PLA. Note that there are far fewer red spots in cells stimulated with FGF23 mutants relative to FGF23$^{WT}$. **d**, L6-FGFR1c$^{WT}$ cells were treated with FGF23$^{WT}$ (40 nM) alone, FGF23$^{WT}$ (40 nM) mixed with FGF23$^{ΔSRBS}$ (40 nM) or FGF23$^{ΔNT}$ (40 nM) and whole cell extracts were probed with pFGFR antibodies. Scale bar, 10 μm. Immunoblotting (**b** and **d**, $n$ = 3 biologically independent experiments) and PLA (**c**, $n$ = 6 randomly chosen microscope fields from two biologically independent experiments) were quantitated as described in the Methods section and are presented as the mean ± standard deviation. $P$ values were determined by two-way ANOVA followed by Tukey's multiple comparisons post hoc test.

to FGF23 and αKlotho (Fig. 5a and Extended Data Fig. 5a,b). To generate a FGFR1c variant functioning exclusively as primary receptor, we introduced a I203E/S223E double mutation in its D2 domain. As mentioned above, these two residues in FGFR1c$^S$ engage in hydrophobic and hydrogen bonding contacts with residues in the core of the FGF23 (that is, FGF23–FGFR1c$^S$ interface, Fig. 4a), which are indispensable for recruitment of FGFR1c$^S$ to the ternary complex as implied by the severe loss of function of FGF23$^{ΔSRBS}$ (Fig. 4b,c). However, the corresponding residues in FGFR1c$^P$ are dispensable for quaternary complex formation and are solvent exposed (Extended Data Fig. 5b). Consequently, the resulting mutant FGFR1c (termed FGFR1c$^{ΔSLBS}$) is predicted to lose the ability to act as a secondary receptor but should retain the ability to serve as a primary receptor. To make an FGFR1c variant capable of acting solely as secondary receptor, we introduced a A167D/V248D double mutation into its D2 domain. The A167D/V248D double mutation abrogates

formation of highly conserved hydrogen and hydrophobic contacts between FGF23 and the FGFR1$^P$ D2 domain (Extended Data Fig. 5a,b). Thus, the resulting FGFR1c$^{A167D/V248D}$ mutant (termed FGFR1c$^{ΔPLBS}$) is predicted to lose the ability to function as a primary receptor (that is, form a ternary complex). However, FGFR1c$^{ΔPLBS}$ should retain its ability to function as a secondary receptor because the corresponding residues in FGFR1c$^S$ do not play any role in the quaternary complex formation and are completely solvent exposed.

To perform the complementation assay, an L6 cell line co-expressing FGFR1c$^{ΔSLBS}$ and FGFR1c$^{ΔPLBS}$ (L6-FGFR1c$^{ΔSLBS}$ + FGFR1c$^{ΔPLBS}$) was generated along with two control L6 cell lines individually expressing FGFR1c$^{ΔSLBS}$ (L6-FGFR1c$^{ΔSLBS}$) and FGFR1c$^{ΔPLBS}$ (L6-FGFR1c$^{ΔPLBS}$). These three cell lines and L6-FGFR1c$^{WT}$ (positive control) were exposed to a fixed concentration of FGF23 and soluble αKlotho for increasing time intervals. FGFR1c signalling was assessed by monitoring tyrosine

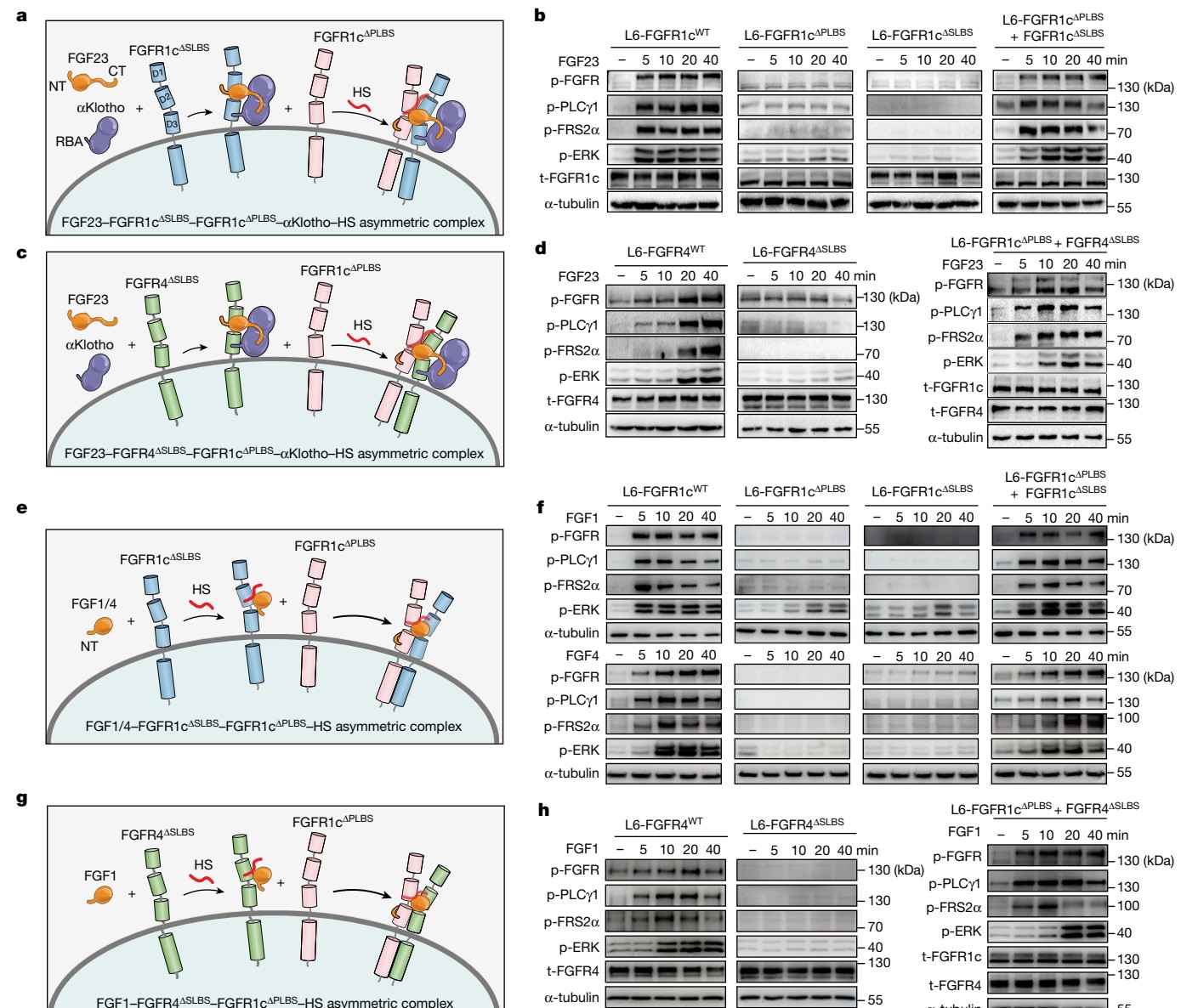

**Fig. 5 | Both endocrine and paracrine FGFs signal via asymmetric receptor dimers. a**, Schematic diagram showing that, in response to FGF23 and αKlotho cotreatment, FGFR1c^ΔSLBS and FGFR1c^ΔPLBS can complement each other and form a 1:1:1:1:1 FGF23–FGFR1c^ΔSLBS–FGFR1c^ΔPLB–αKlotho–HS asymmetric complex. **b**, Immunoblot analyses of whole extracts from untreated or FGF23- and αKlotho-cotreated L6 cell lines singly expressing FGFR1c^WT, FGFR1c^ΔPLBS and FGFR1c^ΔSLBS or co-expressing FGFR1c^ΔSLBS with FGFR1c^ΔPLBS probed as in Fig. 2. **c**, Schematic diagram showing that FGF23, αKlotho and FGFR4^ΔSLBS (serving as a primary receptor) form a ternary complex and recruit FGFR1c^ΔPLBS as a secondary receptor, in the presence of HS. **d**, FGF23- and αKlotho-cotreated L6 cell lines stably expressing FGFR4^WT, FGFR4^ΔSLBS alone or FGFR4^ΔSLBS together with FGFR1c^ΔPLBS were analysed for FGFR activation/signalling using western blotting of total cell lysates as in Fig. 2. **e**, Schematic diagram showing that, in response to paracrine FGF1/4 and HS, FGFR1c^ΔSLBS and FGFR1c^ΔPLBS can complement each other and form 1:1:1:1 FGF–FGFR1c^ΔSLBS–FGFR1c^ΔPLBS–HS asymmetric signalling complexes. **f**, L6 cell lines expressing FGFR1c^ΔSLBS and FGFR1c^ΔPLBS individually or co-expressing them were treated with FGF1 (1 nM) and cell extracts were immunoblotted as in Fig. 2. **g**, Schematic diagram showing that FGF1, HS and FGFR4^ΔSLBS (serving as primary receptor) form a stable complex which subsequently recruits FGFR1c^ΔPLBS as a secondary receptor. **h**, L6 cell lines expressing FGFR4^WT or FGFR4^ΔSLBS singly or co-expressing FGFR4^ΔSLBS with FGFR1c^ΔPLBS were treated with FGF1 (1 nM) and cell extracts were immunoblotted as in Fig. 2. Experiments were performed in biological triplicates with similar results. CT, C terminus; NT, N terminus.

phosphorylation of FGFR, PLCγ1 and FRS2α, and subsequent activation of MAPK pathway. Compared with the L6-FGFR1c^WT cell line, there was negligible FGFR signalling in L6-FGFR1c^ΔPLBS or L6-FGFR1c^ΔSLBS cells (Fig. 5b), which implies that these variants on their own are unable to form signalling-competent quaternary complexes. In contrast, co-expression of FGFR1c^ΔSLBS with FGFR1c^ΔPLBS (Fig. 5b) resulted in robust activation of an FGFR signalling pathway. These data imply that FGFR1c^ΔSLBS can complement FGFR1c^ΔPLBS in forming a signalling-competent FGF23–FGFR1c^ΔSLBS–FGFR1c^ΔPLBS–αKlotho–HS quaternary

complex, in which FGFR1c^ΔSLBS assumes the role of primary receptor while FGFR1c^ΔPLBS is adopted as the secondary receptor (Fig. 5a). Formation of an asymmetric FGF23–FGFR1c^ΔSLBS–FGFR1c^ΔPLBS–αKlotho–HS signalling complex was also validated by PLA (Extended Data Fig. 5c). Compared with L6-FGFR1c^WT cells, FGF23 and αKlotho cotreatment failed to generate any fluorescent signals on the surface of L6-FGFR1c^ΔSLBS and L6-FGFR1c^ΔPLBS cells, which implies that FGFR1c^ΔSLBS and L6-FGFR1c^ΔPLBS variants alone fail to undergo receptor dimerization (that is, quaternary complex formation). In contrast, stimulation

of L6-FGFR1c$^{\Delta SLBS}$ + FGFR1c$^{\Delta PLBS}$ co-expressing cells with FGF23 and αKlotho led to appearance of copious and intense florescent puncta, implying that FGFR1c$^{\Delta SLBS}$ and FGFR1c$^{\Delta PLBS}$ have complemented each other and assembled into an asymmetric FGF23–FGFR1c$^{\Delta SLBS}$–FGFR1c$^{\Delta PLBS}$–αKlotho–HS signalling complex.

## Cell-based receptor heterodimerization

As mentioned above, both FGFR$^P$–FGFR$^S$ and FGF23–FGFR$^S$ interfaces are nearly invariant among the three quaternary complexes. This observation afforded us with another opportunity to interrogate the asymmetry of the complex. Specifically, we reasoned that a primary FGFR1c$^P$ should be able to pair with FGFR3c or FGFR4 as the secondary receptor to yield functional 1:2:1:1 FGF23–FGFR1c–FGFR3c–αKlotho–HS or FGF23–FGFR1c–FGFR4–αKloth–HS heterodimeric quaternary complexes (Fig. 5c). To test this possibility, we carried out a receptor complementation assay using an FGFR4 variant harbouring a double I197E/S217E mutation (termed FGFR4$^{\Delta SLBS}$)—corresponding to FGFR1c$^{\Delta SLBS}$—as the primary receptor and FGFR1c$^{\Delta PLBS}$ as the secondary receptor. An L6 cell line co-expressing FGFR1c$^{\Delta PLBS}$ with FGFR4$^{\Delta SLBS}$ (L6-FGFR1c$^{\Delta PLBS}$ + FGFR4$^{\Delta SLBS}$), along with control cell lines individually expressing FGFR4$^{WT}$ (L6-FGFR4$^{WT}$) and FGFR4$^{\Delta SLBS}$ (L6-FGFR4$^{\Delta SLBS}$) were derived. These cell lines were cotreated with FGF23 plus αKlotho, and quaternary signalling complex formation was examined by immunoblot analysis and PLA. As with FGFR1c$^{\Delta SLBS}$ and FGFR1c$^{\Delta PLBS}$, FGFR4$^{\Delta SLBS}$ also failed to be activated in response to FGF23 plus αKlotho cotreatment. However, robust activation of an FGFR signalling pathway took place in the L6-FGFR1c$^{\Delta PLBS}$ + FGFR4$^{\Delta SLBS}$ co-expressing cell line (Fig. 5d). The cell signalling data were mirrored by the PLA (Extended Data Fig. 5c). Specifically, inconsequential florescent signal was present on the surface of L6-FGFR4$^{\Delta SLBS}$ cell line upon costimulation with FGF23 and αKlotho. In contrast, copious and intense punctate fluorescent spots appeared on the surface of L6-FGFR1c$^{\Delta PLBS}$ + FGFR4$^{\Delta SLBS}$ co-expressing cells, which implies that an FGF23–FGFR1c–FGFR4–αKlotho–HS heterodimeric quaternary complex can form on the surface of live cells. These cell-based receptor complementation and hererodimerization data unequivocally confirm the asymmetry of our structurally deduced 1:2:1:1 FGF23–FGFR1c–αKlotho–HS signal transduction complex.

## Asymmetric FGFR dimerization is general

Because the αKlotho coreceptor does not directly participate in FGFR$^S$ recruitment, we suggested that asymmetric mode of receptor dimerization revealed by our FGF23–FGFR–αKlotho–HS cryo-EM structures may also be relevant to paracrine FGFs. To test this conjecture, we studied the impacts of asymmetric FGFR$^P$–FGFR$^S$ dimer interface mutations on the abilities of FGFR1c and FGFR2b to mediate paracrine FGF signalling. These two FGFR isoforms were chosen because of their overlapping and unique ligand-binding specificity/promiscuity profile (Extended Data Fig. 5d). FGFR1c responds to paracrine FGF1 (a pan-FGFR ligand) and FGF4, whereas FGFR2b mediates the actions of FGF1, FGF3, FGF7, FGF10 and FGF22. For FGFR2b studies, we generated L6 cell lines expressing either wild-type (FGFR2b$^{WT}$) or FGFR2b mutants (that is, FGFR2b$^{E250A}$, FGFR2b$^{R255A}$, FGFR2b$^{I257A}$ and FGFR2b$^{Y281A}$) analogous to the FGFR1c mutants mentioned above. Both FGFR1c and FGFR2b mutants were impaired in their capacity to undergo ligand-induced tyrosine trans auto-phosphorylation, which was also reciprocated in reduced PLCγ1 and FRS2α phosphorylation (Extended Data Fig. 6).

Next, we carried out receptor complementation and heterodimerization assays as was done for FGF23. For FGFR2b complementation study, we established L6-FGFR2b$^{\Delta SLBS}$, L6-FGFR2b$^{\Delta PLBS}$ and L6-FGFR2b$^{\Delta SLBS}$ + FGFR2b$^{\Delta PLBS}$ cell lines corresponding to FGFR1c cell lines used for FGF23 study. In response to FGF1 or FGF4 stimulation, cells expressing FGFR1c$^{\Delta PLBS}$ or FGFR1c$^{\Delta SLBS}$ alone failed to elicit any appreciable FGFR1c signalling whereas L6-FGFR1c$^{\Delta SLBS}$ + FGFR1c$^{\Delta PLBS}$ cells responded with robust FGFR1c activation and signalling (Fig. 5e,f). Likewise, FGF1, FGF3, FGF7 and FGF10 each induced FGFR activation and signalling only in L6-FGFR2b$^{\Delta SLBS}$ + FGFR2b$^{\Delta PLBS}$ co-expressors (Extended Data Fig. 7). Lastly, we studied the possibility of FGFR heterodimerization by paracrine FGFs by focusing on: (1) FGFR1c–FGFR4 and FGFR1b–FGFR2b heterodimerizations by FGF1; (2) FGFR1b–FGFR2b heterodimerization by FGF10; and (3) FGFR2b–FGFR3b heterodimerization by FGF3. For the FGFR1b–FGFR2b heterodimerization assay, we generated an L6 cell line expressing FGFR1b$^{\Delta SLBS}$, equivalent to FGFR2b$^{\Delta SLBS}$. For FGFR2b–FGFR3b heterodimerization by FGF3, we used wild-type FGFR3b (FGFR3b$^{WT}$) as the ΔPLBS equivalent because this isoform naturally does not respond to FGF3. FGF1 failed to activate FGFR signalling in FGFR1c$^{\Delta PLBS}$ and FGFR4$^{\Delta SLBS}$ cell lines. However, strong FGFR signalling was seen in the FGFR1c$^{\Delta PLBS}$ + FGFR4$^{\Delta SLBS}$ co-expressing cell line in response to FGF1 stimulation (Fig. 5g,h). Likewise, both FGF1 and FGF10 induced robust FGFR activation/signalling only in the FGFR1b$^{\Delta SLBS}$ + FGFR2b$^{\Delta PLBS}$ co-expressing cell line but not in the L6-FGFR1b$^{\Delta SLBS}$ and L6-FGFR2b$^{\Delta PLBS}$ cell lines (Extended Data Fig. 8a–c). Lastly, FGF3 provoked signalling in the FGFR2b$^{\Delta SLBS}$ + FGFR3b$^{WT}$ co-expressing cell line but not in the L6-FGFR2b$^{\Delta SLBS}$ and L6-FGFR3b$^{WT}$ cells (Extended Data Fig. 8d,e). Based on these extensive cell-based data we conclude that paracrine FGF signalling is mediated via HS-induced asymmetric 1:2 FGF–FGFR dimers, reminiscent of the HS- and Klotho-induced endocrine 1:2 FGF23–FGFR dimer (Extended Data Fig. 9).

## Concluding remarks

The 1:2:1:1 FGF23–FGFR–αKlotho–HS asymmetric quaternary complex structures and supporting biochemical data presented in this manuscript reveal how Klotho and HS glycosaminoglycan coreceptors act in concert to promote asymmetric 1:2 endocrine FGF–FGFR dimerization necessary for receptor activation and hence FGF hormone signalling. In this model, Klotho coreceptors tether FGF hormone and its primary receptor together and thus generate a stable endocrine FGF–FGFR$^P$ complex in the context of 1:1:1 FGF–FGFR$^P$–Klotho ternary complexes. In doing so, Klotho coreceptors effectively offset the HS incompetency in stabilizing the endocrine FGF–FGFR$^P$ complex (Extended Data Fig. 9). The stabilized endocrine FGF–FGFR$^P$ complex is then assisted by a HS coreceptor to recruit a secondary FGFR$^S$. Importantly, the Klotho coreceptor dependency restricts the site of action of FGF hormones to Klotho expressing tissues/organs (that is, kidney and parathyroid glands in the case of FGF23). Consistent with a lack of a direct role of Klotho in FGFR$^S$ recruitment, we show that the asymmetric mode of receptor dimerization is also applicable to paracrine FGFs. Unlike FGF hormones, paracrine FGFs have substantial affinity for HS and hence can rely solely on HS as an obligatory coreceptor to both stably bind FGFR$^P$ and recruit FGFR$^S$. Despite sharing structural similarity, Klotho and HS-induced endocrine 1:2 FGF–FGFR dimers are thermodynamically inferior to HS-induced paracrine 1:2 FGF–FGFR dimers. Specifically, owing to the weak HS binding affinity of the FGF hormone, FGFR$^S$ is bound less tightly in the endocrine 1:2 FGF–FGFR dimer, which accounts for the weaker receptor activation/signalling capacity of FGF hormones relative to paracrine FGFs as posited by our 'threshold model' for FGF signalling specificity[38]. Remarkably, the asymmetric receptor dimerization model is compatible with receptor heterodimerization, which is likely to serve as an additional mechanism to qualitatively and quantitatively fine-tune FGF signalling.

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

# Methods

## Expression constructs

The ligation-independent In-Fusion HD cloning kit (no. 639648, Clontech) was used to construct neomycin and hygromycin resistant pEF1α-IRES-neo and pEF1α-IRES-hygro lentiviral vectors encoding wild-type full-length human FGFR2c, FGFR3c and FGFR4 according to the protocol previously described for the human FGFR1c lentiviral expression construct[6]. The pET-30a-based bacterial expression construct for N-terminally his-tagged human FGF23 (residues Tyr25 to Ile251) was described previously[6]. pHLsec expression vectors encoding the extracellular ligand-binding region (that is, D2–D3 region) of FGFR3c (residues Asp142 to Arg365; FGFR3c^ecto) and FGFR4 (residues Asp142 to Arg365; FGFR4^ecto) were made following the same strategy described previously for the human FGFR1c^ecto (ref. 6). Single and multiple site mutations and truncations were introduced into expression constructs encoding the wild-type proteins using a Q5 site-directed mutagenesis kit (no. E0554S, New England Biolabs Inc.). The expression constructs were verified by restriction enzyme digestion and DNA sequencing.

## Recombinant protein expression and purification

To produce minimally glycosylated ectodomains of FGFR1c, 3c and 4, N-acetylglucosaminyltransferase I (GnTI⁻) deficient HEK293S cells were transiently transfected with respective pHLsec expression constructs via cationic polymer polyethyleneimine (PEI, no. 23966-1, Polysciences, Inc.) following a published protocol[40]. Then, 24 h posttransfection, tryptone N1 (TN1, catalogue no. 19553, Organotechnie) was added to the medium to promote protein expression. At day three, secreted FGFR ectodomains from 1 l of conditioned medium were captured on a heparin affinity HiTrap column (no. 17040703, GE Healthcare) and eluted with 20 column volume of salt gradient (0–2 M). Fractions containing FGFR^ecto proteins were pooled, concentrated to 5 ml and applied to a Superdex-75 gel filtration column (no. 28989333, GE Healthcare). FGFR^ecto proteins were eluted isocratically in 25 mM HEPES pH = 7.5 buffer containing 1 M NaCl. Wild-type and mutated FGF23 proteins were expressed in *E. coli* as inclusion bodies, refolded in vitro and purified to homogeneity using sequential cation exchange and SEC following an established protocol[6]. Secreted αKlotho^ecto was purified from conditioned media of a HEK293S GnTI⁻ cell line stably expressing the entire extracellular domain of human αKlotho (residues Met1 to Ser981; αKlotho^ecto) using a heparin affinity HiTrap column followed by SOUCRE Q anion and Superdex 200 column chromatography, as described previously[6].

## Cryo-EM specimen preparation, image processing, model building and refinement

FGF23–FGFR1c–αKlotho–HS, FGF23–FGFR3c–αKlotho–HS or FGF23–FGFR4–αKlotho–HS quaternary complexes were prepared by mixing FGF23 with one of the FGFR ectodomains, αKlotho^ecto and a heparin dodecasaccharide 12 (HO12, Iduron Ltd) using a 1:1:1:1 molar ratio. The mixtures were concentrated to approximately 5 mg ml⁻¹, applied to a Superdex 200 column (no. 28989335, GE Healthcare) and eluted isocratically in 25 mM HEPES pH = 7.5 buffer containing 100 mM NaCl. Peak fractions were analysed by SDS–PAGE and top fractions containing the highest concentration and purity of the quaternary complex were used directly for grid preparation without further concentration to avoid protein aggregation. The final concentrations of FGF23–FGFR1c–αKlotho–HS, FGF23–FGFR3c–αKlotho–HS and FGF23–FGFR4–αKlotho–HS complexes for grid preparation were 1.5, 2.4 and 1.5 mg ml⁻¹, respectively.

To prepare the cryo-EM grids, 2–3 µl of purified protein complex at approximately 1.5–2.5 mg ml⁻¹ was applied to glow discharged gold grid (UltrAuFoil). The grid was then blotted for 1–2 s under 0 or 1 force at 100% humidity using a Mark IV Vitrobot (FEI) before plunging into liquid ethane. Micrographs of the FGF23–FGFR1c–αKlotho–HS and FGF23–FGFR4–αKlotho–HS complexes were acquired on a Talos Arctica microscope with K2 direct electron detector at ×36,000 magnification (corresponding to 1.096 Å per pixel). Accumulated doses used were 50.37 e⁻/Å² and 53.84 e⁻/Å² for FGF23–FGFR1c–αKlotho–HS and FGF23–FGFR4–αKlotho–HS, respectively. Micrographs of the FGF23–FGFR3c–αKlotho–HS complex were collected on a Titan Krios microscope equipped with a K2 direct electron detector and an energy filter. The magnification used was ×130,000, with a pixel size of 1.048 Å, and an accumulated dose of 72.44 e⁻/Å². Leginon[41] was used to target the holes with 5–100 nm of ice thickness, resulting in 10,186, 6,409 and 16,602 micrographs being collected for each of three quaternary complexes.

WARP[42] was used for motion correction and contrast transfer function estimation for all three cryo-EM datasets. Micrographs with an overall resolution worse than 5.5 Å were excluded. The final number of micrographs used were 9,501, 5,164 and 15,049, respectively, yielding more than one million particles for each complex. Particle stacks were then imported to cryoSPARC[43] for two-dimensional classification, ab-initio reconstruction with three or four models and three-dimensional classification. Finally, 1,497,967 (FGF23–FGFR1c–αKlotho–HS), 291,540 (FGF23–FGFR3c–αKlotho–HS) and 856,877 (FGF23–FGFR4–αKlotho–HS) particles were used for heterogeneous refinement with C1 symmetry, resulting in 2.74, 3.20 and 3.03 Å resolutions, respectively. Components/domains from FGF23–FGFR1c–αKlotho X-ray structures (PDB: 5W21) were manually docked into cryo-EM density maps using Chimera[44] and the rigid body was refined. Initial models were then adjusted in Coot[45] and real-space refined in Phenix[46]. Refinement and model statistics are shown in Extended Data Table 1. Representative cryo-EM images, two-dimensional class averages and three-dimensional maps of each quaternary complex are shown in Extended Data Fig. 1.

## MD simulation

The cryo-EM structure of the FGF23–FGFR1c–αKlotho–HS quaternary complex was solvated in a water box of 16 × 16 × 16 nm³. The CHARMM-GUI server was used to generate the configuration, topology[47–50] and the parameter files with the CHARMM36m force field[51]. In addition to protein molecules, the simulation system included about 138,073 water molecules, 393 sodium and 393 chloride ions (mimicking the 150 mM NaCl present in the protein buffer), resulting in a total of 439,298 atoms. A 300 ns all-atom MD simulation trajectory was generated using GROMACS 2021 (ref. 52) at 303 K using a time step of 2 fs. The cubic periodic boundary condition was used during the simulations and the van der Waals interaction was switched off from 1 nm to 1.2 nm. The long-range electrostatic interactions were calculated using the particle mesh Ewald (PME) method. Energy minimization was carried out using the steepest descent algorithm, followed by a 0.4 ns constant particle number, volume and temperature (NVT) and a 20 ns constant particle number, pressure and temperature (NPT) equilibration simulation by gradually decreasing force restraints from 1000 kJ mol⁻¹ nm⁻² to 400 kJ mol⁻¹ nm⁻² (for the NVT stage) and 400 kJ mol⁻¹ nm⁻² to 40 kJ mol⁻¹ nm⁻² (for NPT stage). At the conclusion of the equilibration steps, all force restraints were removed and the MD simulation was performed in the NPT ensemble.

## Generation of cell lines and FGFR signalling assay

HEK293T cells (verified by a morphology check under microscope, mycoplasma negative in 4′,6-diamidino-2-phenylindole (DAPI)) were used for lentiviral vector packaging and production of high-titre viral particles. An L6 myoblast cell line (no. GNR 4, National Collection of Authenticated Cell Cultures) was used as the host for stable expression of full-length (transmembrane) human FGFR1c, FGFR2c, FGFR3c, FGFR4 and mutants thereof. L6 cells endogenously express HSPGs but are devoid of FGFRs and αKlotho coreceptor and hence are

naturally non-responsive to FGF23. However, via controlled ectopic co-expression of cognate FGFRs and αKlotho or exogenous supplementation with soluble αKlotho[ecto], these cells can respond to FGF23 stimulation. Accordingly, L6 cells are excellent hosts for reconstitution studies of FGF23 signalling in a physiological environment. Both HEK293T and L6 cells were maintained in Dulbecco's Modified Eagle Medium (DMEM, no. C11995500BT, Gibco) supplemented with 10% fetal bovine serum (FBS, no. FSD500, ExCell Bio), 100 U ml$^{-1}$ of Penicillin and 100 µg ml$^{-1}$ Streptomycin (no. P1400, Solarbio). Viral packaging and generation of the recombinant lentivirus particles in HEK293T cells were carried out using published protocols[33]. For stable expression of individual wild-type or mutated FGFR cell lines, $2 \times 10^5$ L6 cells were plated in six-well cell culture dishes and infected with lentivirus particles encoding given FGFR in the presence of polybrene (5 µg ml$^{-1}$; no. sc-134220, Santa Cruz Biotechnology). Stable transfectants were selected using G418 (0.5 mg ml$^{-1}$, no. HY-17561, MedChemExpress) or hygromycin (8 µg ml$^{-1}$, no. HY-B0490, MedChemExpress). For the FGFR1c+αKlotho[TM] co-expressing cell line, L6 cells stably expressing FGFR1c[WT] (resistant to G418) were infected with lentiviral particles encoding wild-type or mutated transmembrane αKlotho (αKlotho[TM]) and the co-expressing cells were selected using hygromycin (80 µg ml$^{-1}$, no. HY-B0490, MedChemExpress).

For cell stimulation studies, parental and stably transfected L6 cells were seeded in 12-well cell culture plates at a density of $1 \times 10^5$ cells per well and maintained for 24 h. On the next day, the cells were rinsed three times with phosphate buffered saline (PBS) and then serum starved for 12 h and costimulated with FGF23[WT] and αKlotho[ecto] for 5, 10, 20 and 40 min. For single time point stimulation, samples were harvested after 5 min.

After stimulation, cells were lysed and total lysates samples were analysed by western blotting, as previously described[33]. The following antibodies were used: phosphorylated FGFR (1:1000, no. 3471S, Cell Signaling Technology), phosphorylated FRS2α (1:1000, no. 3864S, Cell Signaling Technology), phosphorylated PLCγ1 (1:1000, no. 2821S, Cell Signaling Technology), phosphorylated ERK1/2 (1:1000, no. 4370S, Cell Signaling Technology), α-tubulin (1:20000, no. 66031-1-Ig, Proteintech), total-FGFR1 (1:1000, no. 9740S, Cell Signaling Technology), total-FGFR2 (1:1000, no. 23328S, Cell Signaling Technology), total-FGFR3 (1:1000, no. ab133644, Abcam), total-FGFR4 (1:1000, no. 8562S, Cell Signaling Technology), HRP conjugated goat anti-mouse IgG (H + L) (1:5000, no. SA00001-1, Proteintech), HRP conjugated goat anti-rabbit IgG(H + L) (1:5000, no. SA00001-2, Proteintech). Blots were developed using enhanced chemiluminescence reagents (no. P10300, NCM Biotech Laboratories) by the ChemiDoc XRS+ system (Bio-Rad) or Amersham ImageQuant 800 (GE).

### Determination of cell-surface expression of mutated FGFRs via endoglycosidase H sensitivity assay

Endoglycosidase H (Endo H) sensitivity was used to analyse potential impacts of ectodomain mutations on FGFR glycosylation/maturation and hence trafficking to the cell surface. In immunoblots, wild-type FGFRs migrate as a doublet of a major diffuse upper and a minor sharp lower band. The upper band represents the fully glycosylated mature FGFR, decorated with complex sugars that has passed the ER quality control and has been successfully trafficked to the cell surface. On the other hand, the faster migrating lower band is an incompletely processed high mannose form that is trapped in ER. Mutations affecting receptor maturation manifest in an increase in proportion of the faster migrating ER-resident band. The mannose-rich form is sensitive to Endo H which cleaves the bond between two N-acetylglucosamine (GlcNAc) subunits directly proximal to the asparagine residue. However, the fully glycosylated cell surface-resident band is resistant to Endo H. Accordingly, cell-surface abundance of FGFRs can be expressed as a ratio of Endo H-resistant fraction over the total receptor expression as determined by treating the receptor with PNGase

F. This enzyme is an amidase that hydrolyzes the bond between the innermost GlcNAc and asparagine irrespective of complex sugar content and thus completely strips the FGFR from all its N-linked sugars. Accordingly, WT or mutant FGFR cell lines were lysed in an NP-40 lysis buffer (Biotime, no. P0013F, supplemented with 1 mM PMSF) for 15 min at 4 °C. First, 20 µg total protein (quantified by BCA assay) were denatured with glycoprotein denaturing buffer at 100 °C for 10 min and then treated with 500 units of Endo H (New England Biolabs, no. P0702S) or peptide-N-glycosidase F (PNGase F) (New England Biolabs, no. P0704S) following the manufacturer's instructions. Endo H- and PNGase F-treated samples were immunoblotted with FGFR isoform-specific antibodies, as detailed above.

### Size-exclusion chromatography–multi-angle dynamic light scattering

The molecular mass of the SEC-purified FGF23–FGFR1c–αKlotho–HS quaternary complex was determined by multi-angle light scattering following the established protocol[6]. Before the experiment, at least 60 ml of degassed running buffer (25 mM HEPES pH 7.5 containing 150 mM NaCl) were passed through the system to equilibrate the column and establish steady baselines for light scattering and refractive index detectors. Then, 50 µl of purified FGF23–FGFR1c–αKlotho–HS quaternary complex (1.5 mg ml$^{-1}$) was injected onto the Superdex 200 10/300 GL column and the eluent was continuously monitored at 280 nm absorbance, laser light scattering and refractive index at a flow rate of 0.5 ml min$^{-1}$. As a control, 50 µl of a purified FGF23–FGFR–αKlotho ternary complex (1.5 mg ml$^{-1}$) sample was analysed under the same condition. The experiments were performed at ambient temperature. Laser light scattering intensity and eluent refractive index values were used to derive molecular mass as implemented by the ASTRA software (Wyatt Technology Corp.).

### Proximity ligation assay

Cells were seeded onto microscope cover glasses (no. WHB-12-CS-LC, WHB) placed inside 12-well cell culture dishes at $1 \times 10^5$ cells per well and allowed to adhere for 24 h. On the next day, cells were washed three times with PBS and serum starved for 12 h. Following costimulation with FGF23[WT] and αKlotho[ecto] for 20 min, cells were washed three times with PBS and fixed with 4% paraformaldehyde for 30 min prior to PLA. The PLA reaction was performed using a Duolink PLA kit following the manufacturer's instructions (no. DUO92101, Sigma-Aldrich) and visualized via fluorescence microscopy. Briefly, slides were treated with blocking solution for 60 min at 37 °C, rinsed three times with wash buffer A and then incubated overnight at 4 °C with two different primary antibodies raised in two different species for each FGFR isoform of interest (FGFR1 (1:20, no. PA5-25979, ThermoFisher) from rabbit, FGFR1 (1:100, no. ab824, Abcam) from mouse, FGFR2 (1:100, no. 23328S, Cell Signaling Technology) from rabbit, FGFR2 (1:50, no. sc-6930, Santa Cruz Biotechnology) from mouse, FGFR3 (1:100, no. MA5-32620, ThermoFisher) from rabbit, FGFR3 (1:100, no. sc-13121, Santa Cruz Biotechnology) from mouse, FGFR4 (1:100, no. 8562S, Cell Signaling Technology) from rabbit, FGFR4 (1:100, no. sc-136988, Santa Cruz Biotechnology) from mouse). Following three rinses with wash buffer A, slides were incubated with oligo-linked secondary antibodies (Duolink anti-mouse minus and anti-rabbit plus) for 1 h at 37 °C. Slides were rinsed again with wash buffer A and immersed in ligase solution for 30 min at 37 °C, to allow formation of circular DNA, followed by incubation with polymerase solution for 100 min at 37 °C for rolling circle amplification in a dark room. Slides were rinsed twice with wash buffer B for 10 min each, followed by rinsing with a 100-fold dilution of buffer B for 1 min. For imaging analysis, slides were covered by coverslips using a minimal volume of Duolink PLA mounting medium containing DAPI. Slides were examined using a confocal laser scanning microscope (C2si, Nikon).

## Statistical analysis and reproducibility

All statistical analyses were carried out using GraphPad Prism 8.0. For statistical analysis of immunoblotting data, densitometric values (determined using ImageJ) from three independent experiments were used. For statistical analysis of PLA data, a number of fluorescent dots and cells (counted manually) was used from six randomly chosen microscope fields from two biologically independent experiments. Processing of the western blotting and PLA data in Figs. 2c,d, 3b–d and 4b,c was done using two-way ANOVA, followed by Tukey. One-way ANOVA, followed by Tukey was applied to process the western blotting data in Extended Data Fig. 3. Protein purifications were repeated at least eight times yielding samples with comparable purity/quantity. Western blotting experiments were carried out in biological triplicates with similar results. PLA assays were repeated at least twice independently, with analogous results.

## Reporting summary

Further information on research design is available in the Nature Portfolio Reporting Summary linked to this article.

## Data availability

Electron density maps and refined models for the FGF23–FGFR1c–αKlotho–HS (EMD-34075, PDB: 7YSH), FGF23–FGFR3c–αKlotho–HS (EMD-34082, PDB: 7YSU) and FGF23–FGFR4–αKlotho–HS (EMD-34084, PDB: 7YSW) quaternary complexes have been deposited in the Electron Microscopy Data Bank and will be released upon acceptance of the manuscript. Raw uncropped western blot images are compiled in Supplementary Fig. 1. Source data are provided with this paper.

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

**Acknowledgements** Financial support was provided by the National Key R&D Program of China (2017YFA0506000 to X.L.), National Natural Science Foundation of China (82073705 & 82273842 to G.C., 82103999 to L.C., 81930108 to G.L.), Oujiang laboratory startup fund (OJQD2022007 to M.M.), Kungpeng Action Plan award (to M.M.), Natural Science Funding of Zhejiang Province (LR22H300002 to G.C., LQ22H300007 to L.C.), Wenzhou Major Scientific and Technological Innovation Project (ZY2021023 to G.C.), Qianjiang Talent Plan of Zhejiang (QJD1902016 to G.C.) and Wenzhou Institute (UCAS) startup fund (WIUCASQD2021043, to Y.W.).

**Author contributions** L.C. expressed, purified and prepared the quaternary complexes, carried out the size-exclusion chromatography–multi-angle light scattering analysis and prepared the structural figures. L.C. and J.S. prepared the cryo-EM grids, and collected and processed the images. A.Z. expressed and purified FGFR ectodomains. L.F. generated the mammalian expression constructs. J.L. expressed the mutant FGF23 proteins. L.F., Z.H., M.F., X.L., Z.P. and G.C. generated western blotting and PLA data and prepared the related figures. Y.W. generated MD simulation data. M.M. conceived the project, built and analysed the structural models and wrote the manuscript. All authors participated in discussion and revision of the manuscript.

**Competing interests** The authors declare no competing interests.

**Additional information**
**Correspondence and requests for materials** should be addressed to Xiaokun Li, Gaozhi Chen or Moosa Mohammadi.

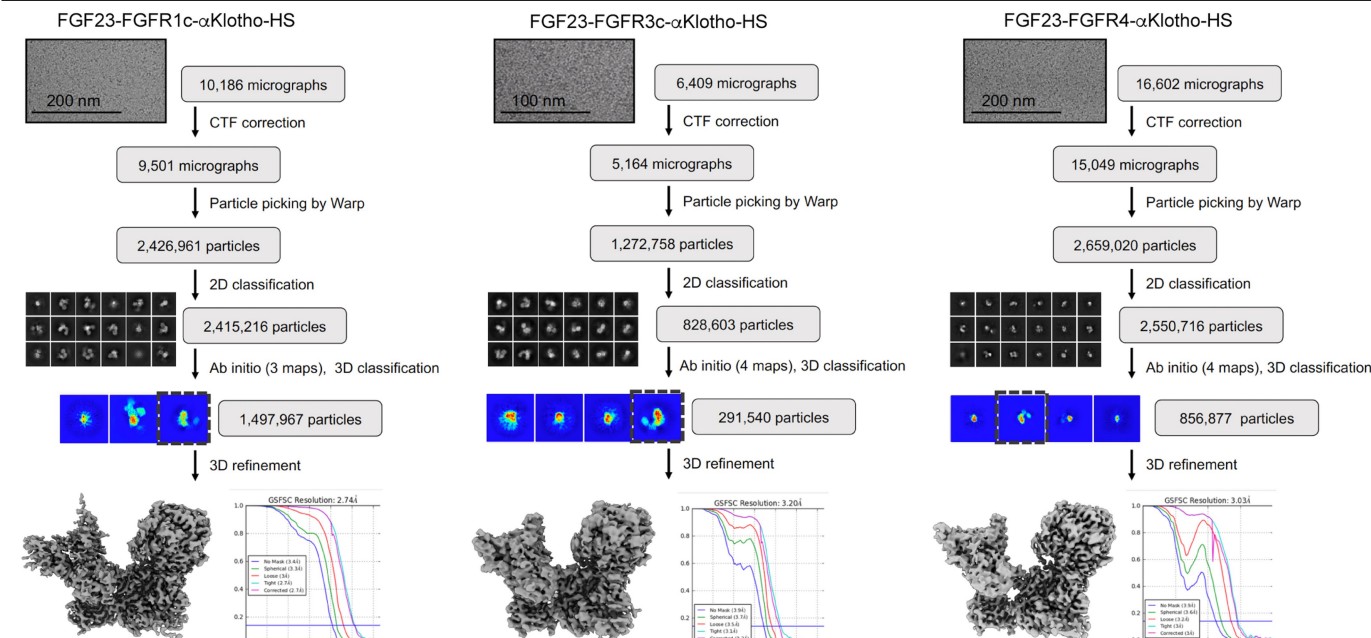

**Extended Data Fig. 1 | Determination of molecular structures of FGF23-FGFR-αKlotho-HS quaternary complexes by Cryo-EM.** Image processing workflow for the FGF23-FGFR-αKlotho-HS quaternary complex containing FGFR1c (left), FGFR3c (middle) or FGFR4 (right) as the receptor component.

Gold-standard Fourier shell correlation (GSFSC) show global resolutions of 2.74, 3.20, and 3.03 Å for FGF23-FGFR1c-αKlotho-HS, FGF23-FGFR3c-αKlotho-HS, and FGF23-FGFR4-αKlotho-HS quaternary complexes, respectively.

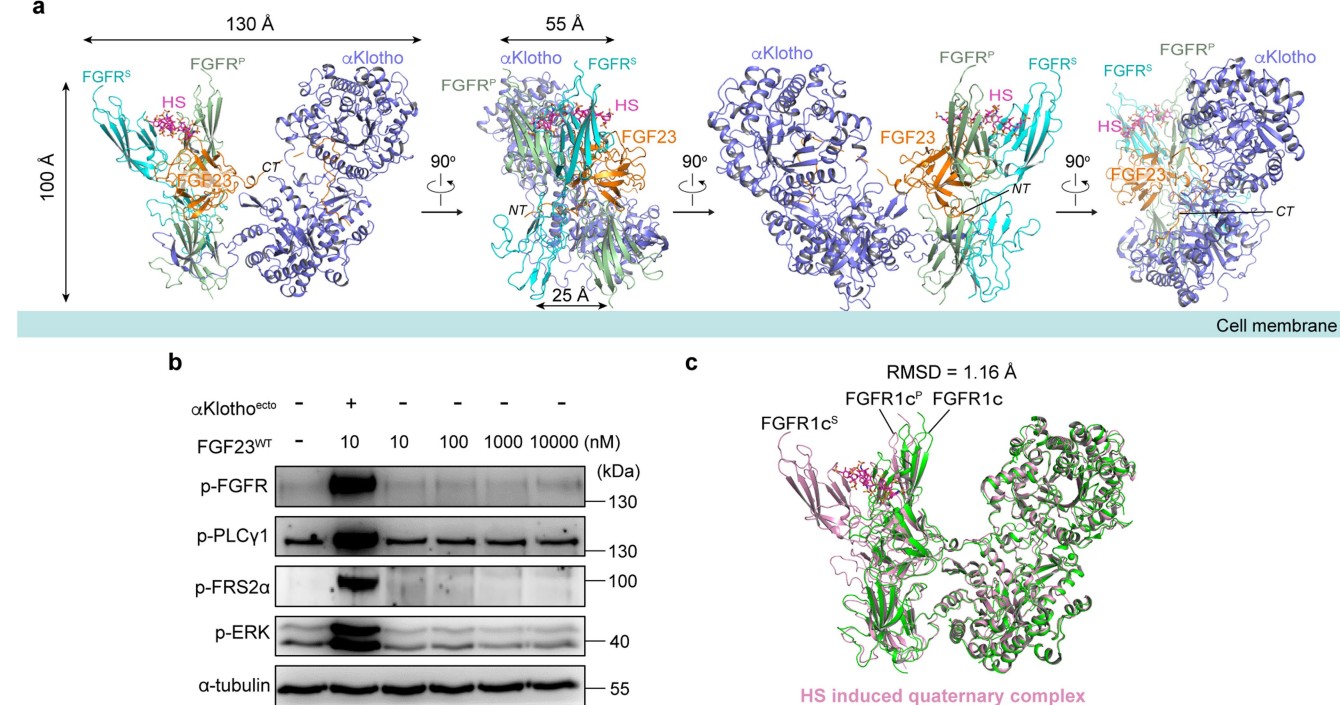

**Extended Data Fig. 2 | Overall topology of the FGF23 signaling complex and its strict αKlotho dependency. A**, Cartoon representation of the cryo-EM structure of the 1:2:1:1 FGF23-FGFR-αKlotho-HS quaternary complex in four different orientations related by 90° rotation along the vertical axis. The asymmetric quaternary complex has an average dimension of 130 Å × 100 Å × 55 Å. The proximity of membrane insertion points of FGFR chains (-25 Å apart) would be conducive to the formation of an asymmetric A-loop trans-phosphorylating dimer of intracellular kinase domain[27]. FGF23, αKlotho and HS are shown in orange, blue and magenta, respectively. Primary receptor (FGFR[P]) and secondary receptor (FGFR[S]) are shown in pale green and cyan. Dashed lines denote residues 172–182 of FGF23, the linker between FGF23's trefoil core and its distal αKlotho binding site, which could not be built due to lack of interpretable electron density. This region does not interact with either αKlotho or FGFR and is likely disordered/flexible. Notably, this region harbors the regulatory subtilisin-like proprotein convertase (SPC) site,[176]RHT[178]R[179]/S[180]AE[182], which includes a furin type protease cleavage site (R179), an O-glycosylation site (T178) and a serine phosphorylation site (S180).

Three enzymes namely GalNAc-T3 (N-acetylgalactosaminyltransferase 3), Fam20C (the family with sequence similarity 20, member C), and a yet to be discovered furin type protease converge on this site to regulate FGF23 processing. The high flexibility of this region is likely necessary for the action of these enzymes. **B**, FGF23 signaling is strictly αKlotho dependent. L6-FGFR1c[WT] cells were treated with increasing concentrations of recombinant FGF23[WT] alone, in combination with soluble αKlotho or left untreated. Whole cell lysates immunoblotted as in Fig. 2c. Note that even supra pharmacological concentrations (as high as 10 micromolar) of FGF23[WT] fails to activate FGFR1c signaling. However, when co-treated with soluble αKlotho, as little as 10 nM FGF23 induces a robust FGFR1c activation. Experiments were performed in biological triplicates with similar results. **C**, Superimposition of X-ray structure of the FGF23–FGFR1c–αKlotho ternary complex (PDB ID: 5W21, colored in green) onto the corresponding FGF23–FGFR1c[P]–αKlotho portion within the cryo-EM structure of 1:2:1:1 FGF23-FGFR1c-αKlotho-HS quaternary complex (colored in pink) shows high degree of similarity (RM)D of 1.16 Å].

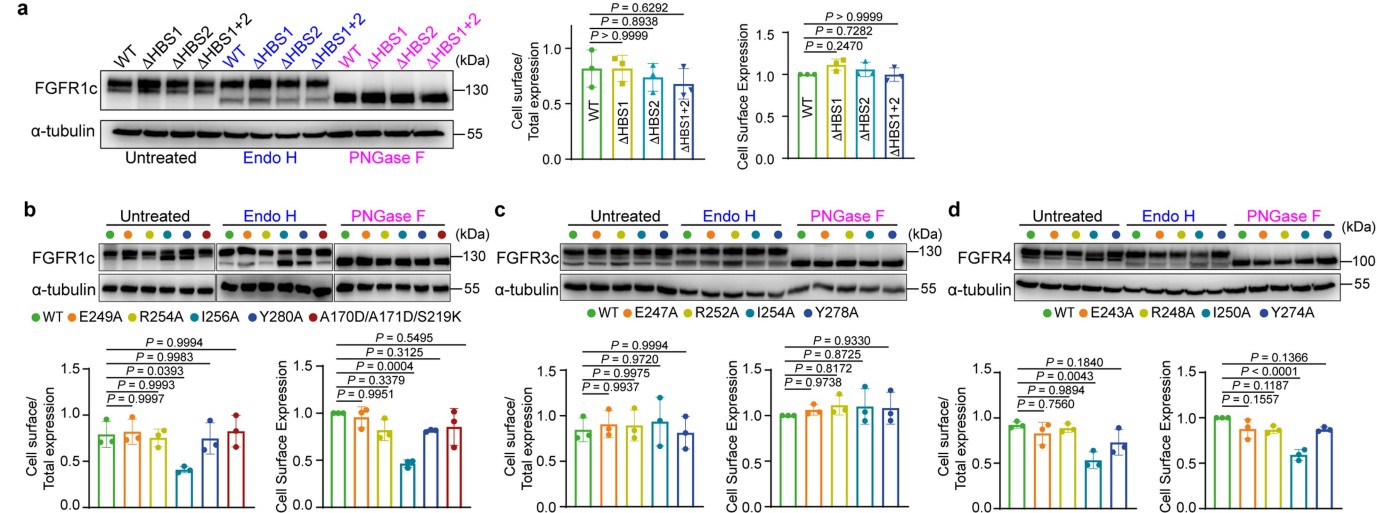

**Extended Data Fig. 3 | Cell surface expression analysis of mutated FGFRs via Endoglycosidase H (Endo H) sensitivity assay. a**, Denatured lysates from L6-FGFR1c$^{WT}$, L6-FGFR1c$^{K175Q/K177Q}$ (FGFR1c$^{ΔHBS1}$), L6-FGFR1c$^{K207Q/R209Q}$ (FGFR1c$^{ΔHBS2}$), and L6-FGFR1c$^{K175Q/K177Q/K207Q/R209Q}$ (FGFR1c$^{ΔHBS1+2}$) cells were incubated with Endoglycosidase H (Endo H) or Peptide-N-Glycosidase F (PNGase F) or left untreated. Samples were immunoblotted with FGFR1c isoform-specific antibody. Immunoblotting data were quantitated as described in the Methods section and are presented as the mean ± SD. **b**–**d**, Denatured lysates from L6 cell lines stably expressing FGFR1c$^{WT}$, its four *Site 2* (FGFR1c$^{E249A}$, FGFR1c$^{R254A}$, FGFR1c$^{I256A}$, FGFR1c$^{Y280A}$) and one *Site 1* (FGFR1c$^{A170D/A171D/S219D}$) mutants (**b**), wild-type FGFR3c (FGFR3c$^{WT}$) or its four *Site 2* mutants (FGFR3c$^{E247A}$, FGFR3c$^{R252A}$, FGFR3c$^{I254A}$, FGFR3c$^{Y278A}$) (**c**), wild-type FGFR4 (FGFR4$^{WT}$) or its four *Site 2* mutants (E243A, R248A, I250A, Y274A) (**d**) were treated with Endo H, PNGase F or left untreated. Samples were immunoblotted with FGFR isoform-specific antibodies as indicated. Experiments were performed in biological triplicates with similar results. Quantitation was done as described in the Methods section and are presented as the mean ± S.D. *P* values were determined by One-way ANOVA followed by Tukey's multiple comparisons *post hoc* test.

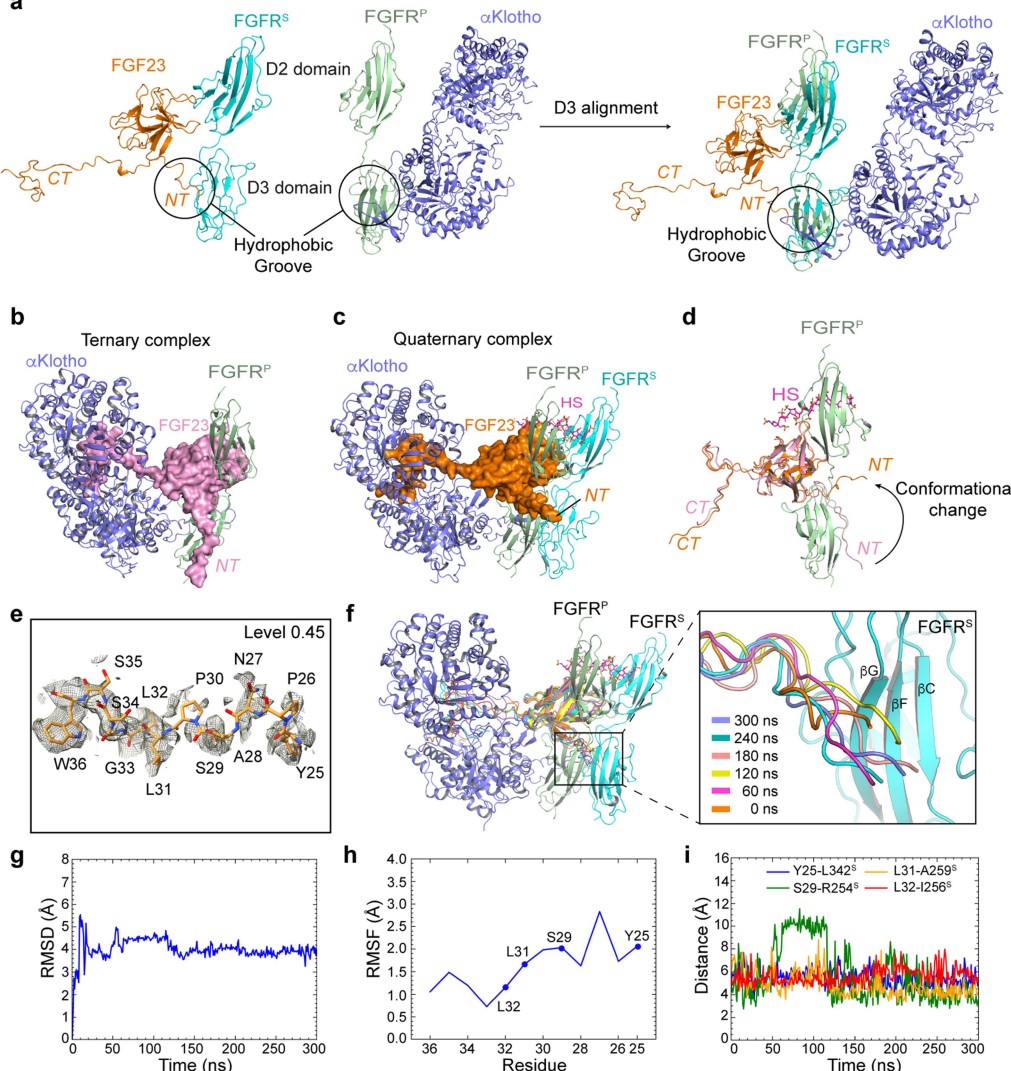

**Extended Data Fig. 4 | FGF23 N-terminus experiences a major conformational change in the quaternary complex. A**, Left and Center: Cartoon representations of the FGF23-FGFR$^S$ and FGFR$^P$-αKlotho components of the quaternary complex shown in the orientation obtained via alignment of D2 domains of FGFR$^S$ and FGFR$^P$. Right: Superimposition of FGF23-FGFR$^S$ and FGFR$^P$-αKlotho components via aligning receptors' D3 domains. Note that FGF23 N-terminus and αKlotho RBA engage the equivalent hydrophobic grooves in D3 of FGFR$^S$ and FGFR$^P$ chains. **b–d**, FGF23 N-terminus undergoes a major conformational change in FGF23–FGFR–αKlotho–HS quaternary complexes. Cartoon and surface (only for FGF23 component) representation of HS-free ternary complex (that is, X-ray structure) (**b**) and HS-induced quaternary complex (that is, cryo-EM structures) (**c**) in the same orientation obtained via alignment of their FGF23 chains. (**d**) Cartoon representation of an overlay between FGF23-FGFR1c from FGF23–FGFR1c–αKlotho ternary complex and FGF23-FGFR1c$^P$ from quaternary complex via FGF23 alignment. Note the dramatic difference in the positions of FGF23 N-terminus between the two structures. **e**, Electron densities of FGF23 N-terminus in cryo-EM structures of FGF23–FGFR1c–αKlotho–HS quaternary complexes at the indicated contour levels. Note that the electron densities for FGF23 N-terminus are weak and patchy. f–i, MD simulation data show that FGF23 N-terminus engages D3 of FGFR1c$^S$. f, Left: Cartoon representation of the cryo-EM structure of FGF23-FGFR1c-

αKlotho-HS quaternary complex. FGF23, FGFR1c$^P$, FGFR1c$^S$ and αKlotho are in orange, green, cyan, and blue, respectively. Right: Zoomed-in view of the boxed region (left) showing conformational progression of the N-terminal tail of FGF23 (that is, Y25 to W36) and D3 of FGFR1c$^S$ in a 300 ns MD simulation trajectory denoted by a color transition from the orange (0 ns) to blue (300 ns) in 60 ns intervals. FGF23 N-terminus interacts with the three stranded βC: βF: βG sheet in FGFR1c$^S$ D3 domain. **g-h**, Changes in RMSD (**g**) and Root Mean Square Fluctuation (RMSF, **h**), respectively, of N-terminal tail of FGF23 during MD simulation. Note that RMSD of N-terminal residues of FGF23 stabilized around 4 Å after 120 ns. Importantly, residues at the distal and proximal ends of FGF23 N-terminus exhibited largest and smallest RMSF, respectively, mirroring their respective cryo-EM electron densities (compare e and h). **i**, Changes in the distances of four selected contact pairs (Y25–L342$^S$, S29–R254$^S$, L31–A259$^S$ and L32–I256$^S$) between N-terminal residues of FGF23 and residues in D3 of FGFR1c$^S$. The distances of Y25–L342$^S$, L31–A259$^S$, and L32–I256$^S$ hydrophobic residue pairs fluctuated around 5 Å indicative of formation of hydrophobic contacts between these residue pairs. Likewise, the pairwise distance for S29-R254$^S$ fluctuated around 4 Å after 120 ns indicative of hydrogen bonding between side chains of this residue pair.

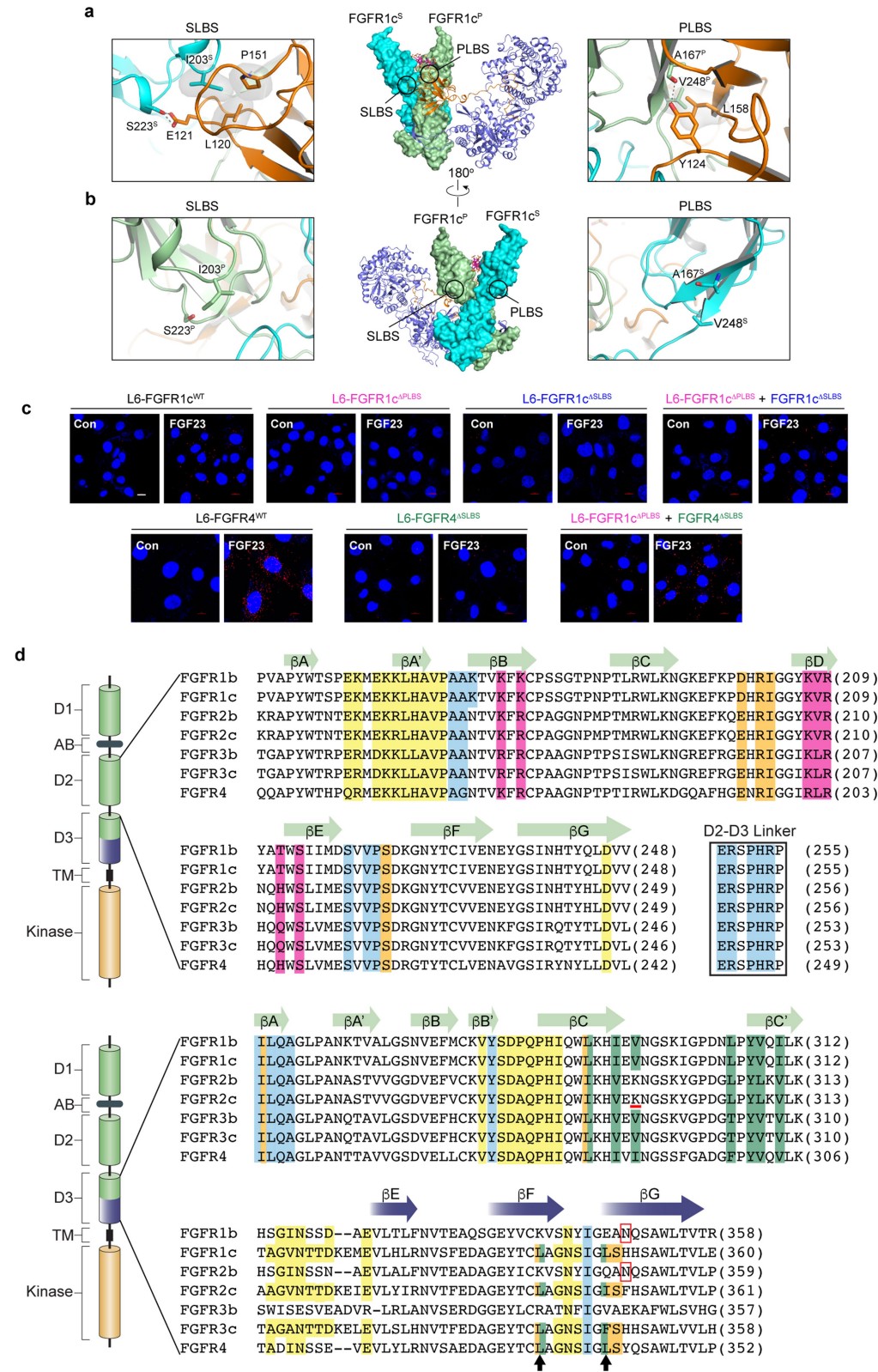

**Extended Data Fig. 5 |** See next page for caption.

**Extended Data Fig. 5 | Rationale for cell-based receptor complementation assay and structure-based sequence alignment of seven FGFR isoforms.**
**a-b**, Rationale for receptor complementation assay. Center: Cartoon/surface representation of the FGF23–FGFR1c–αKlotho–HS quaternary complex in two orientations related by 180° rotation around Y axis. Circles denote approximate circumferences of ligand binding sites in primary and secondary receptors selected for mutagenesis. **a**, Left: expanded view of circled region on the secondary receptor showing that Ile-203 and Ser-223 of FGFR1c$^S$ engage in hydrophobic and hydrogen bonding contacts with residues in FGF23's core. Right: expanded view of circled region on the primary receptor showing that Ala-167 and Val-248 of FGFR1c$^P$ engage in highly conserved hydrophobic contacts with Tyr-124 and Leu-158 of FGF23, respectively. Additionally, Ala-167 also makes a hydrogen bond with Tyr-124. **b**, Left, expanded view of the circled region on the primary receptor showing that the corresponding Ile-203 and Ser-223 in FGFR1c$^P$ are solvent exposed. Thus, an engineered FGFR1c$^{I203E/S223E}$ molecule (i.e., FGFR1c$^{ΔSLBS}$) should preserve the ability to act as primary receptor despite losing the capacity to function as secondary receptor. Right, expanded view of the circled region on secondary receptor showing that the corresponding Ala-167 and Val-248 in FGFR1c$^S$ do not play any role in the quaternary complex formation and are solvent exposed. Thus, an engineered FGFR1c$^{A167D/V248D}$ mutant (i.e., FGFR1c$^{ΔPLBS}$) is predicted to lose the ability to function as primary receptor but could still act as secondary receptor. **c**, Demonstration of receptor complementation between FGFR1c$^{ΔSLBS}$ and FGFR1c$^{ΔPLBS}$ and between FGFR4$^{ΔSLBS}$ and FGFR1c$^{ΔPLBS}$ via PLA. Representative fluorescent microscopy fields of stable cell lines subjected to PLA are shown. Scale bar, 10 μm. Experiments were performed at least two times biological repeat with similar results. **d**, Structure-based sequence alignment of ligand binding regions (i.e., D2, D2-D3 linker and D3) of all seven FGFR isoforms. As a guide, a schematic representation of a prototypical FGFR is shown and its various domains are labeled. The C-terminal half of D3 which undergoes alternative splicing in FGFR1-FGFR3 is highlighted in dark blue. Note that the unspliced N-terminal half of D3 is conserved between b and c isoforms. Secondary structure elements are provided on the top of alignment. The HS interacting residues are all localized within D2 domain and are colored in magenta. Residues mediating the FGF23-FGFR$^P$ and αKlotho-FGFR$^P$ interfaces are colored yellow and green, respectively. Note that residues mediating direct FGFR$^P$-FGFR$^S$ contacts (in light blue) are fully conserved amongst all seven isoforms. As for the FGF23-FGFR$^S$ interface (in orange), only D2 residues that contact FGF23's core, are conserved between the seven isoforms. However, two hydrophobic D3 residues that interact with FGF23 N-terminus at the FGF23-FGFR$^S$ interface are conserved only in FGFR1c-3c and FGFR4 (indicated by black arrow heads). In FGFR1b-3b, these residues, which overlap with binding site for RBA of αKlotho, are replaced by charged residues. Hydrophobicity of D3 groove FGFR1b-3b is further disrupted by the presence of a unique N-linked glycosylation site within their D3 grooves (denoted by a red box). In FGFR2c, a key conserved hydrophobic residue is replaced by a charged Lys-296 (underlined in red) which likely accounts for the inability of FGFR2c to bind αKlotho.

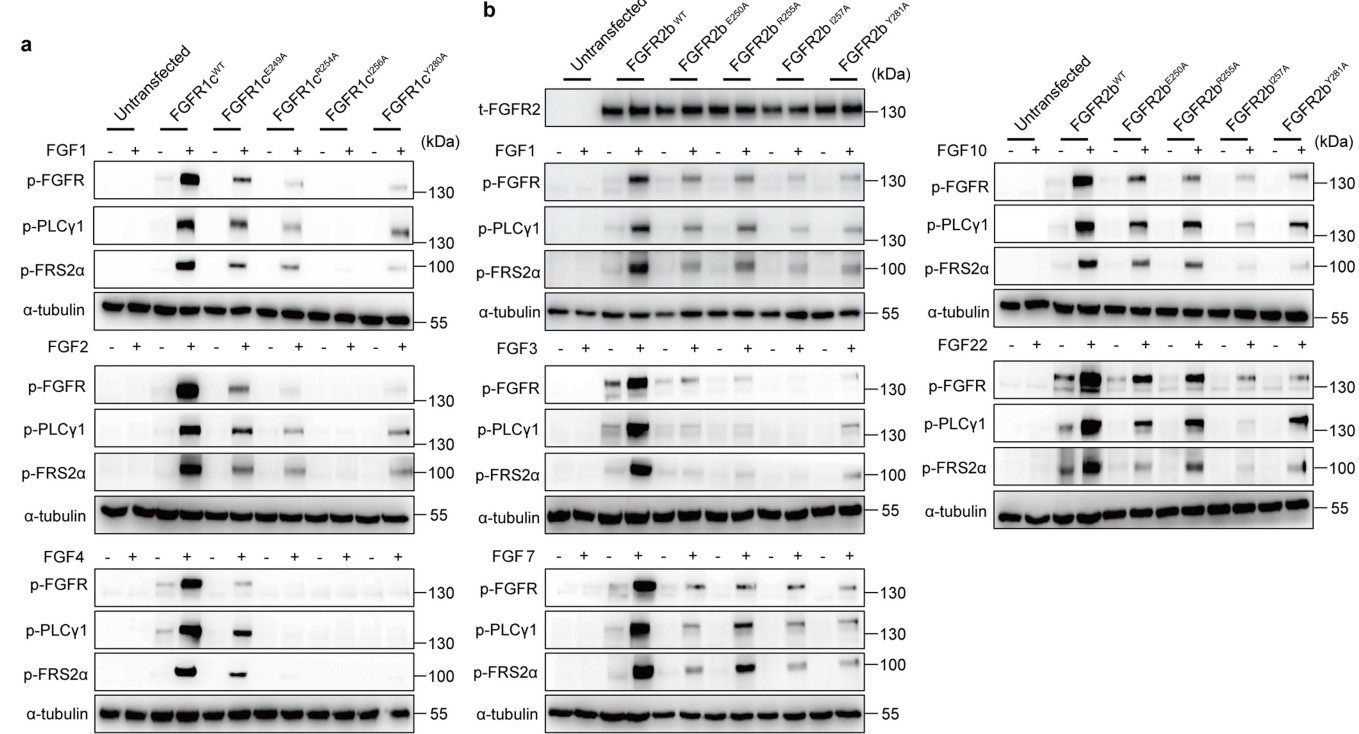

**Extended Data Fig. 6 | The asymmetric FGFR$^P$-FGFR$^S$ interface observed in FGF23-FGFR-αKlotho-HS cryo-EM structures is required for multiple paracrine FGF signaling. a**, Immunoblots of whole cell extracts from untreated or FGF-treated (1 nM) untransfected and transfected L6 cell lines stably expressing FGFR1c$^{WT}$, FGFR1c$^{E249A}$, FGFR1c$^{R254A}$, FGFR1c$^{I256A}$, or FGFR1c$^{Y280A}$ probed with antibodies against phosphorylated FGFR, PLCγ1 and FRS2α.

**b**, Immunoblots of whole cell extracts from untreated or FGF-treated (1 nM for FGF1 and 2 nM for FGF3/7/10/22) treated untransfected and transfected L6 cell lines stably expressing FGFR2b$^{WT}$, FGFR2b$^{E250A}$, FGFR2b$^{R255A}$, FGFR2b$^{I257A}$, or FGFR2b$^{Y281A}$ probed with FGFR2b isoform specific antibody (top) or phosphospecific antibodies as in panel **a**. Experiments were performed in biological triplicates with similar results.

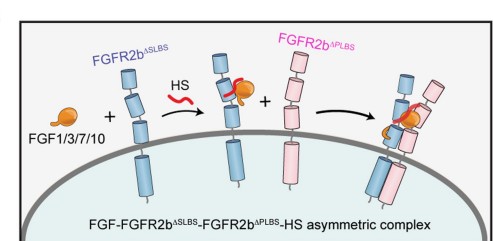

**a**

FGF-FGFR2b^ΔSLBS^-FGFR2b^ΔPLBS^-HS asymmetric complex

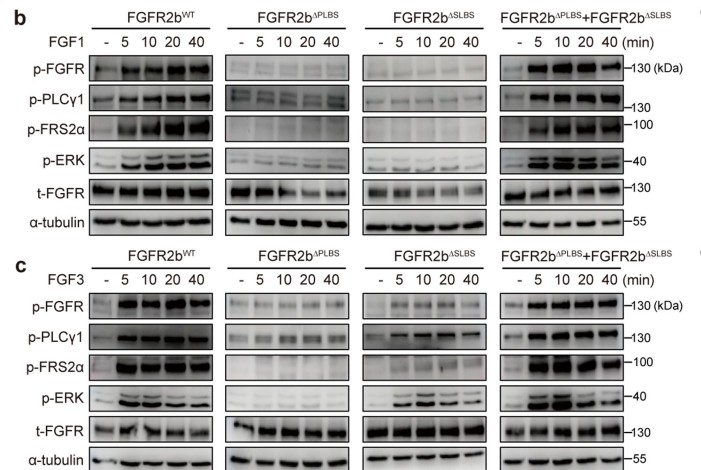

**b**

FGF1

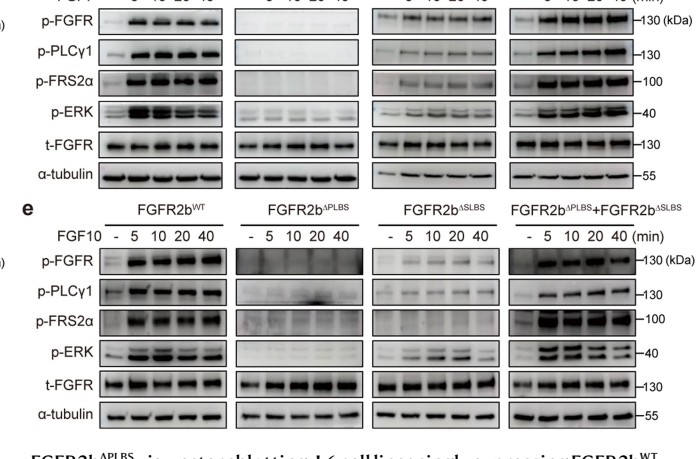

**d**

FGF7

**c**

FGF3

**e**

FGF10

**Extended Data Fig. 7 | Receptor complementation assays demonstrate the asymmetry of four distinct paracrine FGF-FGFR2b-HS signaling complexes.** **a**, Schematic diagram showing that in response to paracrine FGF1/3/7/10 and HS, FGFR2b^ΔSLBS^ and FGFR2b^ΔPLBS^ can complement each other and form 1:1:1:1 FGF-FGFR2b^ΔSLBS^-FGFR2b^ΔPLBS^-HS asymmetric signaling complexes. **b**-**e**, Demonstration of receptor complementation between FGFR2b^ΔSLBS^ and

FGFR2b^ΔPLBS^ via western blotting. L6 cell lines singly expressing FGFR2b^WT^, FGFR2b^ΔPLBS^, FGFR2b^ΔSLBS^, or co-expressing FGFR2b^ΔSLBS^ with FGFR2b^ΔPLBS^ were treated with 1 nM FGF1 (**b**), 2 nM FGF3 (**c**), 2 nM FGF7 (**d**), or 2 nM FGF10 (**e**) for increasing time intervals, and total cell extracts were immunoblotted as indicated. Experiments were performed in biological triplicates with similar results.

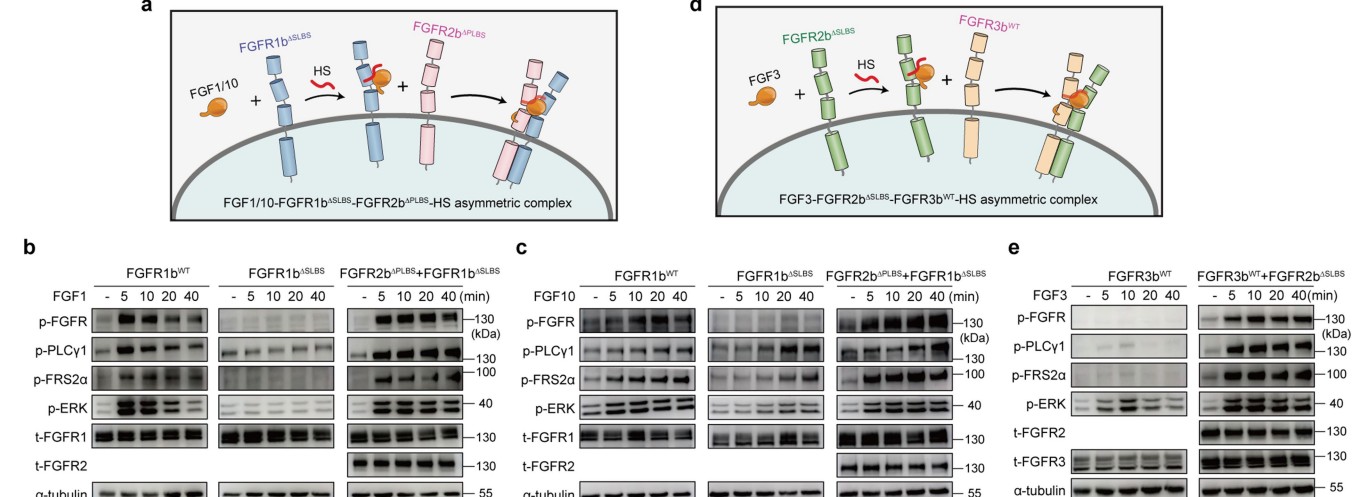

**Extended Data Fig. 8 | Receptor heterodimerization assays validate the asymmetry of paracrine FGF-FGFR-HS signaling complexes. a**, Schematic diagram showing that FGF1 or FGF10, HS, and FGFR1b$^{\Delta SLBS}$ (serving as primary receptor) form stable complexes which subsequently recruit FGFR2b$^{\Delta PLBS}$ as secondary receptor. **b-c**, L6 cell lines singly expressing FGFR1b$^{WT}$ or FGFR1b$^{\Delta SLBS}$, or co-expressing FGFR1b$^{\Delta SLBS}$ + FGFR2b$^{\Delta PLBS}$ were treated with 1 nM FGF1 (**b**) or 2 nM FGF10 (**c**) for increasing time intervals and cell extracts were immunoblotted.

**d**, Schematic diagram showing that in the presence of HS, FGF3 and FGFR2b$^{\Delta SLBS}$ (serving as primary receptor) form a stable complex and subsequently recruit FGFR3b$^{WT}$ as secondary receptor. **e**, L6 cell lines expressing FGFR3b$^{WT}$ alone or co-expressing it with FGFR2b$^{\Delta SLBS}$ were treated with 2 nM FGF3 for increasing time intervals and cell extracts were immunoblotted. Experiments were performed in biological triplicates with similar results.

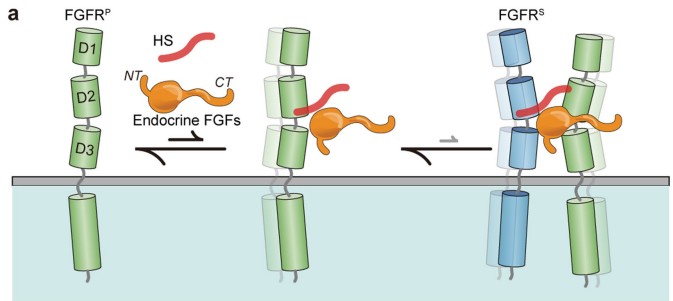

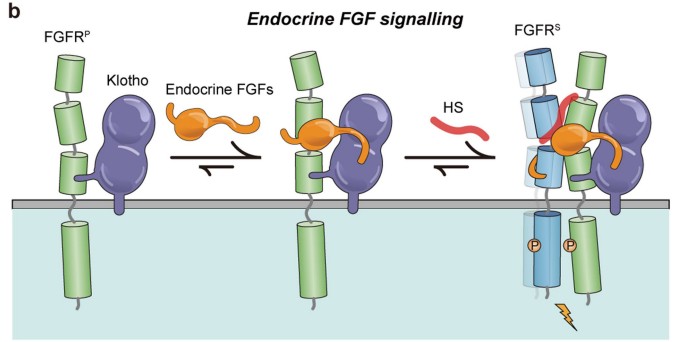

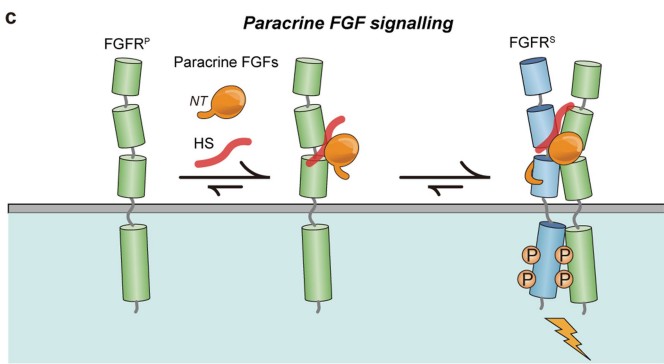

**Extended Data Fig. 9 | Asymmetric receptor dimerization is a universal mechanism in FGF signaling. a**, Due to FGF hormone's weak HS binding affinity, HS alone is incompetent in stabilizing the endocrine FGF-FGFR complex and inducing sustained asymmetric receptor dimerization/activation. Blurring and loose association are used to emphasize the unstable/transient nature of putative FGF-FGFR-HS ternary complex and physiologically inconsequential receptor dimerization/activation. **b**, Membrane bound Klotho co-receptor simultaneously engages FGFR's D3 domain and FGF's C-terminal tail thereby stabilizing the endocrine FGF-FGFR complex within a ternary complex. In so doing, Klotho co-receptor effectively compensates for HS incompetency in stabilizing binary endocrine FGF-FGFR complex. HS is now in position to recruit a second FGFR to the stabilized binary complex thus inducing asymmetric dimerization. Nevertheless, due to FGF hormone's weak HS binding affinity, Klotho and HS-induced endocrine 1:2 FGF-FGFR dimers are still inferior to HS-induced paracrine 1:2 FGF-FGFR dimers in terms of longevity/stability (indicated by slight blurring of FGFR$^S$ in **b**). **c**, Due to their high HS binding affinities, paracrine FGFs can rely on HS as the sole co-receptor to stably bind primary FGFR and recruit a secondary FGFR, thereby inducing formation of rigid and long-lived asymmetric receptor dimers.

**Extended Data Table 1 | Cryo-EM data collection, refinement and validation statistics**

| | #1 FGF23-FGFR1c-αKlotho-HS (EMDB-34075) (PDB 7YSH) | #2 FGF23-FGFR3c-αKlotho-HS (EMDB-34082) (PDB 7YSU) | #3 FGF23-FGFR4-αKlotho-HS (EMDB-34084) (PDB 7YSW) |
|---|---|---|---|
| **Data collection and processing** | | | |
| Magnification | 36,000 x | 130,000 x | 36,000 x |
| Voltage (kV) | 200 | 300 | 200 |
| Electron exposure (e–/Å$^2$) | 50.37 | 72.44 | 53.84 |
| Defocus range (μm) | -0.7 to -2.2 | -0.6 to -2.8 | -0.7 to -2.8 |
| Pixel size (Å) | 1.096 | 1.048 | 1.096 |
| Symmetry imposed | C1 | C1 | C1 |
| Initial particle images (no.) | 10,186 | 6,409 | 16,602 |
| Final particle images (no.) | 9501 | 5164 | 15,049 |
| Map resolution (Å) FSC threshold | 2.74 | 3.20 | 3.03 |
| | | | |
| **Refinement** | | | |
| Initial model used (PDB code) | 5W21 | 5W21 | 5W21 |
| Model resolution (Å) FSC threshold | 2.76, 2.71 (masked) | 3.25, 3.18 (masked) | 3.05, 3.00 (masked) |
| Map sharpening $B$ factor (Å$^2$) | 139.4 | 144.3 | 131.2 |
| Model composition | | | |
| Non-hydrogen atoms | 12480 | 12428 | 12387 |
| Protein residues | 1525 | 1521 | 1517 |
| Ligands | / | / | / |
| $B$ factors (Å$^2$) | | | |
| Protein | 70.185 | 121.374 | 92.158 |
| Ligand | / | / | / |
| R.m.s. deviations | | | |
| Bond lengths (Å) | 0.007 | 0.016 | 0.008 |
| Bond angles (°) | 0.952 | 1.19 | 1.017 |
| Validation | | | |
| Clashscore | 5.47 | 7.64 | 6.26 |
| Poor rotamers (%) | 0.53 | 0.69 | 0.46 |
| Ramachandran plot | | | |
| Favored (%) | 91.95 | 89.53 | 90.69 |
| Allowed (%) | 7.92 | 10.34 | 9.31 |
| Disallowed (%) | 0.13 | 0.13 | 0 |

| | #1 FGF23-FGFR1c-αKlotho-HS | #2 FGF23-FGFR3c-αKlotho-HS | #3 FGF23-FGFR4-αKlotho-HS |
|---|---|---|---|

# Reporting Summary

## Statistics

For all statistical analyses, confirm that the following items are present in the figure legend, table legend, main text, or Methods section.

| n/a | Confirmed | |
|---|---|---|
| ☐ | ☒ | The exact sample size (*n*) for each experimental group/condition, given as a discrete number and unit of measurement |
| ☐ | ☒ | A statement on whether measurements were taken from distinct samples or whether the same sample was measured repeatedly |
| ☐ | ☒ | The statistical test(s) used AND whether they are one- or two-sided *Only common tests should be described solely by name; describe more complex techniques in the Methods section.* |
| ☒ | ☐ | A description of all covariates tested |
| ☒ | ☐ | A description of any assumptions or corrections, such as tests of normality and adjustment for multiple comparisons |
| ☐ | ☒ | A full description of the statistical parameters including central tendency (e.g. means) or other basic estimates (e.g. regression coefficient) AND variation (e.g. standard deviation) or associated estimates of uncertainty (e.g. confidence intervals) |
| ☐ | ☒ | For null hypothesis testing, the test statistic (e.g. *F*, *t*, *r*) with confidence intervals, effect sizes, degrees of freedom and *P* value noted *Give P values as exact values whenever suitable.* |
| ☒ | ☐ | For Bayesian analysis, information on the choice of priors and Markov chain Monte Carlo settings |
| ☒ | ☐ | For hierarchical and complex designs, identification of the appropriate level for tests and full reporting of outcomes |
| ☒ | ☐ | Estimates of effect sizes (e.g. Cohen's *d*, Pearson's *r*), indicating how they were calculated |

*Our web collection on statistics for biologists contains articles on many of the points above.*

## Software and code

Policy information about availability of computer code

| Data collection | Leginon (version 3.5) Data of Sec-Mals was collected by ASTRA 5.3.4.20 Grey values of band in Western blots were collected by ImageJ 1.57j8 |
|---|---|
| Data analysis | CryoSparc (version 2.15); Phenix suite (version 1.9_1692); COOT (version 0.8.2); PyMOL (version 2.5.2); UNICORN (version 7.6, GE Healthcare, USA); UCSF Chimera X (version 1.3); GraphPad (version 8.0, GraphPad Software, Inc., USA). |

For manuscripts utilizing custom algorithms or software that are central to the research but not yet described in published literature, software must be made available to editors and reviewers. We strongly encourage code deposition in a community repository (e.g. GitHub). See the Nature Portfolio guidelines for submitting code & software for further information.

## Data

Policy information about availability of data

All manuscripts must include a data availability statement. This statement should provide the following information, where applicable:
- Accession codes, unique identifiers, or web links for publicly available datasets
- A description of any restrictions on data availability
- For clinical datasets or third party data, please ensure that the statement adheres to our policy

Electron density maps and refined models for the FGF23–FGFR1c–αKlotho–HS (EMD-34075, 7YSH), FGF23–FGFR3c–αKlotho–HS (EMD-34082, 7YSU) and FGF23–

FGFR4–aKlotho–HS (EMD-34084, 7YSW) quaternary complexes have been deposited in the Electron Microscopy Data Bank and will be released upon acceptance of the manuscript. All the data are available within the paper and its supplementary information files. Source data are provided with this paper.

# Human research participants

Policy information about studies involving human research participants and Sex and Gender in Research.

| Reporting on sex and gender | n/a |
| --- | --- |
| Population characteristics | n/a |
| Recruitment | n/a |
| Ethics oversight | n/a |

Note that full information on the approval of the study protocol must also be provided in the manuscript.

# Field-specific reporting

Please select the one below that is the best fit for your research. If you are not sure, read the appropriate sections before making your selection.

☒ Life sciences ☐ Behavioural & social sciences ☐ Ecological, evolutionary & environmental sciences

For a reference copy of the document with all sections, see nature.com/documents/nr-reporting-summary-flat.pdf

# Life sciences study design

All studies must disclose on these points even when the disclosure is negative.

| Sample size | Sample sizes are given in the manuscript. All the cell based FGFR autophosphorylation and signaling studies were repeated at least three times and PLA data were generated by six randomly chosen microscope fields from at least two independent experiment, which is sufficient to derive error bars, p values and statistical significance. No sample-size calculation was performed. |
| --- | --- |
| Data exclusions | No data were excluded from the analyses. |
| Replication | All replication were successful, and the replication numbers are either mentioned in the figure legends or Methods section. Protein purifications were repeated at least 8 times and showed similar chromatography and electrophoresis patterns. Western blot experiments were repeated in biological triplicates. PLA assay were repeated at least two times independently. |
| Randomization | PLA data were generated by six randomly chosen microscope fields from at least two independent experiment. For all other experiments, all data were used for analysis, thus no randomization was needed. |
| Blinding | Blinding applied during protein complex particle picking and Ab-initio reconstruction and heterogeneous refinement. Other experiments does not relevant to blinding as no groups were assigned. |

# Reporting for specific materials, systems and methods

We require information from authors about some types of materials, experimental systems and methods used in many studies. Here, indicate whether each material, system or method listed is relevant to your study. If you are not sure if a list item applies to your research, read the appropriate section before selecting a response.

## Materials & experimental systems

| n/a | Involved in the study |
| --- | --- |
| ☐ | ☒ Antibodies |
| ☐ | ☒ Eukaryotic cell lines |
| ☒ | ☐ Palaeontology and archaeology |
| ☒ | ☐ Animals and other organisms |
| ☒ | ☐ Clinical data |
| ☒ | ☐ Dual use research of concern |

## Methods

| n/a | Involved in the study |
| --- | --- |
| ☒ | ☐ ChIP-seq |
| ☒ | ☐ Flow cytometry |
| ☒ | ☐ MRI-based neuroimaging |

# Antibodies

| | |
|---|---|
| Antibodies used | phosphorylated FGFR (#3471S, Cell Signaling Technology, USA); phosphorylated FRS2α (#3864S, Cell Signaling Technology, USA); phosphorylated PLCγ1 (#2821S, Cell Signaling Technology, USA); phosphorylated ERK1/2 (#4370S, Cell Signaling Technology, USA); α-tubulin (#66031-1-Ig, Proteintech, China); total-FGFR1 (#9740S, Cell Signaling Technology, USA), total-FGFR2 (#23328S, Cell Signaling Technology, USA), total-FGFR3 (#ab133644, Abcam, UK), total-FGFR4 (#8562S, Cell Signaling Technology, USA), HRP conjugated Goat anti-mouse IgG (H+L) (#SA00001-1, Proteintech, China), HRP conjugated Goat Anti-Rabbit IgG(H+L) (#SA00001-2, Proteintech, China); FGFR1 (#PA5-25979, ThermoFisher, USA); FGFR1 (#ab824, Abcam, UK); FGFR4 (#sc-136988, Santa Cruz Biotechnology, USA) |
| Validation | All of the antibodies used for western experiments and PLA are commercially available products, validation are available on manufacturers' websites.<br>anti-phosphorylated FGFR, Cell Signaling Technology, 3471S, western blot (1:1000): https://www.cellsignal.cn/products/primary-antibodies/phospho-fgf-receptor-tyr653-654-antibody/3471<br>anti-phosphorylated FRS2α, Cell Signaling Technology, 3864S, western blot (1:1000): https://www.cellsignal.cn/products/primary-antibodies/phospho-frs2-a-tyr196-antibody/3864<br>anti-phosphorylated PLCγ1, Cell Signaling Technology, 2821S, western blot (1:1000): https://www.cellsignal.cn/products/primary-antibodies/phospho-plcg1-tyr783-antibody/2821<br>anti-phosphorylated ERK1/2, Cell Signaling Technology, 4370S, western blot (1:1000): https://www.cellsignal.cn/products/primary-antibodies/phospho-p44-42-mapk-erk1-2-thr202-tyr204-d13-14-4e-xp-rabbit-mab/4370<br>anti-α-tubulin, Proteintech, 66031-1-Ig, western blot (1:20000): https://ptgcn.com/products/tubulin-Alpha-Antibody-66031-1-Ig.htm<br>anti-total-FGFR1, Cell Signaling Technology, 9740S, western blot (1:1000): https://www.cellsignal.cn/products/primary-antibodies/fgf-receptor-1-d8e4-xp-rabbit-mab/9740<br>anti-total-FGFR2, Cell Signaling Technology, 23328S, western blot (1:1000): https://www.cellsignal.cn/products/primary-antibodies/fgf-receptor-2-d4l2v-rabbit-mab/23328<br>anti-total-FGFR3, Abcam, ab133644, western blot (1:1000): https://www.abcam.cn/products/primary-antibodies/fgfr3-antibody-epr23043-ab133644.html<br>anti-total-FGFR4, Cell Signaling Technology, 8562S, western blot (1:1000) and PLA (1:100): https://www.cellsignal.cn/products/primary-antibodies/fgf-receptor-4-d3b12-xp-rabbit-mab/8562<br>anti-HRP conjugated Goat anti-mouse IgG (H+L), Proteintech, SA00001-1, western blot (1:5000): https://ptgcn.com/products/HRP-conjugated-Affinipure-Goat-Anti-Mouse-IgG-H-L-secondary-antibody.htm<br>anti-HRP conjugated Goat anti- Rabbit IgG (H+L), Proteintech, SA00001-2, western blot (1:5000): https://ptgcn.com/products/HRP-conjugated-Affinipure-Goat-Anti-Rabbit-IgG-H-L-secondary-antibody.htm<br>anti-FGFR1, ThermoFisher, PA5-25979, PLA (1:20): https://www.thermofisher.cn/cn/zh/antibody/product/FGFR1-Antibody-Polyclonal/PA5-25979<br>anti-FGFR1, Abcam, ab824, PLA (1:100): https://www.abcam.cn/products/primary-antibodies/fgfr1-antibody-m5g10-ab824.html<br>anti-FGFR4, Santa Cruz Biotechnology, sc-136988, PLA (1:100): https://www.scbt.com/zh/p/fgfr-4-antibody-a-10 |

# Eukaryotic cell lines

Policy information about cell lines and Sex and Gender in Research

| | |
|---|---|
| Cell line source(s) | L6 myoblast cell line (#GNR 4, National Collection of Authenticated Cell Cultures, China); HEK293T cells (kindly provided by Cell Bank/Stem Cell Bank, Chinese Academy of Sciences, China); N-acetylglucosaminytransferase I (GnTI) deficient HEK293S cells (#CRL-3022, American Type Culture Collection, USA). |
| Authentication | The cell lines were identified by morphology check under microscope in the lab. |
| Mycoplasma contamination | Mycoplasma negative per DAPI staining. |
| Commonly misidentified lines<br>(See ICLAC register) | No commonly misidentified cell lines were used. |

