## [Peer Review File · Nature]

Manuscript Title: Structural Basis for FGF hormone Signaling

Reviewer Comments & Author Rebuttals

Reviewer Reports on the Initial Version:

Referee expertise:

Referee #1: cryo-EM, RTKs

Referee #2: FGF signaling, Klotho

Referee #3: cryo-EM, RTKs

Referees' comments:

Referee #1 (Remarks to the Author):

Chen et al. describe cryoEM studies of FGF23/FGFR/aKlotho/HS dodecasaccharide complexes that suggest an unexpected 1:2:1:1 stoichiometry whether the FGFR ECD is from FGFR1c, FGFR3c, or FGFR4. The Klotho/FGF23/FGFR1c(P) complex very closely resembles the crystal structure of the 1:1:1 complex that was already published. When HS is also included, however, it cooperates with the single FGF23 and the FGFR1c ECD to recruit a second FGFR ECD (FGFRs), which the authors argue is key to receptor activation. This arrangement is quite unexpected and is interesting. The fact that essentially the same arrangement is seen whether the FGFR ECD is from FGFRs 1, 3, 4 or also adds to the conviction that this could represent the situation at the cell surface. The authors perform mutational studies to support the interfaces seen in the complex in the EM structure.

There are some puzzles here. First, if aKlotho is so important for FGF23 signaling, how does it influence binding of the secondary FGFR? And why would it distort the secondary FGFR ECD? It looks as if the FGFRp/FGF23/HS complex should bind FGFRs even in the absence of Klotho if concentrations are high enough. This should be tested. Second, although about 60% of the FGFR1 particles have the organization focused on in the figures, only about 23% or 33% do for FGFR3 and FGFR4 respectively. What are the other structures? Why were they ignored? This is not discussed sufficiently (at all) in the manuscript, and the basis for the choice of the structure described is not provided. This raises doubts about whether this is really a relevant structure. This issue is frequently seen in cryoEM, and the authors need both to justify their selection of this particle type and to make some comments about the other classes. They also need to add some controls to the mutational studies, since similar interactions may also be relevant in FGF1 signaling for example.

Other comments:

1. The introduction seems very long.

2. The phrasing 'hardcore paracrine-acting ligands' on line 57 seems out of place.
3. The discussion on line 184 about the FGF23-FGFR1P interaction appearing 'tighter' here than in the prior crystal structure is too speculative, and should be removed. Comparing H-bond lengths in structures determined by two different methods does not make a compelling argument for interaction strength. This needs to be assessed experimentally. If the authors wish to argue that Klotho makes FGFRp more rigid and this aids recruitment of the FGFRs, this needs to be assessed experimentally.
4. The structure is based on 1,497,967 of 2,426,961 (62%) of particles for FGFR1, What are the other structures? Including the larger one for FGFR1 (middle). For FGFR3, the complex shown represents 23% of particles (291,540 of 1,272,758), and for FGFR4 it is about 33%. These numbers raise doubt about whether there are multiple forms of the complex, but the authors do not discuss this possibility. The question as to whether this is a unique (or even major) complex is also raised by the nature of the SEC traces in Extended Data Fig. 1 and Fig 3d. There seems to be a very mixed population that the authors have not characterized in terms of different stoichiometries.
5. The detailed discussion of the FGFRP-FGFRS interface on page 7 does not help in understanding.
6. It is important that the authors show that the mutations tested in Figure 4 do not also impair FGFR activation by non-endocrine FGFs in cells. Although these interactions are not evident in previous crystal structures of FGF-driven dimers of FGFR ECDs, it is possible that those structures do not represent the cell surface situation, where tethering to the membrane could promote membrane-proximal interactions between FGFR ECDs (especially D3) that might be important for dimer stability in a signaling context. In other words, for this work, it is essential that the authors determine whether the interactions investigated here are specific to endocrine FGF. This is a crucial control that is missing and should be easy to include.
7. In the studies of FGFR mutants, cell surface receptor expression needs to be shown by flow cytometry or immunofluorescence. The total FGFR blots for FGFR1 and 4 show different distributions of upper and lower bands, raising concerns about integrity that should be controlled for in a complete study.
8. Why is FRS2 phosphorylation so much reduced for FGFR3 I254A? Also, why is ERK signaling not reduced more in the FGFR1 R254A case?
9. One thing that the manuscript lacks is a proper biophysical characterization of the 1:2:1:1 complex. The SEC-MALS studies in Fig. 3d are not very compelling or well presented. The authors should consider AUC or scattering (light or X-ray) approaches to better define the stoichiometry, which seems very variable based on SEC data and the EM particle distribution. There is always the worry that the complexes seen in the EM studies reflect the behavior of the proteins when frozen on grids. Also, it is clear many particles (the majority for FGFR3 and FGFR4) have structure distinct from what the authors focus on here. What are the other classes?
10. The complementation experiment in Figure 6 is a nice idea, but again it needs to be determined

whether this is unique to the situation with FGF23 and Klotho. Controls with non-endocrine FGFs are needed – based on the same argument made in point 6.

11. I have the sense that the figures could be better designed. Figures 1 and 2 seem to show essentially the same thing, and Figure 3a and b the same thing again. These could all be consolidated in one clearer figure.

12. There is no quantitation of any of the signaling work. Just a statement that the experiments were performed three times.

13. The final point about heterodimerization is puzzling given that Klotho plays no part in the FGFR/FGFR interactions. Can this same experiment be done with non-endocrine FGFs? Is heterodimerization selectivity different in the case of FGF23 compare with other FGFs?

Referee #2 (Remarks to the Author):

General comments

The authors previously solved the structure of the 1:1:1 ternary complex composed of fibroblast growth factor-23 (FGF23), fibroblast growth factor receptor (FGFR), and alphaKlotho. Because dimerization of FGFRs facilitated by heparan sulfate (HS) is required to activate FGFR tyrosine kinase, it has been postulated that formation of symmetric 2:2:2:2 complexes composed of FGF23, FGFR, alphaKlotho, and HS would be generated upon activation of the canonical intracellular FGF signaling pathway. Contrary to the expectation, the present study has revealed an unexpected structure of the active cell surface complex using cryo-EM, which is composed of asymmetric 1:2:1:1 FGF23-FGFR-alphaKlotho-HS. This is a unique ligand-receptor structure unknown so far. In addition, the authors introduced multiple mutations in FGF23 and FGFRs that were predicted to abolish formation of the quaternary complex and verified that such mutant proteins were unable to signal in cell culture experiments. The data presented are comprehensive and compelling, supporting the authors' conclusion.

This reviewer has no major concerns, but the authors should address the following specific comments.

1) In Figure 4, was the amount of mutant FGFRs expressed on the cell surface equivalent to that of wild-type FGFRs? Because mutant membrane proteins often fail to be transported to the cell surface, one may argue that the reduced ability of the mutant FGFRs to activate the FGF signaling may be due to their failure to appear on the cell-surface. Cell-surface biotinylation assay may help.

2) Line 205 (Page 7): "All three subdomains (i.e. D2, D2-D2 linker, and D3)" should be "All three subdomains (i.e. D2, D2-D3 linker, and D3)".

3) What was the final concentration of FGF23 and Klotho in Figure 4b-d?

4) What was the final concentration of the FGF23 wild-type/mutant proteins in Figure 5b-d? How did the authors confirm that the same amounts of FGF23 wild-type/mutant proteins were applied to the cells?

Referee #3 (Remarks to the Author):

FGF19, FGF21, and FGF23 belong to a family of endocrine FGFs, and the FGF23-FGFR- α Klotho axis plays an important role in regulating the homeostasis of vitamin D and phosphate. The crystal structure of 1:1:1 FGF23-FGFR- α Klotho has been determined before, showing how α Klotho acts as a scaffold that simultaneously recruits FGF23 and FGFR. Here, Chen et. al presents a series of cryo-EM structures of FGF23-FGFR- α Klotho in complex with heparan sulfate (HS), including FGF23-FGFR1- α Klotho-HS, FGF23-FGFR3- α Klotho-HS, and FGF23-FGFR4- α Klotho-HS. These new cryo-EM structures reveal an unexpected 1:2:1:1 stoichiometry in the formation of FGF23-FGFR4- α Klotho-HS complex, and uncover several new interfaces between FGF23 and FGFR-s and between FGFR-s and FGFR-p that are critical for the complex formation and receptor activation. The authors also propose that the N-terminal region of FGF23 might be important in the assembly of 1:2:1:1 complex. The functional significance of some of the newly discovered interfaces were validated by mutagenesis and cell-based experiment. The authors also design a nice complementation assay to prove that the 1:2:1:1 asymmetric complex is a functionally relevant conformation and is required for receptor activation. Overall, these structural works are very interesting. By revealing the structures of FGF23-FGFR- α Klotho in the HS bound, active state, this work significantly advances our understanding of the mechanism underlying the FGF23 induced activation of FGFR. However, there are a few major issues need to be addressed before this work could be published. My specific points are:

(1) In all the three cryo-EM maps reported from this work, it is clear that there are large unmodelled densities presented. See the attached image. Such large density certainly should not be ignored. Although being resolved at low resolution, the shape and size of the density is highly similar to a α Klotho. See the attached image: the model of a α Klotho shown in green is fitted into the unmodelled cryo-EM density. This suggest that a second α Klotho may be involved in the formation of FGF23-FGFR4- α Klotho-HS complex, leading to a stoichiometry of 1:2:2:1, rather than the 1:2:1:1 proposed by the authors. The flexible binding of a second α Klotho may further stabilize the interact between FGFR-s and FGFR-p or between FGFR-s and FGF23, so it might be functionally important. Alternatively, this second α Klotho could potentially lead to the formation of higher ordered assembly of FGFR by recruiting a second set of FGF23 and FGFR-p. Therefore, the authors need to improve the quality of this unmodeled density to reveal its identity by collecting more cryo-EM data and/or performing local 3D classification/refinement. If this is indeed another α Klotho, the functional significance of this second α Klotho needs to be tested by mutagenesis and cell-based experiments, and its functional role needs to be discussed in the paper.

(2) The authors propose that the N-terminal loop of FGF23 engages FGFR-s, thereby contributing to the interaction between FGF23 and FGFR-s. However, I think the N-terminal loop of FGF23 was not resolved in the cryo-EM map at all. Even at low contour level, I can't observe any cryo-EM density corresponding to the N-terminal loop of FGF23. See the attached image. Therefore, the modelling of N-terminal loop of FGF23 shown in Fig 5 is very questionable. The functional role of the N-terminal

loop of FGF23 also can't be strongly supported by the functional experiment. The authors need to improve the cryo-EM density for this region. Again, they could collect more cryo-EM data, perform more extensive global/local 3D classification.

(3) The author didn't perform any mutagenesis to validate the binding mode of HS at FGFRs. It is, no doubt, that such validation is required as the HS binding plays a critical role in the receptor activation by gluing the two FGFRs together. Therefore, the authors need to introduce some mutations to the interfaces between FGFR-p and HS and between FGFR-s and HS, and test the effect of these mutation on the formation of quaternary complex by using the SEC experiment shown in Fig. 3d as well as receptor activation.

(4) The authors tested the importance of site 2 interface between FGFR-s and FGFR-p by using mutagenesis. However, it is unclear to me why they didn't test the importance of site 1 interface between FGFR-s and FGFR-p using the same method. Such validation is required when the complex structure is determined for the first time.

(5) The authors designed a few FGF23 mutants, and compare their biological activities with FGF23 WT. The authors need to show the purity and quantity of purified FGF23 WT and mutants used in the cell-based experiments by SDS-PAGE. This is important because they need to exclude the possibility that lower activities of FGF23 mutants are due to the impurity of the protein.

(6) As shown in Fig 4, the authors modelled a Cu ion between FGFR-s and FGFR-p. The rationale of such ion assignment is unclear to me. Other divalent metal ions, such as zinc, could also potentially bind to this site. The authors need to provide more evidence to support such ion assignment and modelling.

(7) There is no cryo-EM density to support the modelling of a long loop of FGF23 (residues 172 – 182). As this loop doesn't contact either FGFR or α Klotho, it probably becomes very flexible. This loop needs to be removed from the model to prevent any misleading.

(8) It would be useful to prepare a supplementary figure to compare the structure of FGF23-FGFR- α Klotho-HS with that of FGF1-FGFR. This would show the structural differences of FGFR induced by different families of FGFs, and help the readers to better understand the functional role of Klotho in the activation of FGFR.

Referee #1

General comments: Chen et al. describe cryoEM studies of FGF23/FGFR/ α Klotho/HS dodecasaccharide complexes that suggest an unexpected 1:2:1:1 stoichiometry whether the FGFR ECD is from FGFR1c, FGFR3c, or FGFR4. The Klotho/FGF23/FGFR1c(P) complex very closely resembles the crystal structure of the 1:1:1 complex that was already published. When HS is also included, however, it cooperates with the single FGF23 and the FGFR1c ECD to recruit a second FGFR ECD (FGFRs), which the authors argue is key to receptor activation. This arrangement is quite unexpected and is interesting. The fact that essentially the same arrangement is seen whether the FGFR ECD is from FGFRs 1, 3, 4 or also adds to the conviction that this could represent the situation at the cell surface. The authors perform mutational studies to support the interfaces seen in the complex in the EM structure.

Response: We thank the reviewer's appreciation of our asymmetric dimerization model. As to biological relevance of this model, we did systematic dimer interface mutagenesis experiments, including highly stringent receptor complementation and heterodimerization assays in the context of cultured cells. Our cell-based data unequivocally establish that the 1:2:1:1 FGF23-FGFR- α Klotho-HS asymmetric dimers represent the actual situation on the cell surface. Moreover, we are excited to report that, upon addressing the specific comments raised by the reviewer, we discovered that non-endocrine (i.e., paracrine) FGFs also signal via 1:2:1 FGF-FGFR-HS asymmetric receptor dimers arranged in an identical manner as in the 1:2:1:1 FGF23-FGFR- α Klotho-HS asymmetric dimers. The applicability of our asymmetric model to the paracrine FGFs further underscore the physiological importance of the 1:2:1:1 FGF23-FGFR- α Klotho-HS asymmetric complexes.

Reviewer 1 has a total of 16 critiques/comments with several concerning common issues. Specifically, in comments #2, #7 and #12, reviewer requests information on whether FGF23 cell surface signaling unit could entail other/larger arrangements/stoichiometry besides the 1:2:1:1 FGF23-FGFR- α Klotho-HS complex described in the manuscript. Accordingly, we have merged these three critiques into one (i.e., #2-7-12) and will address them together. Likewise comments #3, #9, #13 and #16 concern the common question of whether the non-endocrine (i.e., paracrine) FGFs also signal via asymmetric dimers. Hence, we will address these together under the aggregate critique #3-9-13-16.

Critique #1: There are some puzzles here. First, if α Klotho is so important for FGF23 signaling, how does it influence binding of the secondary FGFR? And why would it distort the secondary FGFR ECD? It looks as if the FGFR^P/FGF23/HS complex should bind FGFRs even in the absence of Klotho if concentrations are high enough. This should be tested.

Response: There seems to be a misunderstanding as to the precise role played by α Klotho co-receptor in the 1:2:1:1 FGF23-FGFR- α Klotho-HS signaling complex. It is important to note that in the absence of its co-receptor α Klotho, FGF23 has weak affinity for its cognate FGFRs, precluding formation of a stable FGF23-FGFR complex. The weak affinity of FGF23 for its cognate FGFRs can be attributed to specific amino acid substitutions in its receptor binding sites (see Extended Data Figure 3 in our previous Nature manuscript, PMID: 29342138). Furthermore, in an earlier publication (PMID: 17339340), we showed that FGF23's HS binding site diverges both compositionally and conformationally from those of paracrine FGFs, which dramatically diminish FGF23's HS binding affinity. These structural/biochemical deviances are the root cause of FGF23's hormonal mode of action and hence its dependency on α Klotho as the co-receptor. On the flip side, FGF23's poor HS and FGFR binding affinities render HS insufficient to stabilize an FGF23-FGFR complex and hence support FGF23 signaling. In other words, under physiological situation, a FGF23-FGFR1c^P-HS is extremely labile/transient and hence is unable to recruit a secondary FGFR^S. Indeed, our new cell-based experiments show that FGF23 signaling is strictly α Klotho dependent (**Extended Data Fig. 3b**). Even at supra pharmacological concentrations as high as 10 micromolar, FGF23 fails to activate FGFR1c signaling. However, when co-treated with soluble α Klotho co-receptor, as little as 10 nM concentration of FGF23 elicits a robust FGFR1c activation (**Extended Data Fig. 3b**).

The co-receptor mechanism of α Klotho in FGF23 signaling was illuminated by the crystal structure of FGF23-FGFR1c- α Klotho ternary complex (PMID: 29342138). In that paper, we showed that α Klotho serves as a molecular scaffold that simultaneously engages FGFR's D3 domain and FGF23's C-terminal tail. By tethering FGF23 and FGFR together, α Klotho enforces FGF23-FGFR proximity and complex stability. The stabilized binary FGF23-FGFR complex within the ternary FGF23-FGFR- α Klotho complex is now in position to recruit a second FGFR via FGFR^P-FGFR^S and FGF23-FGFR^S interactions albeit this still requires the assistance of HS as an additional co-receptor. Thus, we can conclude that α Klotho governs recruitment of the secondary FGFR to primary receptor by enforcing FGF23-FGFR1c^P complex stability. As to the distorted conformation of FGFR^S, it is dictated primarily by the FGFR^P-FGFR^S and to lesser extent FGF23-FGFR^S contacts. Having said so, it is fair to say that α Klotho and HS indirectly contribute to conformational distortion of FGFR^S by promoting FGFR^P-FGFR^S and FGF23-FGFR^S contacts.

In contrast to endocrine FGFs, paracrine FGFs have strong affinities for HS, and bind their cognate FGFRs with measurable affinity in the absence of HS. Consequently, unlike FGF hormones, paracrine FGFs rely only on HS as co-receptor to stably bind primary FGFR and subsequently recruit a secondary FGFR.

Critique #2-7-12: Critique #2: Second, although about 60% of the FGFR1 particles have the organization focused on in the figures, only about 23% or 33% do for FGFR3 and FGFR4 respectively. What are the other structures? Why were they ignored? This is not discussed sufficiently (at all) in the manuscript, and the basis for the choice of the structure described is not provided. This raises doubts about whether this is really a relevant structure. This issue is frequently seen in cryoEM, and the authors need both to justify their selection of this particle type and to make some comments about the other classes. **Critique #7:** The structure is based on 1,497,967 of 2,426,961 (62%) of particles for FGFR1, What are the other structures? Including the larger one for FGFR1 (middle). For FGFR3, the complex shown represents 23% or particles (291,540 of 1,272,758), and for FGFR4 it is about 33%. These numbers raise doubt about whether there are multiple forms of the complex, but the authors do not discuss this possibility. The question as to whether this is a unique (or even major) complex is also raised by the nature of the SEC traces in Extended Data Fig. 1 and Fig 3d. There seems to be a very mixed population that the authors have not characterized in terms of different stoichiometries. **Critique #12:** One thing that the manuscript lacks is a proper biophysical characterization of the 1:2:1:1 complex. The SEC-MALS studies in Fig. 3d are not very compelling or well presented. The authors should consider AUC or scattering (light or X-ray) approaches to better define the stoichiometry, which seems very variable based on SEC data and the EM particle distribution. There is always the worry that the complexes seen in the EM studies reflect the behavior of the proteins when frozen on grids. Also, it is clear many particles (the majority for FGFR3 and FGFR4) have structure distinct from what the authors focus on here. What are the other classes?

Response: We are fully cognizant of the unique power of cryo-EM in capturing multiple arrangements/stoichiometries of supramolecular complexes in vitrified samples as elegantly demonstrated in previous publications (PMID: 34210960; PMID: 26829225; PMID: 34718671; PMID: 35817871). Indeed, we initially considered that FGF23 cell surface signaling unit may entail larger arrangements/stoichiometry besides the 1:2:1:1 FGF23-FGFR- α Klotho-HS complex described in our manuscript. However, only the particles containing 1:2:1:1 FGF23-FGFR- α Klotho-HS complexes yielded high resolution structures. Notably, significant percentages of the particles, particularly in FGF23-FGFR3c- α Klotho-HS and FGF23-FGFR4- α Klotho-HS samples, were poorly resolved (likely due to denaturation at the air-water interface) and hence did not yield any meaningful structures. We would like to point out that protein denaturation is an inevitable problem during vitrification, significantly diminishing the percentage of useable particles for structure determination. Indeed, numerous complex structures have been determined using a fraction less than 20% of particles (PMID: 34819673; PMID: 34880492; PMID:

34381056).

Nevertheless, it is worth reporting that in all three cryo-EM maps, we saw an extra weak density associated with α Klotho component of the 1:2:1:1 quaternary complex (new **Extended Data Fig. 5a**). This density represents a second α Klotho that packs via its KL2 domain against the KL2 domain of the first α Klotho in a symmetric fashion (**Extended Data Fig. 5b**). Based on this observation, we envisioned that FGF23 signaling may entail higher order assemblies featuring a 1:2:2:1 or even a 2:4:2:2 stoichiometry, the latter resulting from symmetric juxtapositioning of two sets of 1:2:1:1 FGF23-FGFR1c- α Klotho-HS complexes (**Extended Data Fig. 5c**). Such higher order complexes seemed plausible particularly in light of the fact that reduced dimensionality in the cell membrane due to membrane tethering could promote weak interactions between α Klotho molecules.

To test this conjecture, we introduced two sets of triple mutations namely N782H/F784S/Y788R and D776A/N779A/Q780S separately into full-length transmembrane α Klotho and co-expressed these α Klotho variants and wild type α Klotho (as control) with FGFR1c in L6 cells. Cell lines were then treated with a fixed concentration of FGF23 with increasing duration of time (**Extended Data Fig. 5d,e**). In a parallel experiment, cells were exposed to varying concentrations of FGF23 (**Extended Data Fig. 5f**). FGFR1c activation/signaling was monitored by western blotting of total cell lysates with phosphospecific phosphorylated antibodies against FGFR and its downstream transducers. In both sets of experiments, the α Klotho variants exhibited comparable capacity as wild type α Klotho to promote FGF23 signaling as evident by similar levels of FGFR1c/PLC γ 1/FRS2 α /ERK phosphorylation (**Extended Data Fig. 5e,f**). These data argued against a physiological role for these putative higher order assemblies in FGF23 signaling. For this reason, we did not pursue these higher order complexes further and did not mention them in our first submission in order not to detract from the physiologically relevant 1:2:1:1 complex.

Critique #3-9-13-16: Critique #3: They also need to add some controls to the mutational studies, since similar interactions may also be relevant in FGF1 signaling for example. **Critique #9:** It is important that the authors show that the mutations tested in Figure 4 do not also impair FGFR activation by non-endocrine FGFs in cells. Although these interactions are not evident in previous crystal structures of FGF-driven dimers of FGFR ECDs, it is possible that those structures do not represent the cell surface situation, where tethering to the membrane could promote membrane-proximal interactions between FGFR ECDs (especially D3) that might be important for dimer stability in a signaling context. In other words, for this work, it is essential that the authors determine whether the interactions investigated here are specific to endocrine FGF. This is a crucial control that is missing and should be easy to include. **Critique #13:**

The complementation experiment in Figure 6 is a nice idea, but again it needs to be determined whether this is unique to the situation with FGF23 and Klotho. Controls with non-endocrine FGFs are needed – based on the same argument made in point 6. **Critique #16:** The final point about heterodimerization is puzzling given that Klotho plays no part in the FGFR/FGFR interactions. Can this same experiment be done with non-endocrine FGFs? Is heterodimerization selectivity different in the case of FGF23 compare with other FGFs?

Response: Like the reviewer, we ourselves have been intrigued by the asymmetric mode of receptor dimerization emerged from our three cryo-EM structures and wondered whether this model could be extended to paracrine FGFs. This seemed plausible given the fact that the dimer interface is highly conserved amongst seven principal FGFRs (**Extended Data Fig. 7c**) and that α Klotho does not directly participate in FGFR^S recruitment. To explore this possibility, we first tested the impacts of asymmetric FGFR^P-FGFR^S interface mutations in FGFR1c and FGFR2b on signaling by their cognate paracrine FGFs. These two FGFR isoforms were chosen because of their overlapping and unique ligand binding specificity/promiscuity profile. FGFR1c responds to paracrine FGF1 (a pan-FGFR ligand) and FGF4, whereas FGFR2b mediates the actions of FGF1, FGF3, FGF7, FGF10 and FGF22. For FGFR1c studies, we adopted the same five L6 cell lines namely FGFR1c^{WT}, FGFR1c^{E249A}, FGFR1c^{R254A}, FGFR1c^{I256A}, and FGFR1c^{Y280A} that were used to validate the asymmetric FGFR^P-FGFR^S dimer interface present in the cryo-EM structure of FGF23-FGFR1c- α Klotho-HS complex. For FGFR2b studies, we generated equivalent L6 cell lines expressing either wild type (FGFR2b^{WT}) or corresponding FGFR2b mutants (i.e., FGFR2b^{E250A}, FGFR2b^{R255A}, FGFR2b^{I257A}, and FGFR2b^{Y281A}). These cell lines were challenged with the same concentrations of cognate paracrine FGFs and receptor activation/signaling was assessed by immunoblotting analysis of total cell lysates with antibodies directed against phosphorylated FGFR, PLC γ 1 and FRS2 α (**For Review Only Figure R1**). Both FGFR1c and FGFR2b mutants were impaired in their capacity to undergo ligand-induced tyrosine trans auto-phosphorylation which was also reflected in reduced PLC γ 1 and FRS2 α phosphorylation (**For Review Only Figure R1**). These cell-based data implied that paracrine FGFs might signal via asymmetric 1:2:1 FGF-FGFR-HS dimers resembling that seen in cryo-EM structures of asymmetric 1:2:1:1 FGF23-FGFR- α Klotho-HS quaternary complexes.

Figure R1 (for review only)

Fig. R1. The Asymmetric dimer interface observed in FGF23-FGFR- α Klotho-HS cryo-EM structures is also essential for paracrine FGF signaling.

a, Immunoblots of whole cell extracts from untreated or FGF1/2/4 (1 nM) treated parental (untransfected) L6 cells and L6 cell lines stably expressing FGFR1c^{WT}, FGFR1c^{E249A}, FGFR1c^{R254A}, FGFR1c^{I256A}, or FGFR1c^{Y280A} probed with antibodies directed against phosphorylated FGFR, PLC γ 1 and FRS2 α . **b**, Immunoblots of whole cell extracts from untreated or FGF1/3/7/10/22 (1 nM for FGF1 and 2 nM for FGF3/7/10/22) treated parental (untransfected) L6 cell lines and L6 cell lines stably expressing FGFR2b^{WT}, FGFR2b^{E250A}, FGFR2b^{R255A}, FGFR2b^{I257A}, or FGFR2b^{Y281A} probed with FGFR2b isoform specific antibody (top) or phosphospecific antibodies as in panel **a**.

To more rigorously test the asymmetry of paracrine FGF-FGFR-HS signaling complexes, we next did receptor complementation and heterodimerization assays as was done for the endocrine FGF23. For FGFR1c complementation study, we adopted the same four cell lines (namely, L6-FGFR1c^{WT}, L6-FGFR1c^{APLBS}, L6-FGFR1c^{ASLBS} and FGFR1c^{ASLBS}+FGFR1c^{APLBS}) that were used to establish the asymmetry of

FGF23 quaternary signaling complex (**For Review only Figure R2a**). For FGFR2b complementation study, we constructed the equivalent receptor variants namely FGFR2b^{ΔSLBS} and FGFR2b^{ΔPLBS} which were then used to establish the corresponding cell lines (i.e., L6-FGFR2b^{ΔSLBS}, L6-FGFR2b^{ΔPLBS} and L6-FGFR2b^{ΔSLBS}+FGFR2b^{ΔPLBS}, **For Review only Figure R3a**). In response to FGF1 or FGF4 stimulation, cells expressing FGFR1c^{ΔPLBS} or FGFR1c^{ΔSLBS} alone failed to elicit any appreciable FGFR1c signaling whereas L6-FGFR1c^{ΔSLBS}+FGFR1c^{ΔPLBS} cells responded with robust FGFR activation and signaling (**For Review only Figure R2b,c**). Likewise, FGF1, FGF3, FGF7 and FGF10 each induced FGFR activation and signaling only in L6-FGFR2b^{ΔSLBS}+FGFR2b^{ΔPLBS} co-expressors (**For Review only Figure R3b-e**). These data demonstrate that paracrine FGFs transmit their signals via asymmetric 1:2:1 FGF-FGFR-HS complexes.

Figure R2 (for review only)

Fig. R2. Demonstration of the asymmetry of two distinct paracrine FGF-FGFR1c-HS signaling complexes via receptor complementation assay.

a, Schematic diagram showing that in response to FGF1/4 and HS, FGFR1c^{ΔSLBS} and

FGFR1c^{ΔPLBS} can complement each other and form 1:1:1:1 FGF-FGFR1c^{ΔSLBS}-FGFR1c^{ΔPLBS}-HS asymmetric signaling complexes. **b-c**, Demonstration of receptor complementation occurring between FGFR1c^{ΔSLBS} and FGFR1c^{ΔPLBS} via western blotting. L6 cell lines singly expressing FGFR1c^{WT}, FGFR1c^{ΔPLBS}, FGFR1c^{ΔSLBS}, or co-expressing FGFR1c^{ΔSLBS} with FGFR1c^{ΔPLBS} were treated with 1 nM FGF1 (**b**) or 1 nM FGF4 (**c**) for increasing time intervals and whole cell extracts were immunoblotted as indicated.

Figure R3 (for review only)

Fig. R3. Receptor complementation assays demonstrate the asymmetry of four distinct paracrine FGF-FGFR2b-HS signaling complexes.

a, Schematic diagram showing that in response to FGF1/3/7/10 and HS, FGFR2b^{ΔSLBS} and FGFR2b^{ΔPLBS} can complement each other and form 1:1:1:1 FGF-FGFR2b^{ΔSLBS}-FGFR2b^{ΔPLBS}-HS asymmetric signaling complexes. **b-e**, Demonstration of receptor

complementation between FGFR2b^{ΔSLBS} and FGFR2b^{ΔPLBS} via western blotting. L6 cell lines singly expressing FGFR2b^{WT}, FGFR2b^{ΔPLBS}, FGFR2b^{ΔSLBS}, or co-expressing FGFR2b^{ΔSLBS} with FGFR2b^{ΔPLBS} were treated with 1 nM FGF1 (b), 2 nM FGF3 (c), 2 nM FGF7 (d), or 2 nM FGF10 (e) for increasing time intervals, and total cell extracts were immunoblotted as indicated.

To provide another strict evidence for the asymmetry of paracrine FGF-FGFR signaling complexes, we next studied the possibility of FGFR heterodimerization by paracrine FGFs focusing on: i) FGFR1c-FGFR4 and FGFR1b-FGFR2b heterodimerizations by FGF1, ii) FGFR1b-FGFR2b heterodimerization by FGF10, and iii) FGFR2b-FGFR3b heterodimerization by FGF3. For FGFR1c-FGFR4 heterodimerization assay, we adopted the same cell lines that were used to demonstrate receptor heterodimerization by endocrine FGF23. For FGFR1b-FGFR2b heterodimerization assay, we generated L6 cell line expressing FGFR1b^{ΔSLBS} equivalent to FGFR2b^{ΔSLBS}. For FGF2b-FGFR3b heterodimerization by FGF3, we used wild type FGFR3b (FGFR3b^{WT}) as the ΔPLBS equivalent because this isoform naturally does not respond to FGF3. Reminiscent of FGF23, FGF1 failed to activate FGFR signaling in FGFR1c^{ΔPLBS} and FGFR4^{ΔSLBS} cell lines. However, strong FGFR signaling was seen in the FGFR4^{ΔSLBS}+FGFR1c^{ΔPLBS} co-expressing cell line in response to FGF1 stimulation (**For Review only Figure R4a**). Likewise, both FGF1 and FGF10 induced robust FGFR activation/signaling only in FGFR1b^{ΔSLBS}+FGFR2b^{ΔPLBS} co-expressing cell line but not in L6-FGFR1b^{ΔSLBS} and L6-FGFR2b^{ΔPLBS} cell lines (**For Review only Figure R4b,c**). Lastly, FGF3 provoked signaling in FGFR2b^{ΔSLBS}+FGFR3b^{WT} co-expressing cell line but not in L6-FGFR2b^{ΔSLBS} and L6-FGFR3b^{WT} cells (**For Review only Figure R4d**). Taken together, these comprehensive cell-based data imply that paracrine FGF1, FGF2, FGF3, FGF4, FGF7, FGF10 and FGF22 all signal via asymmetric dimers of their respective cognate FGFR isoforms.

Figure R4 (for review only)

Fig. R4. Receptor heterodimerization assays validate the asymmetry of the paracrine FGF-FGFR-HS signaling complexes.

a, Left: L6 cell lines expressing FGFR4^{WT} or FGFR4^{ΔSLBS} singly, or co-expressing FGFR4^{ΔSLBS} with FGFR1c^{ΔPLBS} were treated with 1 nM FGF1 for increasing periods of time and cell extracts were immunoblotted with indicated FGFR isoform-specific and phosphospecific antibodies. Right: Schematic diagram showing that FGF1, HS, and FGFR4^{ΔSLBS} (serving as primary receptor) form a stable complex which subsequently recruits FGFR1c^{ΔPLBS} as secondary receptor. **b-c**, Left: L6 cell lines singly expressing FGFR1b^{WT} or FGFR1b^{ΔSLBS}, or co-expressing FGFR1b^{ΔSLBS}+FGFR2b^{ΔPLBS} were treated with 1 nM FGF1 (**b**) or 2 nM FGF10 (**c**) for increasing time intervals and cell extracts were immunoblotted as in panel **a**. Right: Schematic diagram showing that 1 nM FGF1 (**b**) or 2 nM FGF10 (**c**), HS, and FGFR1b^{ΔSLBS} (serving as primary receptor) form stable complexes which subsequently recruit FGFR2b^{ΔPLBS} as secondary receptor. **d**, Left: L6 cell lines expressing FGFR3b^{WT} alone or co-expressing it with FGFR2b^{ΔSLBS} were treated with 2 nM FGF3 for increasing time intervals and cell extracts were immunoblotted as in panel **a**. Right: Schematic diagram showing that in the presence of HS, FGF3 and FGFR2b^{ΔSLBS} (serving as primary receptor) form a stable complex and subsequently recruit FGFR3b^{WT} as secondary receptor.

(Redacted) Based on these comprehensive and stringent cell-based data, we conclude that the asymmetric receptor dimerization is a shared mechanism in signal transmission by all FGFs (endocrine and paracrine). We believe that our discovery of asymmetric receptor dimerization by paracrine FGFs goes beyond the scope of our current study on endocrine FGFs and deserves to be showcased in a separate publication dedicated to paracrine FGFs.

Redacted

Reviewer #4: The introduction seems very long.

Response: Introduction has been made more concise by focusing mainly on FGF hormones. In doing so, we reduced the word size to 750 which also helped with its conformity to Nature guidelines as to manuscript length.

Reviewer #5: The phrasing ‘hardcore paracrine-acting ligands’ on line 57 seems out of place.

Response: The wording “hardcore” has been removed.

Critique #6: The discussion on line 184 about the FGF23-FGFR1^P interaction appearing ‘tighter’ here than in the prior crystal structure is too speculative, and should be removed. Comparing H-bond lengths in structures determined by two different methods does not make a compelling argument for interaction strength. This needs to be assessed experimentally. If the authors wish to argue that Klotho makes FGFR^P more rigid and this aids recruitment of the FGFRs, this needs to be assessed experimentally.

Response: We believe we have not conveyed our point clearly. We simply intended to state that by simultaneously engaging FGF23 and FGFR1c^P, HS further enhances FGF23-FGFR1c^P proximity and stability within the ternary complex component of the quaternary complex. Nevertheless, we understand reviewer’s objection, and have removed the statements on the comparison of the strength of FGF23-FGFR1c between crystal and cryo-EM structures.

Critique #8: The detailed discussion of the FGFR^P-FGFR^S interface on page 7 does not help in understanding.

Response: We have removed the detailed description of the FGFR^P-FGFR^S

interface and instead list the interactions in an **Extended Data Table 2** as part of supplementary materials.

Critique #10: In the studies of FGFR mutants, cell surface receptor expression needs to be shown by flow cytometry or immunofluorescence. The total FGFR blots for FGFR1 and 4 show different distributions of upper and lower bands, raising concerns about integrity that should be controlled for in a complete study.

Response: The reviewer raises a valid concern. Indeed, mutations in the ectodomains of RTKs can have unintended consequences on receptor glycosylation and trafficking, diminishing their cell surface expression. In case of FGFRs, such problems can be detected in immunoblots of cell lysates probed with anti-FGFR antibodies. Specifically, wild type FGFRs appear as a major diffuse band of ~140 and a minor sharper band of ~130 kDa. The upper band represents the fully glycosylated mature FGFR, containing complex sugars that has passed the ER quality control and has been successfully routed to the cell surface. On the other hand, the faster migrating lower band is an incompletely processed high mannose form that is trapped in ER and hence is inaccessible to FGF stimulation. Accordingly, only the fully glycosylated receptor (cell surface resident) binds FGF and undergoes tyrosine phosphorylation, showing up as a single band in western blots directed against phosphorylated A-loop tyrosines. Mutations impacting receptor maturation manifest in an increase in proportion of the faster migrating high mannose containing receptor band. Notably, the mannose rich form is sensitive to endoglycosidase H (Endo H) which cleaves the bond between two N-acetylglucosamine (GlcNAc) subunits directly proximal to the asparagine residue, leaving one N-acetylglucosamine residue on the asparagine. However, the presence of complex sugars protects the fully glycosylated form against Endo H. Consequently, Endo H sensitivity serves as an excellent tool to assess intracellular trafficking through the secretory pathway and hence cell surface expression of various FGFR mutants. Specifically, cell surface receptor expression can be presented as a ratio of Endo H-resistant fraction over the total receptor expression determined by treating the receptor with PNGase F. This enzyme is an amidase that hydrolyzes the bond between the inner most GlcNAc and asparagine irrespective of complex sugar content thus completely stripping the FGFR from all its N-linked sugars. As seen in **Figure 3b-d**, apart from the FGFR1c^{I256A} and its corresponding FGFR4^{I250A}, all the remaining mutants have comparable ratios of Endo H resistant to total FGFR protein similar to wild type FGFRs. Hence, only FGFR1c^{I256A} and corresponding FGFR4^{I250A} had diminished cell surface expression. However, given the fact that the glycosylation and cell surface expression of the corresponding FGFR3c^{I254A} is unaffected, it is unlikely that the impaired signaling by FGFR1c^{I256A} and FGFR4^{I250A} are solely due to reduced cell surface expression.

Critique #11: Why is FRS2 phosphorylation so much reduced for FGFR3 I254A? Also, why is ERK signaling not reduced more in the FGFR1 R254A case?

Response: The pattern of contacts at the D3-D3 interface in our quaternary complexes is typical of cell surface receptor complexes where hydrophobic and hydrogen bonding interactions dominate the core and the edges of interface, respectively. Accordingly, mutations that abrogate the core hydrophobic contacts cause greater loss-of-function effect than mutations that disrupt the peripheral hydrogen bonds. Indeed, we see larger reductions in FRS2 α phosphorylation in all three FGFR isoforms harboring mutations that disrupt the core hydrophobic contact such as the FGFR3 I254A.

PLC γ 1 and FRS2 α are direct downstream substrates of FGFRs, and therefore their tyrosine phosphorylation levels provide a better readout for FGFR activity. In contrast, ERKs are not direct substrates of FGFRs but are phosphorylated by upstream Raf and MEK1/2 kinases such that ERK phosphorylation represents an amplified response. As a result, differences in ERK phosphorylation levels are less pronounced between wild type and mutated receptors.

Critique #14: I have the sense that the figures could be better designed. Figures 1 and 2 seem to show essentially the same thing, and Figure 3a and b the same thing again. These could all be consolidated in one clearer figure.

Response: Figure 1 shows the experimental electron density maps of three complexes at the stated contour levels which we feel strongly about to show because it allows the reviewers/reader to appreciate/analyze the quality of our cryo-EM maps. Original Figure 2, on the other hand, shows surface presentation of the final refined models and hence does not disclose any information as to the quality of our cryo-EM maps. Nevertheless, per reviewer's recommendation, we have eliminated original Figure 2. Likewise, we have improved original Figure 3 (now revised Figure 2) by: 1) eliminating the surface presentation from panel a and b to take care of redundancy; 2) moving panel d into the **Extended Data Fig. 1e** and instead bringing in new cell-based data including immunoblotting and PLA data validating the importance of FGFR-HS binding interactions to the formation of the quaternary FGF23-FGFR- α Klotho-HS complex (as requested by Reviewer 3). The revised manuscript has now a total of 5 main display items.

Critique #15: There is no quantitation of any of the signaling work. Just a statement that the experiments were performed three times.

Response: Per reviewer's request, we have now quantitated the western-blot images using image J software and present the data as bar diagram including statistical analysis using GraphPad Prism in the related figures.

Referee #2

General comments: The authors previously solved the structure of the 1:1:1 ternary complex composed of fibroblast growth factor-23 (FGF23), fibroblast growth factor receptor (FGFR), and alphaKlotho. Because dimerization of FGFRs facilitated by heparan sulfate (HS) is required to activate FGFR tyrosine kinase, it has been postulated that formation of symmetric 2:2:2:2 complexes composed of FGF23, FGFR, alphaKlotho, and HS would be generated upon activation of the canonical intracellular FGF signaling pathway. Contrary to the expectation, the present study has revealed an unexpected structure of the active cell surface complex using cryo-EM, which is composed of asymmetric 1:2:1:1 FGF23-FGFR-alphaKlotho-HS. This is a unique ligand-receptor structure unknown so far. In addition, the authors introduced multiple mutations in FGF23 and FGFRs that were predicted to abolish formation of the quaternary complex and verified that such mutant proteins were unable to signal in cell culture experiments. The data presented are comprehensive and compelling, supporting the authors' conclusion. This reviewer has no major concerns, but the authors should address the following specific comments.

Response: We are grateful to the reviewer for his/her appreciation of the novelty of our structures and the quality and thoroughness of the supporting data.

Critique #1: In Figure 4, was the amount of mutant FGFRs expressed on the cell surface equivalent to that of wild-type FGFRs? Because mutant membrane proteins often fail to be transported to the cell surface, one may argue that the reduced ability of the mutant FGFRs to activate the FGF signaling may be due to their failure to appear on the cell-surface. Cell-surface biotinylation assay may help.

Response: Reviewer 1 expressed similar concern (i.e., **Critique #10**) regarding potential impacts of mutations on FGFR glycosylation/maturation leading to reductions in cell surface expression of certain mutants relative to wild type receptors. Please refer to our response to reviewer 1.

Critique #2: Line 205 (Page 7): "All three subdomains (i.e. D2, D2-D2 linker, and D3)" should be "All three subdomains (i.e. D2, D2-D3 linker, and D3)".

Response: This typo has been fixed.

Critique #3: What was the final concentration of FGF23 and Klotho in Figure 4b-d?

Response: The final concentrations of FGF23 and α Klotho used for the cell-based experiments shown in Figure 4b-d (now revised **Fig. 3e-g**) were 20 nM.

This information has now been explicitly stated in the legend of this figure.

Critique #4: What was the final concentration of the FGF23 wild-type/mutant proteins in Figure 5b-d? How did the authors confirm that the same amounts of FGF23 wild-type/mutant proteins were applied to the cells?

Response: The final concentrations of FGF23 wild-type/mutant proteins applied to cells in experiments shown in Figure 4b-d (previously 5b-d) have been provided within the display items. Samples of the purified FGF23^{WT} and mutants were analyzed by SDS-PAGE to confirm their similar purity and quantity (**Extended Data Fig. 1d**).

Referee #3

General comments: FGF19, FGF21, and FGF23 belong to a family of endocrine FGFs, and the FGF23-FGFR- α Klotho axis plays an important role in regulating the homeostasis of vitamin D and phosphate. The crystal structure of 1:1:1 FGF23-FGFR- α Klotho has been determined before, showing how α Klotho acts as a scaffold that simultaneously recruits FGF23 and FGFR. Here, Chen et. al presents a series of cryo-EM structures of FGF23-FGFR- α Klotho in complex with heparan sulfate (HS), including FGF23-FGFR1- α Klotho-HS, FGF23-FGFR3- α Klotho-HS, and FGF23-FGFR4- α Klotho-HS. These new cryo-EM structures reveal an unexpected 1:2:1:1 stoichiometry in the formation of FGF23-FGFR4- α Klotho-HS complex, and uncover several new interfaces between FGF23 and FGFR-s and between FGFR-s and FGFR-p that are critical for the complex formation and receptor activation. The authors also propose that the N-terminal region of FGF23 might be important in the assembly of 1:2:1:1 complex. The functional significance of some of the newly discovered interfaces were validated by mutagenesis and cell-based experiment. The authors also design a nice complementation assay to prove that the 1:2:1:1 asymmetric complex is a functionally relevant conformation and is required for receptor activation. Overall, these structural works are very interesting. By revealing the structures of FGF23-FGFR- α Klotho in the HS bound, active state, this work significantly advances our understanding of the mechanism underlying the FGF23 induced activation of FGFR.

Response: We thank the reviewer for his/her positive remarks as to the novelty, experimental design and significance of our work.

However, there are a few major issues need to be addressed before this work could be published. My specific points are:

Critique #1: In all the three cryo-EM maps reported from this work, it is clear that there are large unmodelled densities presented. See the attached image. Such large density certainly should not be ignored. Although being resolved at low resolution, the shape and size of the density is highly similar to a α Klotho. See the attached image: the model of a α Klotho shown in green is fitted into the unmodelled cryo-EM density. This suggest that a second α Klotho may be involved in the formation of FGF23-FGFR4- α Klotho-HS complex, leading to a stoichiometry of 1:2:2:1, rather than the 1:2:1:1 proposed by the authors. The flexible binding of a second α Klotho may further stabilize the interact between FGFR-s and FGFR-p or between FGFR-s and FGF23, so it might be functionally important. Alternatively, this second α Klotho could potentially lead to the formation of higher ordered assembly of FGFR by recruiting a second set of FGF23 and FGFR-p. Therefore, the authors need to improve the quality of this unmodeled density to reveal its identity by collecting more cryo-EM data and/or performing local 3D classification/refinement. If this is indeed another

α Klotho, the functional significance of this second α Klotho needs to be tested by mutagenesis and cell-based experiments, and its functional role needs to be discussed in the paper.

Response: We appreciate reviewer's deep analysis of our maps/models. To address this issue, we have added a new **Extended Data Fig. 5** to the revised manuscript. We have been aware of the extra weak density associated with the quaternary complex in our cryo-EM maps (**Extended Data Fig. 5a**). As inferred by the reviewer, this density represents a second α Klotho that packs via its KL2 domain against the corresponding KL2 domain of α Klotho within the quaternary complex in a symmetric fashion (**Extended Data Fig. 5b**). Based on this observation, we had initially contemplated the possibility that FGF23 signaling may entail higher order assemblies featuring a 1:2:2:1 or 2:4:2:2 stoichiometry, the latter arising from symmetric apposition of two sets of 1:2:1:1 FGF23-FGFR1c- α Klotho-HS quaternary complexes (**Extended Data Fig. 5c**). Such higher order assemblies seemed plausible particularly on cell membrane where reduced dimensionality would promote interactions between α Klotho molecules.

To test this conjecture, we generated full length wild type α Klotho and two mutated α Klotho constructs harboring either a N782H/F784S/Y788R or D776A/N779A/Q780S triple mutation designed to disrupt the observed α Klotho- α Klotho contacts (**Extended Data Fig. 5b**). Each construct was stably co-expressed with FGFR1c in L6 cell lines (**Extended Data Fig. 5d**). These cell lines were treated with a fixed concentration of FGF23 for increasing duration of time (**Extended Data Fig. 5e**). In a parallel experiment, cells were exposed to varying concentrations of FGF23 (**Extended Data Fig. 5f**). FGFR1c activation/signaling was monitored by western blotting of total cell lysates with phosphospecific antibodies against FGFR and its downstream signal transducers. In both sets of experiments, the α Klotho variants possessed comparable capacity as wild type α Klotho to promote FGF23 signaling as evident by similar levels of FGFR1c/PLC γ 1/FRS2 α /ERK phosphorylation (**Extended Data Fig. 5e,f**). These data argued against a physiological role for these putative higher order assemblies in FGF23 signaling. For this reason, we did not further pursue these higher order complexes and decided not to mention them in our first submission in order not to detract from the physiologically relevant 1:2:1:1 quaternary complex.

Critique #2: The authors propose that the N-terminal loop of FGF23 engages FGFR-s, thereby contributing to the interaction between FGF23 and FGFR-s. However, I think the N-terminal loop of FGF23 was not resolved in the cryo-EM map at all. Even at low contour level, I can't observe any cryo-EM density corresponding to the N-terminal loop of FGF23. See the attached image. Therefore, the modelling of N-terminal loop of FGF23 shown in Fig 5 is very

questionable. The functional role of the N-terminal loop of FGF23 also can't be strongly supported by the functional experiment. The authors need to improve the cryo-EM density for this region. Again, they could collect more cryo-EM data, perform more extensive global/local 3D classification.

Response: We totally understand reviewer's criticisms regarding the poor density of FGF23 N-terminus and the modest loss in activity of our N-terminally truncated FGF23. We did conduct exhaustive global/local 3D refinements to improve the electron density for FGF23's N-terminus but these attempts were not fruitful. We believe that FGF23 N-terminus interacts promiscuously with hydrophobic D3 groove thus adopting multiple flexible conformations that cannot be resolved in the cryo-EM density. We admitted this fact by stating "*Notably, in all three structures, electron densities for FGF23 N-terminus are poorly defined implying that interactions of FGF23 N-terminus with FGFR^S D3 are flexible*" in our initial submission. Nevertheless, we believe the "flexible/degenerate" binding of FGF23 N-terminus does contribute to recruitment of the secondary FGFR^S chain to the ternary complex and hence receptor dimerization/signaling. Support for this conjecture can be drawn from published data on the highly related FGF19 and FGF21 hormones. Notably, N-terminally truncated FGF21 molecules behave as partial agonists (PMID: 19059246) which is expected based on our current cryo-EM structures. Furthermore, N-terminal swapping experiments involving FGF19 and FGF21 show that N-termini of FGF hormones plays essential role in FGFR selectivity/signaling potential (PMID: 19117008; PMID: 22248288; PMID: 32061104).

Considering these published data, we compared receptor dimerization/activation efficacies of wild type (FGF23^{WT}) and N-terminally truncated FGF23 (FGF23^{ΔNT}) ligands using a wide range of protein concentration in L6-FGFR1c cell line via immunoblotting analysis and PLA, and generated dose-response curves (revised **Figure 4b-d**). At all concentrations, FGF23^{ΔNT} elicited reduced activity relative to FGF23^{WT} and could not reach the maximal activity (E_{max}) exerted by FGF23^{WT} regardless of the amount applied. Moreover, when mixed with FGF23^{WT}, FGF23^{ΔNT} acted as competitive antagonist, producing a net decrease in FGFR1c activation. These cell-based data corroborate the importance of FGF23 N-terminus in promoting formation of quaternary signaling complex.

We also employed molecular dynamics (MD) simulation to further interrogate the role of FGF23 N-terminus in the formation of the FGF23-FGFR1c- α Klotho-HS complex. A 300 ns all-atom MD simulation of the FGF23-FGFR1c- α Klotho-HS complex revealed concerted motions of the N-terminal tail of FGF23 and three stranded β C: β F: β G sheet in FGFR1c^S D3 domain (**Extended Data Fig. 6f**). Notably, RMSD of N-terminal residues of FGF23 stabilized around 4 Å after 120 ns (**Extended Data Fig. 6g**). Importantly,

residues at the distal and proximal ends of FGF23 N-terminus exhibited largest and smallest Root Mean Square Fluctuation (RMSF), respectively, mirroring their respective cryo-EM electron densities (compare **Extended Data Fig. 6e and h**). The distances of selected hydrophobic residue pairs namely Y25–L342^S, L31–A259^S, and L32–I256^S fluctuated around 5 Å indicative of formation of hydrophobic contacts between these residue pairs (**Extended Data Fig. 6i**). Likewise, the pairwise distance for S29–R254^S fluctuated around 4 Å after 120 ns, indicative of hydrogen bonding between side chains of this residue pair. Taken together with our cryo-EM and cell-based data, these MD simulation data show that N-terminal tail of FGF23 engages D3 of FGFR1c^S via hydrogen bonding and hydrophobic interactions thus contributing to the overall stability/functionality of the FGF23-FGFR1c- α Klotho-HS quaternary signal transduction complex.

Critiques #3 and #4: **Critique #3:** The author didn't perform any mutagenesis to validate the binding mode of HS at FGFRs. It is, no doubt, that such validation is required as the HS binding plays a critical role in the receptor activation by gluing the two FGFRs together. Therefore, the authors need to introduce some mutations to the interfaces between FGFR-p and HS and between FGFR-s and HS, and test the effect of these mutation on the formation of quaternary complex by using the SEC experiment shown in Fig. 3d as well as receptor activation. **Critique #4:** The authors tested the importance of site 2 interface between FGFR-s and FGFR-p by using mutagenesis. However, it is unclear to me why they didn't test the importance of site 1 interface between FGFR-s and FGFR-p using the same method. Such validation is required when the complex structure is determined for the first time.

Response: We have grouped these two critiques together because we addressed them using a similar cell-based assay. New experimental data were generated and have been incorporated into revised **Figure 2c-e** and **Figure 3b,e**. To interrogate the significance of the FGFR-HS interactions to the formation of the quaternary FGF23-FGFR- α Klotho-HS complex, we introduced two double mutations (i.e., K175Q/K177Q and K207Q/R209Q) separately or in combination (K175Q/K177Q/K207Q/R209Q) into full length FGFR1c. Based on the cryo-EM structure (i.e., revised **Figure 2b**), these mutations are predicted to abolish interactions of FGFR1c with HS and thus impair FGF23 signaling. To ascertain the importance of site 1 portion of the asymmetric receptor-receptor interface, we introduced A170D/A171D/S219D triple mutation into full length FGFR1c (FGFR1c^{A170D/A171D/S219D}). Based on the cryo-EM structure (i.e., revised **Figure 3a**), this triple mutation is predicted to abolish hydrophobic and hydrogen bonding interactions between FGFR^P and FGFR^S thus impairing formation of the quaternary FGF23-FGFR1c- α Klotho-HS cell surface signaling unit.

L6 cell lines stably expressing each FGFR1c variant were generated (**Figures 2c and 3b**) and co-treated with recombinant FGF23 and soluble α Klotho. FGFR activation/signaling was measured by blotting of cell lysates with phosphospecific antibodies against FGFR1c and its downstream signal transducers (i.e., PLC γ 1/FRS2 α /ERK) (**Figures 2d, 3e**). FGFR1c^{K175Q/K177Q} and FGFR1c^{K207Q/R209Q} double mutants each incurred significant losses in their ability to mediate FGF23 signaling whereas FGFR1c^{K175Q/K177Q/K207Q/R209Q} quadruple mutant was totally non-responsive (**Figure 2d**). Likewise, co-stimulation of an L6 cell line expressing FGFR1c^{A170D/A171D/S219D} with FGF23 and α Klotho failed to elicit any FGF signaling as evident by lack of phosphorylation of FGFR1c and its downstream signaling molecules (**Figure 3e**). FGFR activation/signaling data were supported by proximity ligation assays (PLA) (**Figure 2e**). Specifically, co-treatment with FGF23^{WT} and α Klotho led to appearance of copious and intense punctate fluorescent signals on the surface of L6-FGFR1c^{WT} cell line. In contrast, there were far fewer fluorescent signal on the surface of cell lines expressing FGFR1c^{K175Q/K177Q}, FGFR1c^{K207Q/R209Q}, and FGFR1c^{K175Q/K177Q/K207Q/R209Q} in response to FGF23^{WT} and α Klotho co-treatment (**Figure 2e**). These cell-based data confirm the importance of both FGFR1c-HS and FGFR1c-FGFR1c contacts at site 1 in supporting formation of FGF23-FGFR1c- α Klotho-HS quaternary cell surface signaling complexes.

Critique #5: The authors designed a few FGF23 mutants, and compare their biological activities with FGF23 WT. The authors need to show the purity and quantity of purified FGF23 WT and mutants used in the cell-based experiments by SDS-PAGE. This is important because they need to exclude the possibility that lower activities of FGF23 mutants are due to the impurity of the protein.

Response: Reviewer 2 expressed the same concern in his/her **Critique #4**. Please refer to our response to reviewer 2.

Critique #6: As shown in Fig 4, the authors modelled a Cu ion between FGFR-s and FGFR-p. The rationale of such ion assignment is unclear to me. Other divalent metal ions, such as zinc, could also potentially bind to this site. The authors need to provide more evidence to support such ion assignment and modelling.

Response: We assigned that density sandwiched between primary and secondary FGFR D3 domains as Cu²⁺ based on a previous publication implicating specific Cu²⁺ interactions with the extracellular domains of FGFRs (Patstone *et al.*; PMID: 8631930). Cell culture media (DMEM and DME/F12) used to grow HEK293S GnTI⁻ cells that secrete minimally glycosylated FGFR ectodomains are the likely source of Cu²⁺ ions. A note has been added to the

legend for **Figure 3a** (previously Figure 4) in the revised manuscript stating our rationale for Cu²⁺ assignment.

Critique #7: There is no cryo-EM density to support the modelling of a long loop of FGF23 (residues 172 – 182). As this loop doesn't contact either FGFR or α Klotho, it probably becomes very flexible. This loop needs to be removed from the model to prevent any misleading.

Response: We appreciate reviewer's careful analysis of our maps/models. Indeed, residues 172–182 of FGF23, which links FGF23's trefoil core and its distal α Klotho binding site, lacks any discernable electron density. As interpreted by the reviewer, this region does not interact with either α Klotho or FGFR and is disordered/flexible. Notably, this region harbors the regulatory subtilisin-like proprotein convertase (SPC) site, ¹⁷⁶RHT¹⁷⁸R¹⁷⁹/^S¹⁸⁰AE¹⁸², which includes a furin type protease cleavage site (R179), an O-glycosylation site (T178) and a serine phosphorylation site (S180). Three enzymes namely GalNAc-T3(N-acetylgalactosaminyltransferase3), Fam20C (the family with sequence similarity 20, member C), and a yet to be discovered furin type protease converge on this site to regulate FGF23's proteolytic processing/activity. The high flexibility of this region would likely facilitate access by these enzymes. Per reviewer's recommendation, we removed this region from all three structural models and present it by dashed line to indicate its disordering/flexibility. We have added a note to the legend of **Extended Data Fig. 3a** regarding the disordered nature of this region.

Critique #8: It would be useful to prepare a supplementary figure to compare the structure of FGF23-FGFR- α Klotho-HS with that of FGF1-FGFR. This would show the structural differences of FGFR induced by different families of FGFs, and help the readers to better understand the functional role of Klotho in the activation of FGFR.

Response: Indeed, we have been attentive to ligand-induced differences in FGFR conformation as we gather more structural data on FGF-FGFR complexes. In addition to our current 3 cryo-EM structures of FGF23-FGFR- α Klotho-HS quaternary complexes, we had previously obtained a total of 11 crystal structures featuring distinct FGF-FGFR complexes including: i) nine distinct paracrine FGF-FGFR binary complexes (i.e., HS free), ii) one dimeric (HS-bound; active) 2:2:2 paracrine FGF2-FGFR1c-HS ternary complex, and iii) one endocrine FGF23-FGFR1c-Klotho ternary complex (HS free; inactive). By comparing these crystal structures, which contain members from 5 different FGF subfamilies namely FGF1, FGF7, FGF8, FGF9 and FGF19, we identified ligand-induced differences in FGFR conformation that are unique to a given FGF subfamily (PMID: 22057274; PMID: 12591959; PMID: 16384934; PMID:

28757146; PMID: 29342138). We postulated that such differences in conformation and stability of extracellular FGF-FGFR complexes may serve as a mechanism in regulating divergent functions of 18 FGFs that signal via seven FGFR isoforms (PMID: 19247306; PMID: 23403721).

We indeed included a structural superimposition of FGF23-FGFR1c component from FGF23-FGFR1c- α Klotho ternary complex (HS free, i.e., inactive) with FGF9-FGFR1c complex in our previous Nature paper (PMID: 29342138; Extended Data Fig. 3). Specifically, we showed that the endocrine FGF23-FGFR1c adopts a similar conformation as the paracrine FGF9-FGFR1c complex, and hence by extension as the FGF1-FGFR1c complex. Moreover, as shown in **Extended Data Fig. 3c** of our current paper, the overall conformation of FGF23-FGFR1c^P- α Klotho component of the quaternary complex cryo-EM structures is essentially identical to the crystal structure of ternary FGF23-FGFR1c- α Klotho complex. Therefore, we can conclude that FGF23-FGFR1c^P in our quaternary cryo-EM structures adopts the same conformation as the crystal structures of binary FGF9-FGFR1c or FGF1-FGFR1c complex.

Despite its high structural similarity to binary FGF9-FGFR1c or FGF1-FGFR1c complexes, a binary FGF23-FGFR1c complex is untenable in the absence of its α Klotho co-receptor because of FGF23's weak affinity to its cognate FGFRs. This is attributable to specific amino acid substitutions in FGF23's receptor binding site (see Extended Data Figure 3 in our previous Nature manuscript, PMID: 29342138). Furthermore, as demonstrated in our 2007 paper (PMID: 17339340), the HS binding site composition of FGF23 diverges completely from those of paracrine FGFs, thus dramatically diminishing FGF23's binding affinity. These differences are the root cause of hormonal mode of action of FGF23 and its dependency on α Klotho as a co-receptor. Importantly, FGF23's poor HS and FGFR binding affinities render HS insufficient to stabilize an FGF23-FGFR complex and support FGF23 signaling. In other words, under physiological situation, a FGF23-FGFR1c^P-HS complex is extremely labile/short-lived and hence is unable to recruit a secondary FGFR^S via FGF23-FGFR^S and FGFR^P-FGFR^S interactions. Indeed, the new added cell-based experiments show that FGF23 signaling is strictly α Klotho dependent. Even at supra pharmacological concentrations as high as 10 micromolar, FGF23 fails to activate FGFR1c signaling in L6-FGFR1c, while as low as 10 nM FGF23 elicit robust FGFR1c activation when co-treated with α Klotho (**Extended Data Fig. 3b**).

The co-receptor role of α Klotho in promoting FGF23 signaling was elucidated in our previous Nature paper on the crystal structure of ternary complex (PMID: 29342138). In that paper, we showed that α Klotho serves as a molecular scaffold that concomitantly engages FGFR's D3 domain and FGF23's C-terminal tail. By tethering FGF23 and FGFR together, α Klotho enforces FGF23-FGFR proximity thus imparting complex stability. The

stabilized FGF23-FGFR within the ternary FGF23-FGFR- α Klotho complex is now bestowed with the ability to recruit a second FGFR chain via FGFR^P-FGFR^S and FGF23-FGFR^S interaction with the assistance of HS co-receptor. Thus, α klotho indirectly promotes recruitment of a secondary FGFR to primary receptor by conferring stability to the FGF23-FGFR1c^P complex.

Reviewer Reports on the First Revision:

Referees' comments:

Referee #1 (Remarks to the Author):

Chen et al. have put significant effort in to revising their manuscript on FGF23/FGFR/aKlotho/HS complex structures, and have addressed most of the detailed concerns raised in my previous review. The asymmetric 1:2:1:1 complexes seen (from 1:1:1:1 mixtures) are quite interesting, and suggest a model in which Klotho serves to stabilize interaction of FGF23 with FGFR(P) at the membrane – allowing the resulting FGF23/FGFR(P) complex (with HS) to recruit a second FGFR (FGFR(S)) that makes no contacts at all with Klotho. The FGF23/FGFR(P)/Klotho structure here is essentially the same as that described in an earlier Nature paper by this group, and the main purpose of the Klotho is to recruit FGF23 to the FGFR(P) molecule. In the previous version of the manuscript, I was looking for reasons why the asymmetric FGFR dimer seen here would be specific to the situation with endocrine FGFs and Klotho – and suggested that the authors determine whether the same FGFR mutations that impair FGF23 – designed based on the asymmetry – also affect paracrine FGF signaling. In the rebuttal, the authors now describe experiments revealing that paracrine FGFs do use asymmetric FGFR dimers. This is important.

I am concerned that publishing the present paper without discussing in some detail the finding in the rebuttal that paracrine FGFs also stabilize asymmetric FGFR dimers will mislead the field. Other readers will – as I did in the first version of the manuscript – presume that the asymmetric FGFR dimers are somehow Klotho-specific and/or endocrine FGF specific. As far as I am aware, only 2:2 symmetric dimers were previously seen for paracrine FGFs (with HS), and the 1:2 (FGF23:FGFR) dimers (plus an HS and a Klotho) appear specific to the endocrine system if one reads this paper as it stands. But, the authors clearly state in their rebuttal that there is no such distinction to be made between endocrine and paracrine FGFs on that basis. That's a very important point.

It looks like Klotho has little or nothing to do with FGFR dimerization or the asymmetry (but the paper still implies that it does). Klotho just brings the FGF23 into position, making up for weaker HS binding in effect (and providing cell specificity). There is a very interesting distinction to be made based on HS and Klotho both acting as co-receptors by interacting with ligand and receptor to induce the same asymmetric dimers, but in different ways.

By hiding the fact that the paracrine FGFs (with HS) induce the same asymmetric FGFR dimers as does FGF23 (with HS + Klotho), I think the paper as written is confusing and should not be published in Nature. On the other hand, if the authors brought in the asymmetric dimers with paracrine FGFs into this paper, and focused on the differences in how HS alone (paracrine case) and HS + Klotho (endocrine case) promote the same FGFR dimer complex, that could be a highly valuable contribution.

I also think that the authors still spend too much time/text going through details of the interactions, which limits accessibility of the manuscript for those who are not structurally inclined, and I think

detracts from the bigger picture. Much of this could be removed to focus on the commonality of the asymmetric FGFR dimers seen with endocrine and paracrine FGFs – achieved in slightly different ways and ultimately showcasing the role of Klotho in targeting FGF23 to particular FGFRs in specific cells (that have the Klotho). Indeed, the current title almost implies that this is what the present paper is about.

Referee #3 (Remarks to the Author):

The authors have put substantial effort to address my concerns that were raised in my initial review. The manuscript has been improved remarkably with new mutagenesis results and newly prepared figures. The role of the second klotho is still mysterious that may require further studies in the future. Overall, this is a nice piece of work, and is of high quality. I strongly support its publication at Nature.

Author Rebuttals to First Revision:

Response to Reviewers

Title: FGF hormones Signal via Dual Co-Receptor Dependent Asymmetric FGF Receptor Dimers (Revised title: *Structural Basis for FGF Hormone Signaling*)

Manuscript: 2022-09-14066B

Referee #1 (Remarks to the Author):

Chen et al. have put significant effort in to revising their manuscript on FGF23/FGFR/aKlotho/HS complex structures, and have addressed most of the detailed concerns raised in my previous review. The asymmetric 1:2:1:1 complexes seen (from 1:1:1:1 mixtures) are quite interesting, and suggest a model in which Klotho serves to stabilize interaction of FGF23 with FGFR(P) at the membrane – allowing the resulting FGF23/FGFR(P) complex (with HS) to recruit a second FGFR (FGFR(S)) that makes no contacts at all with Klotho. The FGF23/FGFR(P)/Klotho structure here is essentially the same as that described in an earlier Nature paper by this group, and the main purpose of the Klotho is to recruit FGF23 to the FGFR(P) molecule. In the previous version of the manuscript, I was looking for reasons why the asymmetric FGFR dimer seen here would be specific to the situation with endocrine FGFs and Klotho – and suggested that the authors determine whether the same FGFR mutations that impair FGF23 – designed based on the asymmetry – also affect paracrine FGF signaling. In the rebuttal, the authors now describe experiments revealing that paracrine FGFs do use asymmetric FGFR dimers. They also now show (in rebuttal) structures of paracrine FGFs bound to an asymmetric FGFR dimer that is essentially the same as seen here. This is important.

Response: We thank the reviewer for valuing the importance of our discovery that asymmetric dimerization is universal to the entire FGF family (i.e., both paracrine and endocrine).

I am concerned that publishing the present paper without discussing in some detail the finding in the rebuttal that paracrine FGFs also stabilize asymmetric FGFR dimers will mislead the field. Other readers will – as I did in the first version of the manuscript – presume that the asymmetric FGFR dimers are somehow Klotho-specific and/or endocrine FGF specific. As far as I am aware, only 2:2 symmetric dimers were previously seen for paracrine FGFs (with HS), and the 1:2 (FGF23:FGFR) dimers (plus an HS and a Klotho) appear specific to the endocrine system if one reads this paper as it stands. But, the authors clearly state in their rebuttal that there is no such distinction to be made between endocrine and paracrine FGFs on that basis. That's a very important point.

It looks like Klotho has little or nothing to do with FGFR dimerization or the asymmetry (but the paper still implies that it does). Klotho just brings the FGF23 into position, making up for weaker HS binding in effect (and providing cell specificity). There is a very interesting distinction to be made based on HS and Klotho both acting as co-receptors by interacting with ligand and receptor to induce the same asymmetric dimers, but in different ways.

By hiding the fact that the paracrine FGFs (with HS) induce the same asymmetric FGFR dimers as does FGF23 (with HS + Klotho), I think the paper as written is confusing and should not be published in Nature. On the other hand, if the authors brought in the asymmetric dimers with paracrine FGFs into this paper, and focused on the differences in how HS alone (paracrine case) and HS + Klotho (endocrine case) promote the same FGFR dimer complex, that could be a highly valuable contribution.

I also think that the authors still spend too much time/text going through details of the interactions, which limits accessibility of the manuscript for those who are not structurally inclined, and I think detracts from the bigger picture. Much of this could be removed to focus on the commonality of the asymmetric FGFR dimers seen with endocrine and paracrine FGFs – achieved in slightly different ways and ultimately showcasing the role of Klotho in targeting FGF23 to particular FGFRs in specific cells (that have the Klotho). Indeed, the current title almost implies that this is what the present paper is about.

Response: To address reviewer's concern, we have revised the manuscript as follows:

- 1) We have introduced a new subsection entitled “Asymmetric FGFR dimerization is universal” under which we present all our cell-based data (provided in our rebuttal) that were generated to establish that paracrine FGFs also signal via asymmetric FGFR dimers. The associated data figures have been incorporated into revised manuscript as Extended Data Figures 8 to 11.
- 2) We have added a Concluding Remarks section along with a newly prepared model figure (Extended Data Figure 12 shown below) to discuss/illustrate commonalities and differences between endocrine and paracrine FGF mediated asymmetric receptor dimerization.

Extended Data Fig.12. Asymmetric receptor dimerization is universal mechanism in FGF signaling. **a**, Due to FGF hormone’s weak HS binding affinity, HS alone is incompetent in stabilizing endocrine FGF-FGFR complex and inducing sustained asymmetric receptor dimerization/activation. Blurring and loose association are used to emphasize the unstable/transient nature of putative FGF-FGFR-HS ternary complex and physiologically inconsequential receptor dimerization/activation. **b**, Membrane bound Klotho co-receptor simultaneously engages FGFR’s D3 domain and FGF’s C-terminal tail thereby stabilizing the endocrine FGF-FGFR complex within a ternary complex. In so doing, Klotho co-receptor effectively compensates for HS incompetency in stabilizing binary endocrine FGF-FGFR complex. HS is now in position to recruit a second FGFR to the stabilized binary complex thus inducing asymmetric dimerization. Nevertheless, due to FGF hormone’s weak HS binding affinity, Klotho and HS-induced endocrine FGF-FGFR dimers are still inferior to HS-induced paracrine 1:2 FGF-FGFR dimers in terms of longevity/stability (indicated by blurring of FGFR^S in **b**). Consequently, FGF hormones induce a weaker receptor activation/signaling relative to paracrine FGFs as posited by our “Threshold Model” for FGF signaling specificity¹. **c**, Due to their high HS binding affinities, paracrine FGFs can rely on HS as the sole co-receptor to stably bind primary FGFR and recruit a secondary FGFR, thereby inducing formation of rigid and long-lived asymmetric receptor dimers.

- 3) We have revised the Abstract to more clearly highlight the novelty of our work. Additionally, we now explicitly state that asymmetric dimerization is applicable to all FGFs and that this discovery overturns the current symmetric dimerization paradigm for FGF signaling.

- 4) The manuscript title has been revised to ‘Structural Basis for FGF Hormone Signaling’ which we believe more accurately captures the focus/findings of our study.
- 5) all detailed descriptions of the FGF-FGFR and FGFR-FGFR interfaces have been removed from the main text and instead are listed in Extended Data Table 2 and 3 as part of supplementary materials.

(Redacted) We believe that the cell-based data, which the Reviewer asked for in his/her initial review, are sufficiently conclusive in terms of establishing the extendibility of asymmetric dimer model to paracrine FGFs

Reference:

- 1 Zinkle, A. & Mohammadi, M. A threshold model for receptor tyrosine kinase signaling specificity and cell fate determination. *F1000Res* **7** (2018).

Referee #3 (Remarks to the Author):

The authors have put substantial effort to address my concerns that were raised in my initial review. The manuscript has been improved remarkably with new mutagenesis results and newly prepared figures. The role of the second klotho is still mysterious that may require further studies in the future. Overall, this is a nice piece of work, and is of high quality. I strongly support its publication at Nature.

Response: We thank the reviewer for his/her kind remarks and for supporting the publication of our work.

Reviewer Reports on the Second Revision:

Referees' comments:

Referee #1 (Remarks to the Author):

The authors have responded to my main concern that the use of asymmetric dimerization by FGF/FGFR without Klotho was 'hidden' in the previous version, which would be confusing. The authors now describe the likely universality of this asymmetric dimerization in the abstract and in the text, and this is an important finding. I do find it a little strange that there are no data or models pertaining to this in the main part of the manuscript (and there could be), but that is really an editorial decision. Personally, I might include one of the model figures from Ext Data figs 9a, 10a, 11, or 12c in Fig 5. It could easily be done as a lower set of panels in that figure, and would help the reader.

This is a terrific study with this inclusion, on which the authors should be congratulated.

Author Rebuttals to Second Revision:

Response to Reviewers

Title: Structural Basis for FGF Hormone Signaling

Manuscript: 2022-09-14066C

Referee #1 (Remarks to the Author):

The authors have responded to my main concern that the use of asymmetric dimerization by FGF/FGFR without Klotho was 'hidden' in the previous version, which would be confusing. The authors now describe the likely universality of this asymmetric dimerization in the abstract and in the text, and this is an important finding. I do find it a little strange that there are no data or models pertaining to this in the main part of the manuscript (and there could be), but that is really an editorial decision. Personally, I might include one of the model figures from Ext Data figs 9a, 10a, 11, or 12c in Fig 5. It could easily be done as a lower set of panels in that figure, and would help the reader.

This is a terrific study with this inclusion, on which the authors should be congratulated.

Response: Per reviewer's suggestion, we have incorporated the entire original Extended Data Figure 9 and panel a of the original Extended Data Figure 11 as new panels e through h in a revised Figure 5 (also appended below for reviewer's convenience). We agree with the reviewer that inclusion of these data panels in the main figure has improved the accessibility of the key message of the manuscript on the universality of this asymmetric dimerization. We thank the reviewer for his/her valuable suggestion and for supporting the publication of our work.

Figure 5

Fig. 5. Both endocrine and paracrine FGFs signal via asymmetric receptor dimers.